# Collaborative Mean Estimation Among Heterogeneous Strategic Agents: Individual Rationality, Fairness, and Truthful Contribution

**Alex Clinton** [1]  **Yiding Chen** [2]  **Xiaojin Zhu** [1]  **Kirthevasan Kandasamy** [1]

## Abstract

We study a collaborative learning problem where $m$ agents aim to estimate a vector $\mu = (\mu_1, \dots, \mu_d) \in \mathbb{R}^d$ by sampling from associated univariate normal distributions $\{\mathcal{N}(\mu_k, \sigma^2)\}_{k \in [d]}$. Agent $i$ incurs a cost $c_{i,k}$ to sample from $\mathcal{N}(\mu_k, \sigma^2)$. Instead of working independently, agents can exchange data, collecting cheaper samples and sharing them in return for costly data, thereby reducing both costs and estimation error. We design a mechanism to facilitate such collaboration, while addressing two key challenges: ensuring *individually rational (IR) and fair outcomes* so all agents benefit, and *preventing strategic behavior* (*e.g.*, non-collection, data fabrication) to avoid socially undesirable outcomes. We design a mechanism and an associated Nash equilibrium (NE) which minimizes the social penalty–sum of agents' estimation errors and collection costs–while being IR for all agents. We achieve a $\mathcal{O}(\sqrt{m})$–approximation to the minimum social penalty in the worst case and an $\mathcal{O}(1)$–approximation under favorable conditions. Additionally, we establish three hardness results: no nontrivial mechanism guarantees *(i)* a dominant strategy equilibrium where agents report truthfully, *(ii)* is IR for every strategy profile of other agents, *(iii)* or avoids a worst-case $\Omega(\sqrt{m})$ price of stability in any NE. Finally, by integrating concepts from axiomatic bargaining, we demonstrate that our mechanism supports fairer outcomes than one which minimizes social penalty.

## 1. Introduction

With the rise of machine learning, data has become a crucial resource for organizations. However, collecting data often involves significant costs and resources, such as experiments, simulations, surveys, or market research. If agents *share data*, it can be mutually beneficial, particularly when they have complementary data collection capabilities, as an agent can exchange cheap or easily accessible data for costly or hard-to-obtain data. For example, a hospital may be limited in the type of data it collects, due to local demographic and disease patterns, but values diverse data from other hospitals to improve treatments for all patients. Today, data sharing platforms exist in many sectors, including business (goo; aws; azu; del), science (pub, a; 53 & 68, 2013; cit; pub, b), and public institutions (mad; sfo; nyc).

**Incentive-related challenges.** However, data sharing raises two fundamental incentive-related challenges. *(i)* First, we must *divide the data collection work and distribute the collected data in an individually rational (no agent loses by participating) and fair way*. For example, while it would be most efficient to have agents with the cheapest costs collect all of the data, this would not be fair to such agents and even discourage them from participating. *(ii)* Second, we must *guard against strategic behavior*, ensuring agents cannot manipulate mechanisms for personal gain at others' expense. For example, naive protocols like pooling and sharing all agents' data can lead to free-riding, where an agent may contribute little data if others are expected to contribute more. Moreover, simple checks to prevent free-riding, such as counting the number of data submitted, can be easily bypassed by submitting fabricated (fake) data.

These considerations make defining individual rationality (IR) and fairness nontrivial. While some prior work address free-riding behaviors like non-collection and data fabrication, they assume homogeneous agents with identical collection costs. Moreover, the interplay between IR/fairness and strategic behavior presents novel challenges. In this work, we investigate these challenges in Gaussian mean estimation problem.

### 1.1. Summary of Contributions

**Model.** There are $m$ agents who seek to estimate an unknown vector $\mu \in \mathbb{R}^d$ using samples from $d$ associated univariate normal distributions. Agent $i$ incurs a cost $c_{i,k}$ to draw one sample from the $k^{\text{th}}$ distribution $\mathcal{N}(\mu_k, \sigma^2)$. An

---

[1]Department of Computer Sciences, University of Wisconsin-Madison, Madison, WI, USA [2]Department of Computer Science, Cornell University, Ithaca, New York, USA. Correspondence to: Alex Clinton <aclinton@wisc.edu>.

*Proceedings of the 42$^{nd}$ International Conference on Machine Learning*, Vancouver, Canada. PMLR 267, 2025. Copyright 2025 by the author(s).

agent's penalty (negative utility) is the sum of her estimation error and data collection costs. By sharing data with others, she can improve her estimate while reducing costs, especially by collecting data from distributions with low $c_{i,k}$ and exchanging for those with high $c_{i,k}$. We wish to design a mechanism *(without money)* to facilitate such sharing. In such a mechanism, agents will collect data from these distributions and submit it. The mechanism then returns, to each agent, an estimate for $\mu$, or more generally, information about $\mu$ from which she can estimate $\mu$.

**1. Problem formalism.** In §3, we formalize the three desiderata for a mechanism: *(1) individual rationality (IR):* every agent is no worse off than working on her own, *(2) incentive-Compatibility (IC):* agents are incentivized to collect sufficient data and report it truthfully, and *(3) efficiency:* the social penalty (sum of agent penalties) is minimized. A key challenge in formalizing this problem, novel to our setting, when agents have unequal costs, is that directly minimizing social penalty can violate IR as it demands the lowest-cost agents to collect most of the data, resulting in higher data collection costs for them compared to working alone. Therefore, we propose a baseline mechanism that achieves the smallest possible penalty among all IR mechanisms. However, this baseline is not IC. Our objective is therefore to construct a mechanism whose penalty approximates this baseline while ensuring both IR and IC.

**2. Mechanism.** In §4, we present our mechanism. To ensure truthful data reporting, we rely, partly, on techniques from prior work, which evaluate an agents' submission by comparing it to the submissions of others. While such an evaluation is straightforward when all agents collect similar data amounts from the same distribution, it is challenging in our setting where agents collect varying amounts of data from different distributions. To effectively apply this technique, we first modify the amount of data collected in the above baseline. Our modification ensures that, when possible, a sufficient number of agents collect data from each distribution, so that a mechanism has enough data from other agents when evaluating the submission of any given agent. However, this may require some high-cost agents to collect more data than in the baseline, increasing total costs. Hence, the modification must be carefully designed to prevent a significant increase in social penalty. We prove that our mechanism has a Nash equilibrium (NE) which achieves a $\mathcal{O}(\sqrt{m})$–approximation to the baseline in the worst-case, and an $\mathcal{O}(1)$–approximation under favorable conditions.

**3. Hardness results.** In §5 we supplement the above result with three hardness results. The first two show, for any efficient mechanism, we cannot hope for a dominant strategy equilibrium nor a guarantee of IR at every strategy profile of other agents. The third result characterizes the difficulty of our problem, showing that in the worst case,

the social penalty in any NE of any mechanism can be as large as a factor $\Omega(\sqrt{m})$ away from the baseline.

**4. Fairness considerations.** While minimizing social penalty subject to IR is our primary goal, it places most of the data collection burden onto agents with the lowest costs, leading to potential unfairness. In §6 we show that our mechanism can accommodate fairer divisions of work, which we formalize using concepts from axiomatic bargaining. To our knowledge, this integration of cooperative game theory for fair work allocation and noncooperative game theory for enforcement has not been explored in prior work.

### 1.2. Related Work

**Collaborative learning and truthful reporting.** Several prior work has studied free-riding issues in data sharing and collaborative learning (Blum et al., 2021; Karimireddy et al., 2022; Fraboni et al., 2021; Lin et al., 2019; Huang et al., 2023), but have assumed that agents contribute data truthfully. Dorner et al. (2024) study collaborative mean estimation among competitors and allow agents to alter their submissions, but their strategy space is restricted to two scalars that agents can use to modify the data they report. This is significantly simpler than our setting, where we allow an agent's submission to be as general to the extent possible, even allowing it to depend on the data she has collected. (Hardt et al., 2016) consider a strategic classification problem where participants may willingly incur a cost to manipulate their submission to obtain a better classification outcome. Others have explored mechanisms to enforce truthful reporting (Cai et al., 2015; Chen et al., 2020; Ghosh et al., 2014), employing methods akin to ours by comparing an agent's submission with those of others, but not for data sharing. They also do not formally study fabrication or falsification as part of the agent's strategy, as we do. Prior work has considered designing mechanisms to incentivize participation in collaborative learning (Donahue & Kleinberg, 2021; Capitaine et al., 2024; Tu et al., 2022) but do not address the issue of truthful reporting.

Our work is most closely related to Chen et al. (2023), who studied collaborative univariate normal mean estimation with strategic agents. There are two (orthogonal) challenges to incentivizing truthful contributions while maintaining an efficient system. The first, addressed by Chen et al. (2023), is designing methods to validate an agent's submission using others' data. The second, not addressed by Chen et al. (2023) and the focus of this paper, is ensuring that there is enough data from all agents so that each agent's submission can be sufficiently validated against the others, without compromising on efficiency (social penalty). Chen et al. (2023) avoid the second issue by assuming a *single distribution* and *equal costs* across agents, allowing equal data contribution and ensuring sufficient data from all agents. However, this

breaks down with heterogeneous collection costs. While we adopt a similar strategy to solve the first challenge, doing so requires overcoming technical hurdles. Consequently, our problem formulation in §3, mechanism and analysis in §4, and hardness results in §5 are novel and fundamental to the heterogeneous case.

**Cooperative game theory in data sharing.** Some apply coalitional game concepts, like Shapley value, to value data contributions of many agents (Sim et al., 2020; Jia et al., 2019; Xu et al., 2021). But, they do not study strategic agents attempting to manipulate a mechanism.

**Data collection with money.** Prior work has studied data collection protocols and data marketplaces where agents are incentivized to collect data via payments (Agarwal et al., 2019; 2024; Wang et al., 2020; Cai et al., 2015; Kong et al., 2020; Zhang et al., 2014). However, our focus is on mechanisms for data sharing *without money*.

## 2. Description of the Environment

We will begin with a description of the environment. For any $v \in \mathbb{R}^p$, let $\|v\|_2$ denote the $\ell^2$-norm. For any set $S$, let $\Delta(S)$ denote all probability distributions over $S$.

**Preliminaries.** The universe chooses a vector $\mu \in \mathbb{R}^d$. There are $m$ agents who wish to estimate $\mu$. They do not possess any auxiliary information, such as a prior, about $\mu$, but can collect samples from associated univariate normal distributions $\{\mathcal{N}(\mu_k, \sigma^2)\}_{k \in [d]}$. Agent $i \in [m]$ incurs cost $c_{i,k} \in (0, \infty]$ to draw a single sample from the $k^{\text{th}}$ distribution $\mathcal{N}(\mu_k, \sigma^2)$. If $c_{i,k} = \infty$, it means that the agent cannot sample from the $k^{\text{th}}$ distribution[1]. Let $c = \{c_{i,k}\}_{i \in [m], k \in [d]}$ denote all costs. We will assume that for each distribution there is at least one agent who can sample from it with finite cost. Let $\mathcal{C}$, defined below, be the set of such costs,

$$\mathcal{C} = \big\{ c = \{c_{i,k}\}_{i \in [m], k \in [d]} \in (0, \infty]^{m \times d} :$$
$$\forall k \in [d], \exists i \in [m] \text{ such that } c_{i,k} < \infty \big\}. \quad (1)$$

The variance $\sigma^2$ and costs $c$ are publicly known. While our results generalize to different variances across agents or distributions, we assume a common variance $\sigma$ for simplicity.

**Mechanism.** We now formally define a mechanism. The example in §3.2 will illustrate the ideas. Intuitively, a mechanism accepts data from agents and returns information they can use to estimate $\mu$. We first define "information".

*Information space.* An information space $\mathcal{I}$ is a set chosen by the mechanism designer. The information the mechanism returns to each agent is simply an element of $\mathcal{I}$. The simplest information space is $\mathcal{I} = \mathbb{R}^d$, where the mechanism returns

---

[1] Our model generalizes other models, where agents can either sample from a distribution or not (e.g., (Huang et al., 2023)), by setting $c_{i,k} = 1$ or $c_{i,k} = \infty$, respectively.

an estimate for $\mu$. While our mechanism also uses $\mathcal{I} = \mathbb{R}^d$, this general definition is necessary for our hardness results.

*Mechanism.* A mechanism is a tuple $(\mathcal{I}, b)$, where $\mathcal{I}$ is a chosen information space, and $b$ maps the datasets received from the agents, to some information, i.e. an element in $\mathcal{I}$, for each agent. Let $\mathcal{D} = \big( \bigcup_{n \geq 0} \mathbb{R}^n \big)^d$ be the space of datasets that can be submitted by a single agent—an agent may submit a different number of points from each distribution. Then, the space of mechanisms is $\mathcal{M} = \big\{ M = (\mathcal{I}, b) : b : \mathcal{D}^m \to \mathcal{I}^m \big\}$.

**Agent's strategy space.** A *pure strategy* of an agent $i$ consists of three components: *(i)* First, she will choose $n_i = \{n_{i,k}\}_k$, where $n_{i,k}$ indicates *how many samples* she will collect from distribution $k \in [d]$. She then collects her initial dataset $X_i = \{X_{i,k}\}_k$, where $X_{i,k} = \{x_{i,k,n}\}_{n=1}^{n_{i,k}}$ refers to the $n_{i,k}$ points collected from distribution $k$. *(ii)* She then submits $f_i(X_i) = Y_i = \{Y_{i,k}\}_k$, to the mechanism; the *submission function* $f_i$ maps the dataset collected by the agent to her submission. Here, $Y_{i,k} = \{y_{i,k,n}\}_n$ is ideally $X_{i,k}$, but a dishonest agent may choose to report untruthfully via fabrication, altering, or withholding data. Crucially, the agent's report can depend on the actual dataset $X_i$ she has collected, and when computing $Y_{i,k}$, she may use data from other distributions $\ell \neq k$. *(iii)* Once the mechanism returns the information $I_i$, the agent will estimate $\mu$ using all the information she has, which is her initial dataset $X_i$, her submission $Y_i$, and her information $I_i$ returned by the mechanism, via an *estimator* $h_i(X_i, Y, I_i)$. A *pure strategy* of an agent $i$ is therefore the 3-tuple $(n_i, f_i, h_i) \in \mathcal{S}$, where the strategy space $\mathcal{S} = \mathbb{N}^d \times \mathcal{F} \times \mathcal{H}$ and

$$\mathcal{F} = \big\{ f : \mathcal{D} \to \mathcal{D} \big\}, \quad \mathcal{H} = \big\{ h : \mathcal{D} \times \mathcal{D} \times \mathcal{I} \to \mathbb{R}^d \big\}. \quad (2)$$

A *mixed (randomized) strategy* $s_i \in \Delta(\mathcal{S})$ is a distribution over $\mathcal{S}$. An agent using a mixed strategy samples a pure strategy $(n_i, f_i, h_i) \sim s_i$ via an external random seed independent of data randomness, and executes it. Going forward, $s = \{s_i\}_i$ denotes the strategies of all agents and $s_{-i} = \{s_j\}_{j \neq i}$ denotes the strategies of all agents except $i$.

*Truthful submissions and accepting an estimate.* Let $\mathbf{id} \in \mathcal{F}$ denote the identity submission function, where an agent truthfully submits their collected data, i.e. $\mathbf{id}(X_i) = X_i$. We wish to incentivize each agent to use $f_i = \mathbf{id}$ so that others can benefit from the agent's submission. Next, in this work, we focus on the information set space $\mathcal{I} = \mathbb{R}^d$, where the mechanism returns an estimate for $\mu$ based on all agents' data. Let $h^{\text{ACC}}$ be the estimator which directly accepts the mechanism's estimate, i.e. $h^{\text{ACC}}(\cdot, \cdot, I_i) = I_i$. As we will see, naive mechanisms allow strategic agents to gain by misreporting data ($f_i \neq \mathbf{id}$) and/or altering the mechanism's estimate instead of accepting it ($h_i \neq h^{\text{ACC}}$).

**Agent's penalty.** An agent's penalty (negative utility) $p_i$ is the sum of her $L_2$–risk and data collection costs. As this

depends on the mechanism $M$ and strategies $s$, we write

$$p_i(M, s) := \sup_{\mu \in \mathbb{R}^d} \mathbb{E}\left[\|h_i(X_i, Y_i, I_i) - \mu\|_2^2 + \sum_{k=1}^d c_{i,k} n_{i,k}\right].$$
(3)

Here, the expectation accounts for randomness in the data, the mechanism, and agents' mixed strategies. We take supremum over all $\mu$ since $\mu$ is unknown, and strategies should achieve a small penalty for any $\mu$. For instance, if $\mu = \mu'$, for a fixed $\mu'$, the optimal strategy for an agent in any mechanism is to forgo data collection and choose an estimator which always outputs $\mu'$, i.e. $h_i(\cdot, \cdot, \cdot) = \mu'$, incurring zero penalty. However, this assumes prior knowledge of $\mu$. The supremum ensures the problem is well-defined, and is similar to maximum risk in frequentist statistics (Wald, 1939).

**The social penalty.** The social penalty $P(M, s)$ under a mechanism $M$ and a set of strategies $s$ is the sum of agent penalties, i.e. $P(M, s) = \sum_{i=1}^n p_i(M, s)$.

## 3. Problem Definition

**Goal.** Our goal is to design a mechanism–strategy pair $(M, s^\star) \in \mathcal{M} \times \mathcal{S}$ that incentivizes agents to follow $s^\star$ (incentive compatibility, IC), ensures agents are better off than acting independently (individual rationality, IR), and minimizes the social penalty $P(M, s^\star)$ (efficiency). In § 3.1 and § 3.2, we establish baselines for IR and efficiency, and in § 3.3, we formalize these three desiderata.

### 3.1. A Baseline for Individual Rationality

**Agent working individually.** To establish a baseline for IR, consider an agent acting on her own, who will collect some amount of data and estimate $\mu$ using this data. Let $s_i$ be a (mixed) strategy which chooses the number of data $n_i \in \mathbb{N}^d$ an agent will collect, and an estimator $g_i : \mathcal{D} \to \mathbb{R}^d$ for $\mu$ which uses her own data. The minimum penalty $p_i^{\text{IND}}$ an agent can achieve is,

$$p_i^{\text{IND}} := \inf_{s_i} \left(\sup_{\mu \in \mathbb{R}^d} \mathbb{E}\left[\|g_i(X_i) - \mu\|_2^2 + \sum_{k=1}^d c_{i,k} n_{i,k}\right]\right).$$

Here, the expectation is with respect to the data and any randomness in the agent's strategy. The following fact establishes an expression for $p_i^{\text{IND}}$; the proof is given in §A.1.

**Fact 1.** *We have*

$$p_i^{\text{IND}} = \begin{cases} 2\sigma \sum_{k=1}^d \sqrt{c_{i,k}} & \text{if } c_{i,k} < \infty \ \forall k, \\ \infty & \text{otherwise.} \end{cases}$$
(4)

When designing a mechanism and strategy profile $(M, s^\star)$, individual rationality will require that $p_i(M, s^\star) \leq p_i^{\text{IND}}$.

### 3.2. A Baseline for Efficiency: Minimizing Social Penalty While Satisfying Individual Rationality

Our goal in §3.2 is to establish a baseline for minimizing social penalty (though, as we will see, this baseline will not be IC). We begin by introducing a simple mechanism and a set of agent strategies that achieve low social penalty.

**The sample mean mechanism.** A simple example of a mechanism is one which estimates $\mu$ via the sample means, i.e. it pools the data from all agents from each distribution $\mathcal{N}(\mu_k, \sigma^2)$, takes the average to estimate each $\mu_k$, and then returns this estimate to all agents. For this mechanism, the information space is simply $\mathcal{I} = \mathbb{R}^d$. Let $M_{\text{SM}}$ denote this mechanism. Suppose, each agent $i$ reports $Y_i = \{Y_{i,k}\}_k$, where $Y_{i,k} = \{y_{i,k,n}\}_n$ is agent $i$'s submission for distribution $k$. Let $Y_{:,k} = \cup_{i \in [m]} Y_{i,k}$ be all the data for distribution $k \in [d]$ received from all agents. The mechanism computes an estimate $\widehat{\mu} \in \mathbb{R}^d$ for $\mu$, where $\widehat{\mu}_k = \frac{1}{|Y_{:,k}|} \sum_{y \in Y_{:,k}} y$ is the sample mean of all data received for distribution $k$. Finally, it returns this estimate to all agents, so $b(\{Y_i\}_{i \in [m]}) = \{I_i\}_{i \in [m]}$, where $I_i = \widehat{\mu}$ for all $i \in [m]$. While this mechanism is simple, it achieves the smallest possible penalty for given data collection amounts when all agents submit their data truthfully ($f_i = \text{id}$) and accept the estimate ($h_i = h^{\text{ACC}}$), as demonstrated in the fact below. Its proof is given in §A.2.

**Fact 2.** *Fix the data collection amounts of each agent $n = \{n_i\}_{i \in [m]}$. Then, among all mechanisms, and all possible submission functions and estimators agents could use, the sample mean mechanism, along with truthful submission and accepting the estimate achieves the smallest possible social penalty. It can be expressed as follows,*

$$\inf_{M,\ (f_j, h_j)_j} P(M, (n_j, f_j, h_j)_j) = P(M_{\text{SM}}, (n_j, \text{id}, h^{\text{ACC}})_j)$$

$$= \sum_{k=1}^d \left(\frac{m\sigma^2}{\sum_{i=1}^m n_{i,k}} + \sum_{i=1}^m c_{i,k} n_{i,k}\right).$$
(5)

**Challenges in social penalty minimization.** Before we proceed, we observe that naively minimizing social penalty, as is standard mechanism design, is problematic in our setting, where agents have different data collection costs. We will illustrate this with $d = 1$, where all agents wish to estimate a scalar quantity $\mu_1$. For the purpose of this illustration, let us assume that $c_{1,1} < \cdots < c_{m,1} < \infty$. Let us also assume, for now, that agents are *compliant*, i.e. will follow any set of strategies recommended by a mechanism designer (even if it is not the best strategy for them).

Suppose that the mechanism designer wishes to minimize the social penalty. As agents are compliant, by Fact 2, for any given data collection amounts $n = \{n_i\}_{i \in [m]}$, where $n_i \in \mathbb{N}$, she can use $M_{\text{SM}}$, and recommend agents to submit the data truthfully ($f_i = \text{id}$) and accept the estimate ($h_i = h^{\text{ACC}}$). For convenience let $s_n^{\text{COM}} :=$

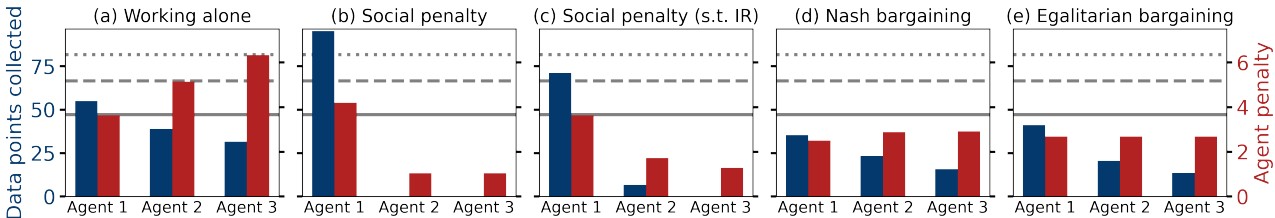

*Figure 1.* An illustration of different allocations of the data collection work in the sample mean mechanism. Here, $d = 1$, $c_{1,1} = 0.033$, $c_{2,1} = 0.066$, $c_{3,1} = 0.1$, and $\sigma = 10$. The blue and red columns represent the number of data collected and penalties incurred, respectively. The solid, dashed, and dotted gray lines indicate the penalties incurred by agents 1, 2, and 3 when working alone. Agents can significantly reduce their penalties by sharing data. However, minimizing the social penalty without IR constraints (Fig. (b)) leaves agent 1 worse off than working alone (Fig. (a)). While our primary objective is reducing social penalty, in §6, we explore axiomatic bargaining concepts to achieve fairer work allocations (Fig. (d) and (e)).

$\{(n_i, \mathbf{id}, h^{\mathrm{ACC}})\}_{i \in [m]}$ refer to the strategy profile where the agents follow the collection amounts $n$ and comply with the recommendations $f_i = \mathbf{id}$ and $h_i = h^{\mathrm{ACC}}$. To minimize social penalty, the mechanism designer can optimize for $n \in \mathbb{N}^{m \times 1}$. From Fact 2 this is equivalent to minimizing

$$P(M_{\mathrm{SM}}, s_n^{\mathrm{COM}}) = \frac{m\sigma^2}{\sum_{i=1}^m n_{i,1}} + \sum_{i=1}^m c_{i,1} n_{i,1}.$$

As agent 1 has the smallest cost, the RHS is minimized by assigning all the work to agent 1, i.e. $n_{1,1} = \sigma\sqrt{m/c_{1,1}}$, and $n_{i,1} = 0$ for all $i \neq 1$. However, as illustrated in Fig. 1 (b), this allocation of work benefits all others at the expense of agent 1. In fact, it is not even IR for agent 1 as she will be better off just working on her own Fig. 1 (a): her penalty will be $p_i(M_{\mathrm{SM}}, s^{\mathrm{COM}}) = \sigma^2/n_{1,1} + c_{1,1} n_{1,1} = (\sqrt{m} + 1/\sqrt{m})\sigma\sqrt{c_{1,1}} > 2\sigma\sqrt{c_{1,1}} = p_1^{\mathrm{IND}}$. It would be hopeless to aim for globally minimum penalty when agents are strategic, when it is not even IR when they are compliant.

**IR mechanisms and strategies.** A more attainable goal, and the one we take in this paper, is to minimize the social penalty subject to IR constraints. For this, let us define $\Theta_{\mathrm{IR}}$ to be all mechanism and strategy tuples $(M, s)$ that are individually rational for *all agents*. We have,

$$\Theta_{\mathrm{IR}} = \left\{ (M, s) \in \mathcal{M} \times \Delta(\mathcal{S})^m : p_i(M, s) \leq p_i^{\mathrm{IND}}, \forall i \in [m] \right\}. \tag{6}$$

**Baseline for efficiency.** We can now present our baseline for efficiency. To find a mechanism and strategy profile pair $(M, s)$ which minimizes the social penalty in $\Theta_{\mathrm{IR}}$, recall from Fact 2, that $(M_{\mathrm{SM}}, s_n^{\mathrm{COM}})$ minimizes the social penalty for *any* given $n \in \mathbb{N}^{m \times d}$. Therefore, we should find $n \in \mathbb{N}^{m \times d}$ to minimize $P(M_{\mathrm{SM}}, s_n^{\mathrm{COM}})$ subject to the IR constraints $p_i(M_{\mathrm{SM}}, s_n^{\mathrm{COM}}) \leq p_i^{\mathrm{IND}}$, $\forall i \in [m]$. Recall that $s_n^{\mathrm{COM}} := \{(n, \mathbf{id}, h^{\mathrm{ACC}})\}_{i \in [m]}$ and that $p_i^{\mathrm{IND}}$ is from (4). From writing this explicitly in terms of $n$ and using Fact 2, if $n^{\mathrm{OPT}}$ minimizes the optimization problem below, it follows that $(M_{\mathrm{SM}}, s_{n^{\mathrm{OPT}}}^{\mathrm{COM}}) = \operatorname{argmin}_{(M,s) \in \Theta_{\mathrm{IR}}} P(M, s)$.

$$\operatorname*{minimize}_n \sum_{k=1}^d \left( \frac{m\sigma^2}{\sum_{j=1}^m n_{j,k}} + \sum_{i=1}^m c_{i,k} n_{i,k} \right) \quad \text{s.t}$$

$$\sum_{k=1}^d \left( \frac{\sigma^2}{\sum_{j=1}^m n_{j,k}} + c_{i,k} n_{i,k} \right) \leq p_i^{\mathrm{IND}}, \quad \forall i \in [m]. \tag{7}$$

Since the optimization problem above is convex, it can be solved using common optimization libraries. We have illustrated an example of the solution, $n^{\mathrm{OPT}}$, in Fig. 1 (c).

**From compliant to strategic agents.** While $(M_{\mathrm{SM}}, s_{n^{\mathrm{OPT}}}^{\mathrm{COM}})$ is IR for all agents and minimizes the social penalty subject to IR constraints, it is not incentive-compatible, as some agents can collect no data to avoid costs, and simply just receive the estimates from the mechanism. For instance, in Fig. 1 (c), even if agent 2 does not collect any data, her penalty will be small as agent 1 is already collecting a sufficiently large amount of data (details provided in §C).

Even a modification of $M_{\mathrm{SM}}$ which counts the amount of data an agents submits to verify $|Y_{i,k}| = n_{i,k}^{\mathrm{OPT}}$ will not work as agents can simply fabricate data. Specifically, instead of collecting $\{n_{i,k}^{\mathrm{OPT}}\}_k$ points, she can collect 0 points, submit as many fabricated (fake) points via $\{Y_{i,k}\}_k$, and appropriately discount the fabricated dataset from her estimator $h_i = \{h_{i,k}\}_k$. That is by choosing $h_{i,k}(\varnothing, Y_i, I_i) = (n_{:,k} \cdot I_{i,k} - \mathrm{sum}(Y_{i,k}))/(n_{:,k} - |Y_{i,k}|)$, where $n_{:,k} = \sum_{j \neq i} n_{j,k}$. While this behavior benefits agent $i$, it hurts others who may use this data.

Hence, our goal (formalized in §3.3), is to design $(M, s^\star) \in \Theta_{\mathrm{IR}}$ so that it is the best strategy for agents to follow $s^\star$ in $M$, and so that $P(M, s^\star)$ approximates $P(M_{\mathrm{SM}}, s_{n^{\mathrm{OPT}}}^{\mathrm{COM}})$.

**Fairness considerations.** Before we proceed, it is worth pointing out, that while $(M_{\mathrm{SM}}, s_{n^{\mathrm{OPT}}}^{\mathrm{COM}})$ is IR for all agents, the outcomes may not necessarily be fair. For instance, in Fig. 1 (c), the penalty of agent 1 is the same as when compared to working alone; while she is not worse off anymore, she has not befitted from collaboration, while the others have. Hence, we may consider other baselines which produce more fair outcomes, even at higher social penalty.

In §6, we use approaches from axiomatic bargaining to define fairer baselines, and show how our mechanism in §4 can be straightforwardly extended to such situations.

### 3.3. Desiderata for Mechanism Design

Our goal is to design a mechanism and strategy profile pair $(M, s^\star) \in \mathcal{M} \times \mathcal{S}$, where agents are collecting a sufficient amount of data and reporting it truthfully, so as to satisfy the following three requirements:

1. *Nash incentive-compatibility (NIC):* $(M, s^\star)$ is NIC if $s^\star$ is a Nash equilibrium (NE) in the mechanism $M$. That is, $\forall i \in [m], p_i(M, (s_i^\star, s_{-i}^\star)) \leq p_i(M, (s_i, s_{-i}^\star))$ for all other $s_i \in \Delta(\mathcal{S})$.

2. *Individual rationality (IR):* $(M, s^\star)$ is IR if each agent's penalty at $s^\star$ is no worse than the smallest penalty $p_i^{\mathrm{IND}}$ she could achieve independently (4). That is, $\forall i \in [m], p_i(M, s^\star) \leq p_i^{\mathrm{IND}}$.

3. *Efficiency:* We say that $(M, s^\star)$ is $\rho$-efficient if it satisfies $P(M, s^\star) \leq \rho \cdot \inf_{(M', s') \in \Theta_{\mathrm{IR}}} P(M', s')$.

**DSIC, IR regardless of other agent behavior.** Before we proceed, we note that in the first two requirements, other agents are assumed to follow the recommended strategy, $s_{-i} = s_{-i}^\star$. One could consider a dominant-strategy incentive-compatibility (DSIC) condition where following $s_i^\star$ is the best strategy for agent $i$ regardless of others' strategies. Similarly, one could consider a stronger IR condition where $p_i(M, (s_i^\star, s_{-i}')) \leq p_i^{\mathrm{IND}}$, for all other agent strategies $s_{-i}'$. However, in §5, we will establish hardness results which demonstrate that any mechanism satisfying either of these properties will be very inefficient.

## 4. Mechanism and Theoretical Results

We will now describe our mechanism, $M_{\mathrm{CBL}}$ (Corrupt Based on Leverage), outlined in Algorithm 2. As an argument, it takes a collection scheme $n \in \mathbb{N}^{m \times d}$. Since our goal is to minimize social penalty, throughout this section we should think of and will take $n = n^{\mathrm{OPT}}$ (see (7)). For the purpose of generalizing our mechanism to fairer solutions in §6, we will continue to write $n$. Our mechanism sets $\mathcal{I} = \mathbb{R}^d$ to be the information space, as it returns an estimate of $\mu$.

*Computing an enforceable division of work.* To ensure truthful data reporting, we adapt techniques from prior work, which evaluate an agents' submission by comparing it to the submissions of others (e.g (Chen et al., 2023; Cai et al., 2015)). This is straightforward when all agents collect similar data amounts from the same distribution, as there will be enough data from others to evaluate any given agent's submission. However, in $n^{\mathrm{OPT}}$, agents with the lowest costs for a distribution typically collect most of the data for that distribution (see Fig. 1(c)). To address this, Algorithm 2

first modifies $n$ $(= n^{\mathrm{OPT}})$ using the Compute-$n$-Approx subroutine, so as to enable such evaluations when possible.

Compute-$n$-Approx consists of the following high level steps. 1) Initialize the approximation $\widetilde{n}$ to be $n$. 2) Continually check if there is an agent $i$ whose penalty for distribution $k$ when working alone $\left(\frac{\sigma^2}{n_{i,k}^{\mathrm{IND}}} + c_{i,k} n_{i,k}^{\mathrm{IND}}\right)$ is already within a factor of 4 of their estimation error when all agents are compliant and follow $\widetilde{n}$, then update $\widetilde{n}$ so that agent $i$ only collects $n_{i,k}^{\mathrm{IND}}$ points. 3) Record which indices of $n$ have been updated via the sets $V_k$. 4) Update $\widetilde{n}$ to ensure that agents whose collection amount has not been modified are not collecting too small a fraction of the total data. 5) Return the approximation $\widetilde{n}$, $(V_k)_{k=1}^d$, and the total amount of data under $\widetilde{n}$ for each distribution, $(T_k)_{k=1}^d$. If $i \notin V_k$, then agent $i$ is reliant on others for data from distribution $k$. This will turn out to be a key component of determining which agents we can incentivize to collect data according to $n_{i,k}$. If $i \in V_k$ then it means that agent $i$ can achieve a good penalty on distribution $k$ without much help from others. The point of calculating $\widetilde{n}$ is to find a division of work which is enforceable (each agent is incentivized to follow it assuming the others do so).

---

**Algorithm 1** Compute-$n$-Approx

1: **Inputs:** collection scheme $n \in \mathbb{N}^{m \times d}$
2: $\widetilde{n} \leftarrow n$ # $\widetilde{n}$ will be an approximation to $n$ for which truthful reporting is enforceable.
3: $V_k \leftarrow \varnothing \quad \forall k \in [d]$ # Agents who will receive no new data from others for distribution $k$.
4: **while** $\exists (i, k) \in [m] \backslash V_k \times [d]$ such that $\frac{\sigma^2}{n_{i,k}^{\mathrm{IND}}} + c_{i,k} n_{i,k}^{\mathrm{IND}} \leq 4 \left(\frac{\sigma^2}{\sum_{j=1}^m \widetilde{n}_{j,k}} + c_{i,k} \widetilde{n}_{i,k}\right)$ **do**
5: $\quad \widetilde{n}_{i,k} \leftarrow n_{i,k}^{\mathrm{IND}}$
6: $\quad V_k \leftarrow V_k \cup \{i\}$
7: $T_k \leftarrow \sum_{i=1}^m \widetilde{n}_{i,k} \quad \forall k \in [d]$
8: **while** $\exists (i, k) \in [m] \backslash V_k \times [d]$ such that $\widetilde{n}_{i,k} > 0$ and $\widetilde{n}_{i,k} T_k < \left(n_{i,k}^{\mathrm{IND}}\right)^2$ **do**
9: $\quad \widetilde{n}_{i,k} \leftarrow \frac{\left(n_{i,k}^{\mathrm{IND}}\right)^2}{T_k}$
10: Return $\widetilde{n}, (V_k)_{k=1}^d, (T_k)_{k=1}^d$

---

*Algorithm 2 walk through.* After computing the enforceable approximation $\widetilde{n}$, the mechanism designer publishes the mechanism and the recommended strategies $s^\star$ (which we will define shortly). Each agent then chooses their strategy (not necessarily following $s^\star$) before collecting and submitting their data to the mechanism. After receiving agent submissions, Algorithm 2 determines if based on the input, the following favorable condition holds for $(c, n)$:

$$\forall i, k \quad \frac{\sigma^2}{n_{i,k}^{\mathrm{IND}}} + c_{i,k} n_{i,k}^{\mathrm{IND}} \geq \frac{\sigma^2}{\sum_{j=1}^m n_{j,k}} + c_{i,k} n_{i,k}. \quad (8)$$

---

**Algorithm 2** $M_{\text{CBL}}$

---

1: **Inputs:** collection scheme $n \in \mathbb{N}^{m \times d}$        # For minimizing social penalty, take $n = n^{\text{OPT}}$ (7)

2: **Mechanism designer:**

3:      $n^\star, (V_k)_{k=1}^d, (T_k)_{k=1}^d \leftarrow$ Compute-$n$-Approx$(n)$

4:      Publish the information space $\mathcal{I} = \mathbb{R}^d$, the mechanism (lines 9–29), and the recommended strategies $s^\star$ in (9).

5: **Each agent $i$:**

6:      Choose a strategy $s_i \in \Delta(\mathcal{S})$ and use it to select $\left((n_{i,k})_k, f_i, h_i\right) \sim s_i$.

7:      Sample $\{n_{i,k}\}_k$ points from the distributions to collect $X_i = \{X_{i,k}\}_k$, where $X_{i,k} \in \mathbb{R}^{n_{i,k}}$.

8:      Submit $Y_i = (Y_{i,1}, \ldots, Y_{i,d}) = f_i(X_{i,1}, \ldots, X_{i,d})$ to the mechanism.

9: **Mechanism:**

10:      $Y_{-i,k} \leftarrow \bigcup_{j \neq i} Y_{j,k} \quad \forall (i,k) \in [m] \times [d]$

11:      **if** the conditions in (8) are satisfied **then**

12:          **for** $(i,k) \in [m] \times [d]$ **do**

13:              **if** $i \in V_k$ **then**

14:                  $I_{i,k} \leftarrow \frac{1}{|Y_{i,k}|} \sum_{y \in Y_{i,k}} y$        # Use sample mean of only the agent's data.

15:              **else**

16:                  **if** $n^\star_{i,k} = 0$ **then**

17:                      $I_{i,k} \leftarrow \frac{1}{|Y_{-i,k}|} \sum_{y \in Y_{-i,k}} y$        # Use sample mean of others' data.

18:                  **else**

19:                      $Y'_{-i,k} \leftarrow$ sample $T_k - n^\star_{i,k}$ points from $Y_{-i,k}$ without replacement

20:                      $Z_{i,k} \leftarrow$ sample $\min\left(\left|Y'_{-i,k}\right|, n^\star_{i,k}\right)$ points from $Y'_{-i,k}$ without replacement

21:                      $\eta^2_{i,k} \leftarrow \alpha^2_{i,k} \left(\frac{1}{|Y_{i,k}|} \sum_{y \in Y_{i,k}} y - \frac{1}{|Z_{i,k}|} \sum_{z \in Z_{i,k}} z\right)^2$        # See (11) for $\alpha_{i,k}$.

22:                      $Z'_{i,k} \leftarrow \left\{z + \varepsilon_z : z \in Y'_{-i,k} \setminus Z_{i,k}, \ \varepsilon_z \sim \mathcal{N}(0, \eta^2_{i,k})\right\}$ # Lines 20–23 adapted from (Chen et al., 2023)

23:                      $I_{i,k} \leftarrow \dfrac{\frac{1}{\sigma^2 |Y_{i,k} \cup Z_{i,k}|} \sum_{y \in Y_{i,k} \cup Z_{i,k}} y + \frac{1}{(\sigma^2 + \eta^2_{i,k})|Z'_{i,k}|} \sum_{z \in Z'_{i,k}} z}{\frac{1}{\sigma^2} |Y_{i,k} \cup Z_{i,k}| + \frac{1}{\sigma^2 + \eta^2_{i,k}} \left|Z'_{i,k}\right|}$

24:      **else**

25:          **for** $(i,k) \in [m] \times [d]$ **do**

26:              **if** $i \in \arg\min_j c_{j,k}$ **then**

27:                  $I_{i,k} \leftarrow \frac{1}{|Y_{i,k}|} \sum_{y \in Y_{i,k}} y$        # Use sample mean of only the agent's data.

28:              **else**

29:                  $I_{i,k} \leftarrow \frac{1}{|Y_{-i,k}|} \sum_{y \in Y_{-i,k}} y$        # Use sample mean of others' data.

30: **Each agent $i$:**

31:      Post-process the estimates via the estimator function $h_i(X_i, Y_i, I_i)$.

---

This condition checks if, for each distribution, only sharing data for that distribution according to $n$ is better for an agent than working on her own, when all agents are compliant. This condition is favorable because, when (8) is satisfied, we have enough data to validate the data of every agent that relies on the mechanism to achieve a good penalty, and hence we will be able to obtain a good bound on the efficiency.

If (8) does not hold, for each distribution, we have the agent with the lowest collection cost, collect their individually rational amount of data for that distribution. The sample mean of this data, for each distribution, is then returned to all agents.

When (8) holds, Algorithm 2 checks two conditions. First

if $i \in V_k$ then agent $i$ receives the sample mean of only their data for distribution $k$ (which is fine as $i \in V_k$ implies that agent $i$ achieves a good penalty for distribution $k$ working alone). Second, if $n^\star_{i,k} = 0$ then there is no data to validate so $i$ receives the sample mean using the data other agents submitted for distribution $k$. If neither of these conditions hold, the algorithm begins running a corruption process similar to Chen et al. (2023) and returns the weighted average of the clean data agent $i$ submitted and corrupted data (created from the data of the other agents) for $I_{i,k}$. The corruption coefficients used in this process, $\alpha_{i,k}$, are defined in §B. In our proof, we show that this choice for $\alpha_{i,k}$ ensures this penalty is minimized at the value of $\widetilde{n}_{i,k}$ returned by Algorithm 1.

**Recommended strategies.** We now define the recommended strategies $s^\star$. Let $(\widetilde{n}_{i,k})_{k=1}^d$ denote the values re-

turned by Algorithm 1 when Compute-$n$-Approx($n$) is executed in Algorithm 2. In the following definition, the superscript L indicates leverage, as condition (8) implies that the mechanism has enough data from the other agents (when submitting truthfully) to incentivize sufficient data collection from the remaining agent. On the other hand, NL denotes no leverage. Define $s_i^\star := (n_i^\star, f_i^\star, h_i^\star)$ where $f_i^\star = \mathbf{id}$, $h_i^\star = h^{\mathrm{ACC}}$, and

$$
n_i^\star = \begin{cases} \left(n_{i,k}^{\mathrm{L}}\right)_{k=1}^d & \text{if } (c,n) \text{ satisfies } (8) \\ \left(n_{i,k}^{\mathrm{NL}}\right)_{k=1}^d & \text{otherwise} \end{cases}, \quad \text{where,} \quad (9)
$$

$$
n_{i,k}^{\mathrm{L}} := \widetilde{n}_{i,k} \qquad n_{i,k}^{\mathrm{NL}} := \begin{cases} n_{i,k}^{\mathrm{IND}} & \text{if } i \in \mathrm{argmin}_j\, c_{j,k} \\ 0 & \text{otherwise} \end{cases}.
$$

We now state the first theoretical result of this paper, which gives the properties of Algorithm 2 and the recommended strategies $s^\star$ in (9). In §5, we show that the worst-case $\Omega(\sqrt{m})$ bound on the price of stability is unavoidable.

**Theorem 1.** *Let $c \in \mathcal{C}$ and Algorithm 2 be executed with and $n = n^{\mathrm{OPT}}$ (see (1), (7)). Then, $(M_{\mathit{CBL}}, s^\star)$ is NIC, IR, and $\sqrt{m}$–efficient. Moreover, if $\left(c, n^{\mathrm{OPT}}\right)$ satisfies (8), then $(M_{\mathit{CBL}}, s^\star)$ is 8–efficient.*

**On condition (8).** Condition (8) can be interpreted as follows: it is true if, for each distribution, we separately allowed agents to share data for that distribution according to $n\ (= n^{\mathrm{OPT}})$, would agents be better than working on their own. For example, this is easily satisfied if all agents have the same or similar costs for each distribution. More generally, any problem instance where agents 1) collect no more data for each distribution than they would individually and 2) receive at least as much data for each distribution as they would collect individually, also satisfies (8).

**Proof challenges.** Beyond the design challenges highlighted above, we mention some of the technical challenges in our proof. First, we show that (8) and $i \notin V_k$ form a sufficient condition to guarantee the existence of $\alpha_{i,k}$ (as defined previously), and then lower bound $\alpha_{i,k}$ by $\sqrt{n_{i,k}^\star}$. This ensures that for each agent it is exactly optimal to collect $n_{i,k}^\star$ points and contribute them truthfully. Next, we need to show that collecting $n_{i,k}^\star$ points is also efficient for the agents. In particular, penalizing agents too severely via large $\alpha_{i,k}$ may be necessary to incentivize truthful contribution but result in poor efficiency.

## 5. Hardness Results

Theorem 1 shows that $s_i^\star$ is a NE and that at this NE, each agent is better off than working on her own. As discussed in §3.3, we may consider stronger IR and IC conditions, which hold regardless of other agents' strategies. In particular, is there an *efficient* mechanism-strategy pair $(M, s)$

which is either (1) *always IR*: $\forall i \in [m]$, $\forall s_{-i}' \in \Delta(\mathcal{S})^{m-1}$, $p_i\left(M, \left(s_i, s_{-i}'\right)\right) \le p_i^{\mathrm{IND}}$, or (2) *a dominant strategy profile*: $\forall i \in [m]$, $\forall s_i' \in \Delta(\mathcal{S})$, $\forall s_{-i}' \in \Delta(\mathcal{S})^{m-1}$, $p_i\left(M, \left(s_i, s_{-i}'\right)\right) \le p_i\left(M, \left(s_i', s_{-i}'\right)\right)$. The following two theorems answers these in the negative, showing that *every* agent will incur large penalty at *every* strategy profile.

**Theorem 2.** *For any $c \in \mathcal{C}$, if $M$ is a mechanism under which $s$ is always IR then $\forall i \in [m]$, $\forall s_{-i}' \in \Delta(\mathcal{S})^{m-1}$, $p_i\left(M, \left(s_i, s_{-i}'\right)\right) \ge \frac{p_i^{\mathrm{IND}}}{2}$.*

**Theorem 3.** *For any $c \in \mathcal{C}$, if $M$ is a mechanism under which $s$ is a dominant strategy profile then $\forall i \in [m]$, $\forall s_{-i}' \in \Delta(\mathcal{S})^{m-1}$, $p_i\left(M, \left(s_i, s_{-i}'\right)\right) \ge \frac{p_i^{\mathrm{IND}}}{2}$.*

Hence, no agent performs better than a factor $1/2$ compared to working alone, implying agent penalties remains large and do not decrease with the number of agents $m$. In contrast, agent penalties under $(M_{\mathrm{SM}}, n^{\mathrm{OPT}})$ and our mechanism typically decreases with $m$. To illustrate this, suppose $d = 1$ and $c_{1,i} = c'$. The minimum social penalty is $2\sigma\sqrt{c'm}$, with agents incurring a penalty of $\frac{2\sigma\sqrt{c'}}{\sqrt{m}}$. For strategic agents, as (8) is satisfied, our mechanism achieves a social penalty of $16\sigma\sqrt{c'm}$ and an agent penalty of $\frac{16\sigma\sqrt{c'}}{\sqrt{m}}$. In contrast, the lower bound from Theorems 2 and 3 gives a social penalty of at least $2\sigma m\sqrt{c'}$ and an agent penalty of $2\sigma\sqrt{c'}$, both of which are $\mathcal{O}(\sqrt{m})$ larger than those in our mechanism.

Our fourth theorem establishes that an $\Omega(\sqrt{m})$ price of stability is unavoidable in the worst case. While Theorems 2 and 3 also apply to homogeneous settings, this result is fundamental to the heterogeneous setting with multiple distributions and varying agent data collection costs.

**Theorem 4.** *There exists a set of costs $c \in \mathcal{C}$ such that the following is true. Let $\Theta_{\mathrm{IR}}$ be as defined in (6) for these costs $c$. For any mechanism $M \in \mathcal{M}$ and any Nash equilibrium $s^\star \in \Delta(\mathcal{S})$ of this mechanism, we have*

$$
P(M, s^\star) \ge \Omega(\sqrt{m}) \cdot \inf_{(M', s') \in \Theta_{\mathrm{IR}}} P(M', s').
$$

## 6. Fairness

Thus far we have focused on designing $(M, s^\star) \in \Theta_{\mathrm{IR}}$ to minimize social penalty. While this ensures IR, it is still unfair for agents with small data collection costs, as they will collect most, if not all the data. For example, in Fig. 1 (c), Agent 1 is doing only as well as working on her own, while others reap all the benefits. However, one can leverage ideas from axiomatic bargaining, to choose fairer divisions of work. Some common bargaining solutions include Nash bargaining (Nash et al., 1950), egalitarian bargaining (Mas-Colell et al., 1995), and the Kalai-Smorodinsky solution (Kalai & Smorodinsky, 1975). We have reviewed

these solutions in detail in §C. Both of these solutions correspond to solving simple optimization problems. In Fig. 1 (d) and (e), we give examples of these solutions. In both cases, all agents benefit more fairly even though the social penalty is worse (by definition).

**Specifying a bargaining solution.** Fortunately, our mechanism can also accommodate such fair divisions of work, while still enforcing truthful reporting. To formalize this, recall the sample mean mechanism $M_{\text{SM}}$ (§3.2), the compliant strategy profile $s_n^{\text{COM}}$ where agents collect data according to $n \in \mathbb{N}^{m \times d}$ (§3.2), and an agent's penalty when working alone $p_i^{\text{IND}}$ (Fact 1). Now define,

$$\mathcal{N} = \left\{ n \in \mathbb{N}^{m \times d} : p_i(M_{\text{SM}}, s_n^{\text{COM}}) \leq p_i^{\text{IND}} \;\; \forall i \in [m] \right\}. \tag{10}$$

We refer to any $n^{\text{B}} \in \mathcal{N}$ as a *bargaining solution*. It specifies an individually rational division of work, i.e. how much data each agent should collect from each distribution, when all agents comply in the sample mean mechanism. In fact, $\mathcal{N}$ is precisely the utility possibility set in bargaining (Mas-Colell et al., 1995). By definition $n^{\text{OPT}} \in \mathcal{N}$ (see (7)), but a mechanism designer may prefer a fairer bargaining solution.

**Bargaining-specific approximations.** If the goal is to implement a bargaining solution $n^{\text{B}}$ other than $n^{\text{OPT}}$, in place of efficiency, we should consider a new notion measuring how effectively the mechanism implements $n^{\text{B}}$ in terms of agent penalties. We say that $(M, s^{\star})$ $\rho$–approximates $n^{\text{B}} \in \mathcal{N}$, if $\forall i \in [m]$, we have $p_i(M, s^{\star}) \leq \rho \cdot p_i(M_{\text{SM}}, s_{n^{\text{B}}}^{\text{COM}})$, i.e. all agents are within a factor $\rho$ of the penalty they would incur if they collected data according to $n^{\text{B}}$ and complied in the sample mean mechanism. The following theorem states that when $n = n^{\text{B}}$ satisfies the same favorable condition in (8), we can guarantee efficient data sharing.

**Theorem 5.** *Let $c \in \mathcal{C}$ and $n^{\text{B}} \in \mathcal{N}$ (see (1), (10)). Suppose Algorithm 2 is executed with $n = n^{\text{B}}$. Then, $(M_{\text{CBL}}, s^{\star})$ is NIC and IR. Moreover, if $(c, n^{\text{B}})$ satisfies the condition in (8), then $(M_{\text{CBL}}, s^{\star})$ is an 8–approximation to $n^{\text{B}}$.*

## 7. Conclusion

We study collaborative mean estimation where strategic agents estimate a vector $\mu \in \mathbb{R}^d$ by sampling from distributions with different costs. We design an IR, NIC, and efficient mechanism, supported by matching hardness results, and show that it accommodates fairer divisions of work without additional modifications.

## Acknowledgements

This work was partially supported by NSF grant 2441796.

## Impact Statement

The paper focuses on providing algorithms to enable multiple parties to benefit from data sharing. As the paper is primarily theoretical in nature we do not feel the need to highlight particular societal and ethical consequences beyond the standard considerations, concerns, and practices that accompany sharing data.

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

# A. Proofs of referenced facts

### A.1. Proof of Fact 1

Let us first consider agents who can sample from all distributions, i.e $c_{i,k} < \infty$ for all $k \in [d]$. Suppose this agent collects $X_i$, via $n_i = \{n_{i,k}\}_k$ samples from each distribution, incurring cost $\sum_k c_{i,k} n_{i,k}$. It can be shown using standard arguments (Lehmann & Casella, 2006), that for any such $n_{i,k}$, the *deterministic* sample mean $h^{\mathrm{SM}} = \{h_k^{\mathrm{SM}}\}_k$, where $h_k^{\mathrm{SM}}(X_i) = \frac{1}{|X_{i,k}|} \sum_{x \in X_{i,k}} x$, which simply takes the average of all points from distribution $k$, is minimax optimal. Moreover, for any $\mu \in \mathbb{R}^d$, the expected squared estimation error for each $k$ is $\mathbb{E}[(h_k^{\mathrm{SM}}(X_i) - \mu_k)^2] = \sigma^2 / n_{i,k}$. Consequently, $\mathbb{E}[\|h^{\mathrm{SM}}(X_i) - \mu\|_2^2] = \sum_{k=1}^{n} \sigma^2 / n_{i,k}$. A rational agent will choose $n_i = \{n_{i,k}\}_k$ to minimize her penalty $\sum_{k=1}^{d} \sigma^2 / n_{i,k} + c_{i,k} n_{i,k}$. As this decomposes, a straightforward calculation shows that her penalty is minimized by *deterministically* choosing $n_{i,k} = \sigma / \sqrt{c_{i,k}}$ samples from each distribution $k$. For convenience, let $n_i^{\mathrm{IND}}$ denote these choices for $n_{i,k}$. Next, consider agents for whom $c_{i,k'} = \infty$ for some $k' \in [d]$. As this agent cannot obtain any samples from distribution $k'$ on her own, her maximum risk in estimating $\mu_{k'}$ is infinite, and consequently $\sup_\mu \mathbb{E}[\|h_i(X_i, \varnothing, \varnothing) - \mu\|_2^2] = \infty$ for any estimator $h_i$. Therefore, their penalty will also be infinite. Putting these together yields the claim.

### A.2. Proof of Fact 2

By definition

$$P(M_{\mathrm{SM}}, (n_j, \mathbf{id}, h^{\mathrm{ACC}})_j) = \sum_{i=1}^{m} \sup_{\mu \in \mathbb{R}^d} \left( \mathbb{E}\left[ \|h^{\mathrm{ACC}}(X_i, \mathbf{id}, \widehat{\mu}) - \mu\|_2^2 + \sum_{k=1}^{d} c_{i,k} n_{i,k} \right] \right)$$
$$= \sum_{i=1}^{m} \sup_{\mu \in \mathbb{R}^d} \mathbb{E}\left[ \|h^{\mathrm{ACC}}(X_i, \mathbf{id}, \widehat{\mu}) - \mu\|_2^2 \right] + \sum_{i=1}^{m} \sum_{k=1}^{d} c_{i,k} n_{i,k}.$$

Since we are taking $n = \{n_i\}_{i \in [m]}$ to be fixed, minimizing the social penalty is equivalent to minimizing the first term in the second line. It can be shown using standard arguments (Lehmann & Casella, 2006), that for any $n$, the deterministic sample mean of the collected points $h^{\mathrm{SM}} = \{h_k^{\mathrm{SM}}\}_k = \left\{ \frac{1}{|X_{:,k}|} \sum_{x \in X_{:,k}} x \right\}_k$ (where $X_{:,k} = \cup_{i \in [m]} X_{i,k}$) is the is minimax optimal. Therefore,

$$\sum_{i=1}^{m} \sup_{\mu \in \mathbb{R}^d} \mathbb{E}\left[ \|h^{\mathrm{ACC}}(X_i, \mathbf{id}, \widehat{\mu}) - \mu\|_2^2 \right] \geq \sum_{i=1}^{m} \sup_{\mu \in \mathbb{R}^d} \mathbb{E}\left[ \|h^{\mathrm{SM}} - \mu\|_2^2 \right]$$

But notice that when agents submit truthfully, the sample mean of the collected points is the same as the sample mean of the submitted so $h^{\mathrm{ACC}}(X_i, \mathbf{id}, \widehat{\mu}) = h^{\mathrm{SM}}$. Therefore the lower bound above is achieved using $M_{\mathrm{SM}}$ when agents submit truthfully and accept the estimate so we conclude

$$P(M_{\mathrm{SM}}, (n_j, \mathbf{id}, h^{\mathrm{ACC}})_j) = \inf_{M, (f_j, h_j)_{j \in [m]}} P(M, (n_j, f_j, h_j)_j).$$

Using that the data dimensions are independent, we find

$$P(M_{\mathrm{SM}}, (n_j, \mathbf{id}, h^{\mathrm{ACC}})_j) = \sum_{i=1}^{m} \sup_{\mu \in \mathbb{R}^d} \mathbb{E}\left[ \|h^{\mathrm{SM}} - \mu\|_2^2 \right] + \sum_{i=1}^{m} \sum_{k=1}^{d} c_{i,k} n_{i,k}.$$
$$= \sum_{i=1}^{m} \sup_{\mu \in \mathbb{R}^d} \mathbb{E}\left[ \sum_{k=1}^{d} \left( \frac{1}{X_{:,k}} \sum_{x \in X_{:,k}} x - \mu_k \right)^2 \right] + \sum_{i=1}^{m} \sum_{k=1}^{d} c_{i,k} n_{i,k}$$
$$= \sum_{i=1}^{m} \sup_{\mu \in \mathbb{R}^d} \sum_{k=1}^{d} \mathbb{E}\left[ \left( \frac{1}{X_{:,k}} \sum_{x \in X_{:,k}} x - \mu_k \right)^2 \right] + \sum_{i=1}^{m} \sum_{k=1}^{d} c_{i,k} n_{i,k}$$
$$= \sum_{i=1}^{m} \sup_{\mu \in \mathbb{R}^d} \sum_{k=1}^{d} \frac{\sigma^2}{X_{:,k}} + \sum_{i=1}^{m} \sum_{k=1}^{d} c_{i,k} n_{i,k}$$

$$= \sum_{k=1}^{d} \frac{m\sigma^2}{X_{:,k}} + \sum_{i=1}^{m} \sum_{k=1}^{d} c_{i,k} n_{i,k}$$

$$= \sum_{k=1}^{d} \left( \frac{m\sigma^2}{\sum_{i=1}^{m} n_{i,k}} + \sum_{i=1}^{m} c_{i,k} n_{i,k} \right).$$

## B. Definition of $G_{i,k}(\alpha_{i,k})$

The corruption coefficients used in Algorithm 2, $\alpha_{i,k}$, are each defined to be the smallest number larger than $\sqrt{n_{i,k}^\star}$ such that $G_{i,k}(\alpha_{i,k}) = 0$ where $G_{i,k}$ is defined as

$$G_{i,k}(\alpha_{i,k}) := \frac{4\alpha_{i,k}}{\sqrt{T_k}} \left( \frac{4\alpha_{i,k}^2 T_k}{D_{i,k} n_{i,k}^\star} - 1 - c_{i,k} \frac{16\alpha_{i,k}^2 T_k n_{i,k}^\star}{\sigma^2 D_{i,k}} \right)$$

$$- \exp\left( \frac{T_k}{8\alpha_{i,k}^2} \right) \left( \frac{4\alpha_{i,k}^2}{T_k} \left( \frac{T_k}{n_{i,k}^\star} + 1 \right) - 1 \right) \sqrt{2\pi} \text{Erfc}\left( \sqrt{\frac{T_k}{8\alpha_{i,k}^2}} \right). \tag{11}$$

and $D_{i,k} := T_k - 2n_{i,k}^\star$.

## C. Examples of different bargaining solutions

This section gives the bargaining solutions and associated computations shown in Fig. 1 (i.e. when $d = 1$, $c_{1,1} = 0.033$, $c_{2,1} = 0.066$, $c_{3,1} = 0.1$, and $\sigma = 10$). Each bargaining solution corresponds to a simple optimization problem. The following computations were obtained by feeding the respective optimization problems into Mathematica with the aforementioned problem parameters.

**Working alone.** Recall from §A.1 that when agents work alone they will collect $\frac{\sigma}{\sqrt{c_{i,1}}}$ points to minimize their penalty. This results in a penalty of $2\sigma\sqrt{c_{i,1}}$. Therefore,

$$n_{1,1}^{\text{IND}} = 55 \qquad n_{2,1}^{\text{IND}} = 39, \qquad n_{3,1}^{\text{IND}} = 32,$$

$$\implies p_1^{\text{IND}} \approx 3.63, \quad p_2^{\text{IND}} \approx 5.13, \quad p_3^{\text{IND}} \approx 6.23.$$

**Social penalty minimization without IR constraints.** Without IR constraints, the agent with the cheapest collection cost for each distribution will be responsible for collecting all of the data for that distribution. Because agents can have the same collection costs, there are multiple agents which could have the cheapest collection cost resulting in different bargaining solutions with the same social penalties. In this case, for the sake of convenience, assume that $i = \arg\min_i c_{i,k}$ chooses one of the agents with the cheapest cost for distribution $k$ in which case the following bargaining solution is one of potentially many which minimizes the social penalty

$$n^{\text{SP}} = \left( n_{i,k}^{\text{SP}} \right)_{i,k} \qquad \text{where} \qquad n_{i,k}^{\text{SP}} = \begin{cases} \arg\min_{n_{i,k}} \left( m\frac{\sigma^2}{n_{i,k}} + c_{i,k} n_{i,k} \right) & \text{if } i = \arg\min_i c_{i,k} \\ 0 & \text{otherwise} \end{cases}$$

$$= \begin{cases} 2\sigma\sqrt{mc_{i,k}} & \text{if } i = \arg\min_i c_{i,k} \\ 0 & \text{otherwise} \end{cases}.$$

In our example this results in

$$n_{1,1}^{\text{SP}} = 95, \qquad n_{2,1}^{\text{SP}} = 0, \qquad n_{3,1}^{\text{SP}} = 0,$$

$$\implies p_1 \left( M_{\text{SM}}, s_{n^{\text{SP}}}^{\text{COM}} \right) \approx 4.19, \qquad p_2 \left( M_{\text{SM}}, s_{n^{\text{SP}}}^{\text{COM}} \right) = p_3 \left( M_{\text{SM}}, s_{n^{\text{SP}}}^{\text{COM}} \right) \approx 1.05.$$

**Social penalty minimization with respect to IR constraints.** Recall that minimizing social penalty subject to IR constraints is given by the optimization problem in (7). Although the cheapest agent for a particular distribution can no longer collect all of the data (as this would not be IR), it turns out that the solution to (7) still has the cheapest agents

collecting most of the data. More specifically, the cheapest agents collect slightly more data than they would on their own and receive a small amount of data from the other agents which offsets this increase in penalty. It can be shown that under the solution to (7) there is a set of cheap agents whose penalty is exactly their penalty when working alone. If it were the case that this penalty was lower than when working alone, more of the work of data collection could be shifted to this agent which would result in a decreased social penalty. Solving (7) in our example we find that

$$n_{1,1}^{\mathrm{OPT}} = 71, \qquad n_{2,1}^{\mathrm{OPT}} = 7, \qquad n_{3,1}^{\mathrm{OPT}} = 0,$$
$$\implies p_1\left(M_{\mathrm{SM}}, s_{n^{\mathrm{OPT}}}^{\mathrm{COM}}\right) \approx 3.63, \qquad p_2\left(M_{\mathrm{SM}}, s_{n^{\mathrm{OPT}}}^{\mathrm{COM}}\right) \approx 1.72, \qquad p_3\left(M_{\mathrm{SM}}, s_{n^{\mathrm{OPT}}}^{\mathrm{COM}}\right) \approx 1.28.$$

**Nash bargaining.** Nash bargaining typically refers to the bargaining solution which maximizes the product of each agents utility minus the utility they can achieve when working alone (Mas-Colell et al., 1995). In our setting we can think of penalty as negative utility. Thus, while there are multiple ways to define the Nash bargaining solution in our problem, it is reasonable to define it as

$$n^{\mathrm{NB}} = \operatorname*{argmax}_{n} \prod_{i=1}^{k} \left(p_i^{\mathrm{IND}} - p_i\left(M_{\mathrm{SM}}, s_n^{\mathrm{COM}}\right)\right).$$

i.e. the bargaining solution which maximizes product of the differences between an agents penalty when working alone and when collaborating with others. In our example we get that

$$n_{1,1}^{\mathrm{NB}} = 35, \qquad n_{2,1}^{\mathrm{NB}} \approx 23, \qquad n_{3,1}^{\mathrm{NB}} \approx 16,$$
$$\implies p_1\left(M_{\mathrm{SM}}, s_{n^{\mathrm{NB}}}^{\mathrm{COM}}\right) \approx 2.50, \qquad p_2\left(M_{\mathrm{SM}}, s_{n^{\mathrm{NB}}}^{\mathrm{COM}}\right) \approx 2.89, \qquad p_3\left(M_{\mathrm{SM}}, s_{n^{\mathrm{NB}}}^{\mathrm{COM}}\right) \approx 2.92.$$

**Egalitarian bargaining.** Egalitarian bargaining is typically defined as the bargaining solution under which all agents achieve the same utility and for which this utility is maximized (Mas-Colell et al., 1995). In our problem penalty can be viewed as negative utility. While there are many ways in which egalitarian bargaining can be defined, it is reasonable to define the egalitarian bargaining solution to be

$$n^{\mathrm{EB}} = \operatorname*{argmin}_{n} \max_{i} p_i\left(M_{\mathrm{SM}}, s_n^{\mathrm{COM}}\right).$$

In our example we get that

$$n_{1,1}^{\mathrm{EB}} = 41 \qquad n_{2,1}^{\mathrm{EB}} = 21 \qquad n_{3,1}^{\mathrm{EB}} = 14$$
$$\implies p_1\left(M_{\mathrm{SM}}, s_{n^{\mathrm{EB}}}^{\mathrm{COM}}\right) = p_2\left(M_{\mathrm{SM}}, s_{n^{\mathrm{EB}}}^{\mathrm{COM}}\right) = p_3\left(M_{\mathrm{SM}}, s_{n^{\mathrm{EB}}}^{\mathrm{COM}}\right) \approx 2.69.$$

# D. Proof of Theorem 1

The proof of the theorem can be divided into two cases based on whether or not $\left(c, n^{\mathrm{B}}\right)$ satisfies the condition given in (8). Proving the theorem when (8) does not hold is given in §D.1. When (8) holds the proof is given in §D.2.

## D.1. Proof when condition (8) is not satisfied

**Incentive Compatibility and Individual Rationality.** If $\left(c, n^{\mathrm{B}}\right)$ does not satisfy (8), the mechanism employs the following simple procedure for each agent and distribution. If an agent does not have the lowest collection cost for a distribution, they receive the sample mean of all of the data for that distribution. Otherwise, the agent only receives the sample mean of the data they submitted for that distribution.

Note that when an agent has the lowest collection cost, it is best for her to collection $n_{i,k}^{\mathrm{IND}}$ points and use the sample mean. Moreover, she is no worse off submitting this data truthfully, and accepting the estimate from the mechanism. Also, when an agent doesn't have the lowest collection cost for a particular distribution, she will receive all the data from agents that do have the cheapest collection cost. Because their collection cost is cheaper, the best strategy for the agent receiving the data is to not collect any data (as the cost of data collection outweighs the marginal decrease in estimation error). Therefore, assuming the other agents follow $s_{-i}^\star$, the best strategy for agent $i$ is $s_i^\star$, i.e. $s^\star$ is a Nash equilibrium under $M_{\mathrm{CBL}}$ when (8) isn't satisfied. Finally NIC implies IR since an agent working on their own is a valid strategy.

**Efficiency.** Recall that the social penalty is minimized by, for each distribution $k \in [d]$, having one of the agent with the cheapest cost to sample that distribution, collect $\frac{\sigma \sqrt{m}}{\min_i \sqrt{c_{i,k}}}$ points from distribution $k$ and freely share them with the other agents after which each agent will take the sample mean of the data for each distribution. Therefore we have that

$$p_i(M_{\text{SM}}, s_n^{\text{COM}}) \leq \sum_{k=1}^{d} \left( \frac{\sigma^2}{\frac{\sigma\sqrt{m}}{\min_j \sqrt{c_{j,k}}}} + \mathbb{1}\left[i \in \operatorname*{argmin}_j c_{j,k}\right] \min_j c_{j,k} \frac{\sigma\sqrt{m}}{\min_j \sqrt{c_{j,k}}} \right)$$

$$= \sum_{k=1}^{d} \left( \sigma \frac{\min_j \sqrt{c_{j,k}}}{\sqrt{m}} + \mathbb{1}\left[i \in \operatorname*{argmin}_j c_{j,k}\right] \sigma\sqrt{m} \min_j \sqrt{c_{j,k}} \right)$$

Here the inequality is used in case there are multiple agents with the same cheapest collection cost in which only one of them collects the data. Using the definition of the mechanism we also have that

$$p_i\left(M_{\text{CBL}}, s^\star\right) = \sum_{k=1}^{d} \left( \frac{\sigma^2}{\left|\operatorname{argmin}_j c_{j,k}\right| \frac{\sigma}{\min_i \sqrt{c_{i,k}}}} + \mathbb{1}\left[i \in \operatorname*{argmin}_j c_{j,k}\right] \min_i c_{i,k} \frac{\sigma}{\min_i \sqrt{c_{i,k}}} \right)$$

Therefore,

$$\frac{p_i\left(M_{\text{CBL}}, s^\star\right)}{p_i(M_{\text{SM}}, s_n^{\text{COM}})} = \frac{\sum_{k=1}^{d} \left( \frac{\sigma^2}{\left|\operatorname{argmin}_j c_{j,k}\right| \frac{\sigma}{\min_i \sqrt{c_{i,k}}}} + \mathbb{1}\left[i \in \operatorname{argmin}_j c_{j,k}\right] \min_i c_{i,k} \frac{\sigma}{\min_i \sqrt{c_{i,k}}} \right)}{\sum_{k=1}^{d} \left( \sigma \frac{\min_j \sqrt{c_{j,k}}}{\sqrt{m}} + \mathbb{1}\left[i \in \operatorname{argmin}_j c_{j,k}\right] \sigma\sqrt{m} \min_j \sqrt{c_{j,k}} \right)}$$

$$\leq \frac{\sum_{k=1}^{d} \frac{\sigma^2}{\left|\operatorname{argmin}_j c_{j,k}\right| \frac{\sigma}{\min_i \sqrt{c_{i,k}}}}}{\sum_{k=1}^{d} \sigma \frac{\min_j \sqrt{c_{j,k}}}{\sqrt{m}}}$$

$$\leq \sqrt{m}.$$

Therefore, $(M_{\text{CBL}}, s^\star) \sqrt{m}$–approximates $n^{\text{B}}$. When $n^{\text{B}} = n^{\text{OPT}}$ it follows from summing the penalties of all the agents that $(M_{\text{CBL}}, s^\star)$ is $\sqrt{m}$–efficient.

### D.2. Proof when condition (8) is satisfied

When (8) is satisfied the proof becomes just an instance of Theorem 5 by taking $n^{\text{B}} = n^{\text{OPT}}$.

## E. Proof of Theorem 5

We will first show in §E.1 that the constants used in Algorithm 2 are well defined. We then prove Nash incentive compatibility, individual rationality, and efficiency in §E.2, §E.3, and §E.4 respectively.

### E.1. Proving $\alpha_{i,k}$ is well defined when $i \notin V_k$ and $n_{i,k}^\star > 0$

We have discussed how it may not be the case that for all $(i, k) \in [m] \times [d]$ a solution to $G_{i,k}(\alpha_{i,k})$ exists. However, we can show that $i \notin V_k$, $n_{i,k}^\star > 0$ is a sufficient condition for the existence of such an $\alpha_{i,k}$. We can also show that $\alpha_{i,k} \geq \sqrt{n_{i,k}^\star}$. The following lemma proves these two claims. Thus $\alpha_{i,k}$ as used in Algorithm 2 is well defined.

**Lemma 1.** *Let $G_{i,k}(\alpha_{i,k})$ be defined as in (11). For any agent $i \notin V_k$ such that $n_{i,k}^\star > 0$, $\exists \alpha_{i,k} \geq \sqrt{n_{i,k}^\star}$ such that $G_{i,k}(\alpha_{i,k}) = 0$.*

*Proof.* We will show that $\lim_{\alpha_{i,k} \to \infty} G_{i,k}(\alpha_{i,k}) = \infty$ and $G_{i,k}\left(\sqrt{n_{i,k}^\star}\right) \leq 0$. Since $G_{i,k}\left(\sqrt{n_{i,k}^\star}\right)$ is continuous, we will conclude that $\exists \alpha_{i,k} \geq \sqrt{n_{i,k}^\star}$ such that $G_{i,k}(\alpha_{i,k}) = 0$.

By the definition of $G_{i,k}(\alpha_{i,k})$ we have

$$\lim_{\alpha_{i,k}\to\infty} G_{i,k}(\alpha_{i,k})$$

$$= \lim_{\alpha_{i,k}\to\infty} \left( \frac{4\alpha_{i,k}}{\sqrt{T_k}} \left( \frac{4\alpha_{i,k}^2 T_k}{D_{i,k} n_{i,k}^\star} - 1 - c_{i,k} \frac{16\alpha_{i,k}^2 T_k n_{i,k}^\star}{\sigma^2 D_{i,k}} \right) \right.$$

$$\left. - \exp\left( \frac{T_k}{8\alpha_{i,k}^2} \right) \left( \frac{4\alpha_{i,k}^2}{T_k} \left( \frac{T_k}{n_{i,k}^\star} + 1 \right) - 1 \right) \sqrt{2\pi}\,\mathrm{Erfc}\left( \sqrt{\frac{T_k}{8\alpha_{i,k}^2}} \right) \right)$$

$$\geq \lim_{\alpha_{i,k}\to\infty} \left( \frac{4\alpha_{i,k}}{\sqrt{T_k}} \left( \frac{4\alpha_{i,k}^2 T_k}{D_{i,k} n_{i,k}^\star} - 1 - c_{i,k} \frac{16\alpha_{i,k}^2 T_k n_{i,k}^\star}{\sigma^2 D_{i,k}} \right) - 100 \left( \frac{4\alpha_{i,k}^2}{T_k} \left( \frac{T_k}{n_{i,k}^\star} + 1 \right) - 1 \right) \right)$$

$$= \lim_{\alpha_{i,k}\to\infty} \left( \frac{4\alpha_{i,k}}{\sqrt{T_k}} \left( \frac{4\alpha_{i,k}^2 T_k \sigma^2 - D_{i,k} n_{i,k}^\star \sigma^2 - 16 c_{i,k} \alpha_{i,k}^2 T_k (n_{i,k}^\star)^2}{D_{i,k} n_{i,k}^\star \sigma^2} \right) - 100 \left( \frac{4\alpha_{i,k}^2}{T_k} \left( \frac{T_k}{n_{i,k}^\star} + 1 \right) - 1 \right) \right)$$

$$= \lim_{\alpha_{i,k}\to\infty} \left( \frac{4\alpha_{i,k}}{D_{i,k} n_{i,k}^\star \sigma^2 \sqrt{T_k}} \left( 4T_k \alpha_{i,k}^2 (\sigma^2 - 4c_{i,k}(n_{i,k}^\star)^2) - D_{i,k} n_{i,k}^\star \sigma^2 \right) - 100 \left( \frac{4\alpha_{i,k}^2}{T_k} \left( \frac{T_k}{n_{i,k}^\star} + 1 \right) - 1 \right) \right).$$

Notice that $\sigma^2 - 4c_{i,k}(n_{i,k}^\star)^2 > 0 \iff \frac{\sigma}{2\sqrt{c_{i,k}}} > n_{i,k}^\star \iff \frac{n_{i,k}^{\mathrm{IND}}}{2} > n_{i,k}^\star$ which is implied by Lemma 10 since $i \notin V_k$. Therefore, since the first term is positive for sufficiently large $\alpha_{i,k}$ and has $\alpha_{i,k}^3$ while the second term has $\alpha_{i,k}^2$, $\lim_{\alpha_{i,k}\to\infty} G_{i,k}(\alpha_{i,k}) = \infty$.

Recall the definition of $G_{i,k}(\alpha_{i,k})$. Notice that $\alpha_{i,k} \geq \sqrt{n_{i,k}^\star} \Rightarrow -1 + 4\left( \frac{1}{T_k} + \frac{1}{n_{i,k}^\star} \right) \alpha_{i,k}^2 \geq 0$. Lemma 8 gives us the following lower bound on the Gaussian complementary error function

$$\mathrm{Erfc}_{\mathrm{LB}}(x) := \frac{1}{\sqrt{\pi}} \left( \frac{\exp(-x^2)}{x} - \frac{\exp(-x^2)}{2x^3} \right).$$

Therefore, for $\alpha_{i,k} \geq \sqrt{n_{i,k}^\star}$, we have the following upper bound on $G_{i,k}(\alpha_{i,k})$

$$G_{i,k}(\alpha_{i,k})_{\mathrm{UB}} := \frac{4\alpha_{i,k} \left( -1 + \frac{4T_k \alpha_{i,k}^2}{n_{i,k}^\star (T_k - 2n_{i,k}^\star)} - \frac{16 c_{i,k} n_{i,k}^\star T_k \alpha_{i,k}^2}{(T_k - 2n_{i,k}^\star)\sigma^2} \right)}{\sqrt{T_k}}$$

$$- e^{\frac{T_k}{8\alpha_{i,k}^2}} \sqrt{2\pi} \left( -1 + \frac{4\left( 1 + \frac{T_k}{n_{i,k}^\star} \right) \alpha_{i,k}^2}{T_k} \right) \mathrm{Erfc}_{\mathrm{LB}}\left( \sqrt{\frac{T_k}{8\alpha_{i,k}^2}} \right). \tag{12}$$

Using Lemma 14, we can write $G_{i,k}\left( \sqrt{n_{i,k}^\star} \right)_{\mathrm{UB}}$ as

$$G_{i,k}\left( \sqrt{n_{i,k}^\star} \right)_{\mathrm{UB}} = -\frac{64(n_{i,k}^\star)^{3/2} (c_{i,k} n_{i,k}^\star (T_k)^3 + (2(n_{i,k}^\star)^2 - (T_k)^2)\sigma^2)}{(T_k)^{5/2}(T_k - 2n_{i,k}^\star)\sigma^2}.$$

Now observe that

$$G_{i,k}\left( \sqrt{n_{i,k}^\star} \right)_{\mathrm{UB}} \leq 0 \iff -\frac{64(n_{i,k}^\star)^{3/2} (c_{i,k} n_{i,k}^\star (T_k)^3 + (2(n_{i,k}^\star)^2 - (T_k)^2)\sigma^2)}{(T_k)^{5/2}(T_k - 2n_{i,k}^\star)\sigma^2} \leq 0$$

$$\iff c_{i,k} n_{i,k}^\star (T_k)^3 + (2(n_{i,k}^\star)^2 - (T_k)^2)\sigma^2 \geq 0.$$

But, since $i \notin V_k$, we have $n_{i,k}^\star T_k \geq (n_{i,k}^{\mathrm{IND}})^2$ as a result of the second while loop in Algorithm 1. Thus,

$$c_{i,k} n_{i,k}^\star (T_k)^3 + (2(n_{i,k}^\star)^2 - (T_k)^2)\sigma^2 \geq c_{i,k}(n_{i,k}^{\mathrm{IND}})^2(T_k)^2 + (2(n_{i,k}^\star)^2 - (T_k)^2)\sigma^2$$

$$= \sigma^2(T_k)^2 + 2(n_{i,k}^\star)^2\sigma^2 - \sigma^2(T_k)^2$$

$$= 2(n_{i,k}^\star)^2\sigma^2 \geq 0.$$

Therefore, $G_{i,k}(\sqrt{n_{i,k}^\star})_{\mathrm{UB}} \leq 0$ which gives us that $G_{i,k}(\sqrt{n_{i,k}^\star}) \leq 0$. Since $G_{i,k}(\alpha_{i,k})$ is continuous, together with $G_{i,k}(\sqrt{n_{i,k}^\star}) \leq 0$ and $\lim_{\alpha_{i,k}\to\infty} G_{i,k}(\alpha_{i,k}) = \infty$, we conclude that $\exists \alpha_{i,k} \geq \sqrt{n_{i,k}^\star}$ such that $G_{i,k}(\alpha_{i,k}) = 0$. □

**E.2. $M_{\text{CBL}}$ is Nash incentive compatible**

We will follow the proof structure of Nash incentive compatibility in Chen et al. (2023). Like Chen et al. (2023), we will decompose the proof into two lemmas. The first lemma shows that for any distribution over $n_i$, the optimal strategy is to use $f_i = f_i^\star$ and $h_i = h_i^\star$. The second lemma shows that the optimal strategy is to set $n_i = n_i^\star$. Together, these lemmas imply that $s_i = (n_i^\star, f_i^\star, h_i^\star)$ is the optimal strategy. Although the proof structure will be the same, we will need to make changes to handle our more general problem setting.

**Lemma 2.** *For all $i \in [m]$ and $s_i \in \Delta(\mathcal{S})$ we have that*

$$p_i\left(M_{\text{CBL}}, (s_i^\star, s_{-i}^\star)\right) \leq p_i\left(M_{\text{CBL}}, (s_i, s_{-i}^\star)\right)$$

*where $s^\star$ is the pure strategy defined in* (9).

We can decompose the proof into the following two lemmas:

**Lemma 3.** *For all $i \in [m]$ and $s_i \in \Delta(\mathcal{S})$, define the randomized strategy $\widetilde{s}_i \in \Delta(\mathcal{S})$ by sampling $(n_i, f_i, h_i) \sim s_i$ and selecting $(n_i, f_i^\star, h_i^\star)$. Then, we have that*

$$p_i\left(M_{\text{CBL}}, (\widetilde{s}_i, s_{-i}^\star)\right) \leq p_i\left(M_{\text{CBL}}, (s_i, s_{-i}^\star)\right).$$

*Proof.* See H.1. $\qquad\square$

**Lemma 4.** *For all $i \in [m]$ and $s_i \in \Delta(\mathcal{S})$ such that $f_i = f_i^\star, h_i = h_i^\star$*

$$p_i\left(M_{\text{CBL}}, (s_i^\star, s_{-i}^\star)\right) \leq p_i\left(M_{\text{CBL}}, (s_i, s_{-i}^\star)\right).$$

*Proof.* See H.2. $\qquad\square$

*Proof of Lemma 2.* For any strategy $s_i \in \Delta(\mathcal{S})$ we can construct the strategy $s_i'$ which samples $(n_i, f_i, h_i) \sim s_i$ and selects $(n_i, f_i^\star, h_i^\star)$. Applying Lemmas 3 and 4 we get

$$p_i\left(M_{\text{CBL}}, (s_i^\star, s_{-i}^\star)\right) \leq p_i\left(M_{\text{CBL}}, (s_i', s_{-i}^\star)\right) \leq p_i\left(M_{\text{CBL}}, (s_i, s_{-i}^\star)\right).$$

$\qquad\square$

**E.3. $M_{\text{CBL}}$ is individually rational**

If an agent works on their own they, by definition, achieve their individually rational penalty. Furthermore, an agent may choose to work on their own as this is an option contained within their strategy space. Therefore, because we have shown that $M_{\text{CBL}}$ is incentive compatible, any agent has no incentive to deviate from their recommended strategy, hence they must achieve a penalty no worse than their individually rational penalty. Therefore, $M_{\text{CBL}}$ is individually rational.

**E.4. $(M_{\text{CBL}}, s^\star)$ is an 8–approximation to $n^{\text{B}}$**

Let $p_{i,k}$ denote the penalty agent $i$ incurs as a result of trying to estimation $\mu_k$, i.e. the sum of her estimation error for distribution $k$ plus the cost she incurs collecting $n_{i,k}$ points from distribution $k$. We will show in Lemma 5 that for all $(i,k) \in [m] \times [d]$

$$\frac{p_{i,k}\left(M_{\text{CBL}}, s^\star\right)}{p_{i,k}\left(M_{\text{SM}}, \{(n^{\text{B}}, \mathbf{id}, h^{\text{ACC}})\}_j\right)} \leq 8$$

Using this, we will get

$$\frac{p_i\left(M_{\text{CBL}}, s^\star\right)}{p_i\left(M_{\text{SM}}, \{(n^{\text{B}}, \mathbf{id}, h^{\text{ACC}})\}_j\right)} = \frac{\sum_{k=1}^d p_{i,k}\left(M_{\text{CBL}}, s^\star\right)}{\sum_{k=1}^d p_{i,k}\left(M_{\text{SM}}, \{(n^{\text{B}}, \mathbf{id}, h^{\text{ACC}})\}_j\right)}$$

$$\leq \frac{\sum_{k=1}^d 8p_{i,k}\left(M_{\text{SM}}, \{(n^{\text{B}}, \mathbf{id}, h^{\text{ACC}})\}_j\right)}{\sum_{k=1}^d p_{i,k}\left(M_{\text{SM}}, \{(n^{\text{B}}, \mathbf{id}, h^{\text{ACC}})\}_j\right)}$$

$$= 8$$

This also tells us that we can bound the price of stability on the social penalty by writing

$$\frac{\sum_{i=1}^{m} p_i\left(M_{\text{CBL}}, s^\star\right)}{\sum_{i=1}^{m} p_i\left(M_{\text{SM}}, \left\{(n^{\text{B}}, \mathbf{id}, h^{\text{ACC}})\right\}_j\right)} \leq \frac{\sum_{i=1}^{m} 8p_i\left(M_{\text{SM}}, \left\{(n^{\text{B}}, \mathbf{id}, h^{\text{ACC}})\right\}_j\right)}{\sum_{i=1}^{m} p_i\left(M_{\text{SM}}, \left\{(n^{\text{B}}, \mathbf{id}, h^{\text{ACC}})\right\}_j\right)} = 8$$

**Lemma 5.** *For all* $(i, k) \in [m] \times [d]$ *we have*

$$\frac{p_{i,k}\left(M_{\text{CBL}}, s^\star\right)}{p_{i,k}\left(M_{\text{SM}}, \left\{(n^{\text{B}}, \mathbf{id}, h^{\text{ACC}})\right\}_j\right)} \leq 8$$

*Proof.* We break the proof into two cases depending on whether or nor $i \in V_k$. When $i \in V_k$ we further subdivide into the following three sub cases: (a) $n_{i,k}^{\text{B}} = 0$, (b) $n_{i,k}^\star = n_{i,k}^{\text{B}} > 0$, and (c) $n_{i,k}^\star \neq n_{i,k}^{\text{B}} > 0$.

**Case 1.** $(i \in V_k)$  In this case we know that $n_{i,k}^\star = n_{i,k}^{\text{IND}}$ as a result of the first while loop in Algorithm 1. Furthermore, we know from Algorithm 2 that $I_{i,k} = \frac{1}{Y_{i,k}} \sum_{y \in Y_{i,k}} y$. Therefore, because agent $i$ reports truthfully under $s^\star$, we have that

$$p_{i,k}\left(M_{\text{CBL}}, s^\star\right) = \frac{\sigma^2}{n_{i,k}^\star} + c_{i,k} n_{i,k}^\star = \frac{\sigma^2}{n_{i,k}^{\text{IND}}} + c_{i,k} n_{i,k}^{\text{IND}}$$

Since $i \in V_k$, we also know that

$$\frac{\sigma^2}{n_{i,k}^{\text{IND}}} + c_{i,k} n_{i,k}^{\text{IND}} \leq 4\left(\frac{\sigma^2}{\sum_{j=1}^{m} n_{i,k}'} + c_{i,k} n_{i,k}'\right)$$

where $n'$ is the value of $\widetilde{n}$ on the iteration when $i$ was added to $V_k$. Furthermore, we know that $\frac{n_{i,k}^{\text{IND}}}{n_{i,k}^{\text{B}}} > \frac{1}{2}$ as a result of Lemma 11. But this implies that $\sum_{j=1}^{m} n' > \frac{1}{2} \sum_{i=1}^{m} n_{i,k}^{\text{B}}$. Therefore,

$$\frac{\sigma^2}{\sum_{j=1}^{m} n_{i,k}'} + c_{i,k} n_{i,k}' < \frac{2\sigma^2}{\sum_{j=1}^{m} n_{j,k}^{\text{B}}} + c_{i,k} n_{i,k}'$$

$$= \frac{2\sigma^2}{\sum_{j=1}^{m} n_{j,k}^{\text{B}}} + c_{i,k} n_{i,k}^{\text{B}}$$

$$< 2\left(\frac{\sigma^2}{\sum_{j=1}^{m} n_{j,k}^{\text{B}}} + c_{i,k} n_{i,k}^{\text{B}}\right)$$

Combining these two inequalities, we get

$$\frac{p_{i,k}\left(M_{\text{CBL}}, s^\star\right)}{p_{i,k}\left(M_{\text{SM}}, \left\{(n^{\text{B}}, \mathbf{id}, h^{\text{ACC}})\right\}_j\right)} = \frac{\frac{\sigma^2}{n_{i,k}^{\text{IND}}} + c_{i,k} n_{i,k}^{\text{IND}}}{\frac{\sigma^2}{\sum_{j=1}^{m} n_{j,k}^{\text{B}}} + c_{i,k} n_{i,k}^{\text{B}}} \leq \frac{8\left(\frac{\sigma^2}{\sum_{j=1}^{m} n_{j,k}^{\text{B}}} + c_{i,k} n_{i,k}^{\text{B}}\right)}{\frac{\sigma^2}{\sum_{j=1}^{m} n_{j,k}^{\text{B}}} + c_{i,k} n_{i,k}^{\text{B}}} = 8$$

Therefore, in this case we can conclude that

$$\frac{p_{i,k}\left(M_{\text{CBL}}, s^\star\right)}{p_{i,k}\left(M_{\text{SM}}, \left\{(n^{\text{B}}, \mathbf{id}, h^{\text{ACC}})\right\}_j\right)} \leq 8$$

**Case 2. (a)** $(i \notin V_k, \ n_{i,k}^{\text{B}} = 0)$  In this case $i$ simply receives the sample mean of all of the other agents' uncorrupted data for distribution $k$. In this case we can apply Lemma 13 to get

$$\frac{p_{i,k}\left(M_{\text{CBL}}, s^\star\right)}{p_{i,k}\left(M_{\text{SM}}, \left\{(n^{\text{B}}, \mathbf{id}, h^{\text{ACC}})\right\}_j\right)} = \frac{\frac{\sigma^2}{\sum_{j \neq i} n_{j,k}^\star}}{\frac{\sigma^2}{\sum_{j \neq i} n_{i,k}^{\text{B}}}} = \frac{\sum_{j \neq i} n_{i,k}^{\text{B}}}{\sum_{j \neq i} n_{j,k}^\star} \leq 2$$

Therefore, in this case we can conclude that

$$\frac{p_{i,k}\left(M_{\text{CBL}}, s^\star\right)}{p_{i,k}\left(M_{\text{SM}}, \left\{(n^{\text{B}}, \mathbf{id}, h^{\text{ACC}})\right\}_j\right)} \leq 2$$

**Case 2. (b)** ($i \notin V_k$, $n_{i,k}^\star = n_{i,k}^{\mathrm{B}} > 0$)   In this case $i$ instead receives a weighted average of clean and uncorrupted data for distribution $k$ from the other agents. Using Lemma 12 and that $n_{i,k}^\star = n^{\mathrm{B}}$, we have

$$
\frac{p_{i,k}\left(M_{\mathrm{CBL}}, s^\star\right)}{p_{i,k}\left(M_{\mathrm{SM}}, \{(n^{\mathrm{B}}, \mathbf{id}, h^{\mathrm{ACC}})\}_j\right)} = \frac{p_{i,k}\left(M_{\mathrm{CBL}}, s^\star\right)}{\frac{\sigma^2}{\sum_{i=1}^n n_{i,k}^{\mathrm{B}}} + c_{i,k} n_{i,k}^{\mathrm{B}}}
$$

$$
= \frac{p_{i,k}\left(M_{\mathrm{CBL}}, s^\star\right)}{\frac{\sigma^2}{\sum_{i=1}^n n_{i,k}^{\mathrm{B}}} + c_{i,k} n_{i,k}^\star}
$$

$$
\leq \frac{p_{i,k}\left(M_{\mathrm{CBL}}, s^\star\right)}{\frac{\sigma^2}{2\sum_{i=1}^n n_{i,k}'} + c_{i,k} n_{i,k}^\star}
$$

Let $T_k = \sum_{i=1}^m n_{i,k}'$, i.e. how it is specified in Algorithm 1. Lemma 17 gives us an explicit expression for $p_{i,k}\left(M_{\mathrm{CBL}}, s^\star\right)$. Substituting this in, we get

$$
\frac{p_{i,k}\left(M_{\mathrm{CBL}}, s^\star\right)}{\frac{\sigma^2}{2\sum_{i=1}^n n_{i,k}'} + c_{i,k} n_{i,k}^\star}
$$

$$
= \frac{T_k}{n_{i,k}^\star (2c_{i,k} n_{i,k}^\star T_k + \sigma^2)} \left( 2c_{i,k}(n_{i,k}^\star)^2 + \sigma^2 + \frac{\alpha_{i,k}\left(16c_{i,k}(n_{i,k}^\star)^2 T_k \alpha_{i,k}^2 + (n_{i,k}^\star (T_k - 2n_{i,k}^\star) - 4T_k \alpha_{i,k}^2)\sigma^2\right)}{-n_{i,k}^\star T_k \alpha_{i,k} + 4(n_{i,k}^\star + T_k)\alpha_{i,k}^3} \right)
$$

$$
= \frac{T_k(2c_{i,k}(n_{i,k}^\star)^2 + 6c_{i,k} n_{i,k}^\star T_k + \sigma^2)}{(n_{i,k}^\star + T_k)(2c_{i,k} n_{i,k}^\star T_k + \sigma^2)} + \frac{4c_{i,k}(n_{i,k}^\star)^2(T_k)^3 - 2(n_{i,k}^\star)^2 T_k \sigma^2 - n_{i,k}^\star (T_k)^2 \sigma^2}{(n_{i,k}^\star + T_k)(-n_{i,k}^\star T_k + 4n_{i,k}^\star \alpha_{i,k}^2 + 4T_k \alpha_{i,k}^2)(2c_{i,k} n_{i,k}^\star T_k + \sigma^2)} \tag{13}
$$

Here the second equality is a direct result of Lemma 18. Because we know $\alpha_{i,k} \geq \sqrt{n_{i,k}^\star}$, the denominator of the second term is positive. Additionally the numerator of the second term is also positive. To see this observe that

$$
4c_{i,k}(n_{i,k}^\star)^2(T_k)^3 - 2(n_{i,k}^\star)^2 T_k \sigma^2 - n_{i,k}^\star (T_k)^2 \sigma^2
$$

$$
= 3c_{i,k}(n_{i,k}^\star)^2(T_k)^3 - 2(n_{i,k}^\star)^2 T_k \sigma^2 + c_{i,k}(n_{i,k}^\star)^2(T_k)^3 - n_{i,k}^\star (T_k)^2 \sigma^2
$$

$$
\geq 3c_{i,k}(n_{i,k}^\star)^2(T_k)^3 - 2(n_{i,k}^\star)^2 T_k \sigma^2 + c_{i,k} n_{i,k}^\star T_k^2 \frac{\sigma^2}{c_{i,k}} - n_{i,k}^\star (T_k)^2 \sigma^2 \tag{14}
$$

$$
= 3c_{i,k}(n_{i,k}^\star)^2(T_k)^3 - 2(n_{i,k}^\star)^2 T_k \sigma^2
$$

$$
\geq 3c_{i,k}(n_{i,k}^\star)^2 T_k \left( 4\frac{\sigma^2}{c_{i,k}} \right) - 2(n_{i,k}^\star)^2 T_k \sigma^2 \tag{15}
$$

$$
= 10(n_{i,k}^\star)^2 T_k \sigma^2
$$

$$
> 0
$$

Line (14) uses that $n_{i,k}^\star T_k \geq \left(n_{i,k}^{\mathrm{IND}}\right)^2 = \frac{\sigma^2}{c_{i,k}}$ and line (15) uses that $T_k > 2n_{i,k}^{\mathrm{IND}} = \frac{\sigma}{\sqrt{c_{i,k}}}$ which is a result of Lemma 9. Both of these statements make use of the fact that $i \notin V_k$. This means that the penalty ratio bound is decreasing as $\alpha_{i,k}$ increases. Therefore, taking $\alpha_{i,k} = \sqrt{n_{i,k}^\star}$ gives an upper bound on this entire expression. Therefore, by Lemma 20, substituting in $\alpha_{i,k} = \sqrt{n_{i,k}^\star}$ into (13) yields:

$$
\frac{T_k(2c_{i,k}(n_{i,k}^\star)^2 + 6c_{i,k} n_{i,k}^\star T_k + \sigma^2)}{(n_{i,k}^\star + T_k)(2c_{i,k} n_{i,k}^\star T_k + \sigma^2)} + \frac{4c_{i,k}(n_{i,k}^\star)^2(T_k)^3 - 2(n_{i,k}^\star)^2 T_k \sigma^2 - n_{i,k}^\star (T_k)^2 \sigma^2}{(n_{i,k}^\star + T_k)(-n_{i,k}^\star T_k + 4n_{i,k}^\star \alpha_{i,k}^2 + 4T_k \alpha_{i,k}^2)(2c_{i,k} n_{i,k}^\star T_k + \sigma^2)}
$$

$$
= \frac{2T_k \left( c_{i,k} n_{i,k}^\star \left( 4n_{i,k}^\star + 11T_k \right) + \sigma^2 \right)}{\left( 4n_{i,k}^\star + 3T_k \right) \left( 2c_{i,k} n_{i,k}^\star T_k + \sigma^2 \right)} \tag{16}
$$

Since $i \notin V_k$, Lemmas 9 and 10 tell us that $T_k > 2n_{i,k}^{\mathrm{IND}}$ and $\frac{n_{i,k}^{\mathrm{IND}}}{2} > n_{i,k}^\star$. Therefore, $T_k > 4n_{i,k}^\star$. We can use this fact to

bound (16) as follows

$$\frac{2T_k\left(c_{i,k}n_{i,k}^\star\left(4n_{i,k}^\star+11T_k\right)+\sigma^2\right)}{\left(4n_{i,k}^\star+3T_k\right)\left(2c_{i,k}n_{i,k}^\star T_k+\sigma^2\right)}$$

$$=\frac{8c_{i,k}(n_{i,k}^\star)^2T_k}{\left(4n_{i,k}^\star+3T_k\right)\left(2c_{i,k}n_{i,k}^\star T_k+\sigma^2\right)}+\frac{2T_k\left(11c_{i,k}n_{i,k}^\star T_k+\sigma^2\right)}{\left(4n_{i,k}^\star+3T_k\right)\left(2c_{i,k}n_{i,k}^\star T_k+\sigma^2\right)}$$

$$=\frac{8c_{i,k}(n_{i,k}^\star)^2T_k}{8c_{i,k}(n_{i,k}^\star)^2T_k+6c_{i,k}n_{i,k}^\star T_k^2+4n_{i,k}^\star\sigma^2+3T_k\sigma^2}+\frac{3T_k\left(\frac{2}{3}11c_{i,k}n_{i,k}^\star T_k+\frac{2}{3}\sigma^2\right)}{\left(4n_{i,k}^\star+3T_k\right)\left(2c_{i,k}n_{i,k}^\star T_k+\sigma^2\right)}$$

$$\leq\frac{8c_{i,k}(n_{i,k}^\star)^2T_k}{8c_{i,k}(n_{i,k}^\star)^2T_k+24c_{i,k}(n_{i,k}^\star)^2T_k+4n_{i,k}^\star\sigma^2+3T_k\sigma^2}+\frac{\left(4n_{i,k}^\star+3T_k\right)\left(\frac{2}{3}11c_{i,k}n_{i,k}^\star T_k+\frac{2}{3}\sigma^2\right)}{\left(4n_{i,k}^\star+3T_k\right)\left(2c_{i,k}n_{i,k}^\star T_k+\sigma^2\right)}$$

$$\leq\frac{8c_{i,k}(n_{i,k}^\star)^2T_k}{32c_{i,k}(n_{i,k}^\star)^2T_k}+\frac{\frac{11}{3}\left(2c_{i,k}n_{i,k}^\star T_k+\sigma^2\right)}{\left(2c_{i,k}n_{i,k}^\star T_k+\sigma^2\right)}$$

$$\leq 4$$

Therefore, in this case we can conclude that

$$\frac{p_{i,k}\left(M_{\text{CBL}},s^\star\right)}{p_{i,k}\left(M_{\text{SM}},\{(n^{\text{B}},\mathbf{id},h^{\text{ACC}})\}_j\right)}\leq 4$$

**Case 2. (c)** ($i\notin V_k$, $n_{i,k}^\star\neq n_{i,k}^{\text{B}}>0$)   Similar to the previous case, $i$ receives a weighted average of clean and uncorrected data for distribution $k$ from the other agents. However, the difference is that in Algorithm 1, $\widetilde{n}_{i,k}$ was changed to $\frac{\left(n_{i,k}^{\text{IND}}\right)^2}{T_k}$ by the second while loop. Thus, $n_{i,k}^\star=\frac{\left(n_{i,k}^{\text{IND}}\right)^2}{T_k}$. We again know from Lemma 12 that $\sum_{i=1}^n n_{i,k}^{\text{B}}\leq 2\sum_{i=1}^m n_{i,k}'$. Therefore, we have that

$$\frac{p_{i,k}\left(M_{\text{CBL}},s^\star\right)}{p_{i,k}\left(M_{\text{SM}},\{(n^{\text{B}},\mathbf{id},h^{\text{ACC}})\}_j\right)}=\frac{p_{i,k}\left(M_{\text{CBL}},s^\star\right)}{\frac{\sigma^2}{\sum_{i=1}^n n_{i,k}^{\text{B}}}+c_{i,k}n_{i,k}^{\text{B}}}$$

$$\leq\frac{p_{i,k}\left(M_{\text{CBL}},s^\star\right)}{\frac{\sigma^2}{\sum_{i=1}^n n_{i,k}^{\text{B}}}}$$

$$\leq\frac{p_{i,k}\left(M_{\text{CBL}},s^\star\right)}{\frac{\sigma^2}{2\sum_{i=1}^n n_{i,k}'}}$$

The only difference so far from the previous case is that we dropped the $c_{i,k}n_{i,k}^{\text{B}}$ term in the denominator. It turns out that we can get away with doing this, knowing that $n_{i,k}^\star=\frac{\left(n_{i,k}^{\text{IND}}\right)^2}{T_k}$. Let $T_k=\sum_{i=1}^m n_{i,k}'$. Lemma 17 gives us an explicit expression for $p_{i,k}\left(M_{\text{CBL}},s^\star\right)$. Substituting this in, we get

$$\frac{p_{i,k}\left(M_{\text{CBL}},s^\star\right)}{\frac{\sigma^2}{2\sum_{i=1}^n n_{i,k}'}}$$

$$=\frac{T_k}{n_{i,k}^\star\sigma^2}\left(2c_{i,k}(n_{i,k}^\star)^2+\sigma^2+\frac{\alpha_{i,k}\left(16c_{i,k}(n_{i,k}^\star)^2T_k\alpha_{i,k}^2+(n_{i,k}^\star(T_k-2n_{i,k}^\star)-4T_k\alpha_{i,k}^2)\sigma^2\right)}{-n_{i,k}^\star T_k\alpha_{i,k}+4(n_{i,k}^\star+T_k)\alpha_{i,k}^3}\right)$$

$$=\frac{T_k(2c_{i,k}(n_{i,k}^\star)^2+6c_{i,k}n_{i,k}^\star T_k+\sigma^2)}{(n_{i,k}^\star+T_k)\sigma^2}+\frac{4c_{i,k}(n_{i,k}^\star)^2(T_k)^3-2(n_{i,k}^\star)^2T_k\sigma^2-n_{i,k}^\star(T_k)^2\sigma^2}{(n_{i,k}^\star+T_k)(-n_{i,k}^\star T_k+4n_{i,k}^\star\alpha_{i,k}^2+4T_k\alpha_{i,k}^2)\sigma^2}\tag{17}$$

Here, the second equality is a direct result of Lemma 19. For $\alpha_{i,k} \geq \sqrt{n_{i,k}^\star}$, the denominator of the second term is positive. Also notice that the numerator of the second term is the same as the one we showed was positive in the previous case. Therefore, we can again conclude that the penalty ratio is decreasing as $\alpha_{i,k}$ increases and so taking $\alpha_{i,k} = \sqrt{n_{i,k}^\star}$ gives an upper bound on the entire expression. Therefore, by Lemma 21, substituting in $\alpha_{i,k} = \sqrt{n_{i,k}^\star}$ into (17) yields:

$$\frac{T_k(2c_{i,k}(n_{i,k}^\star)^2 + 6c_{i,k}n_{i,k}^\star T_k + \sigma^2)}{(n_{i,k}^\star + T_k)\sigma^2} + \frac{4c_{i,k}(n_{i,k}^\star)^2(T_k)^3 - 2(n_{i,k}^\star)^2 T_k \sigma^2 - n_{i,k}^\star(T_k)^2\sigma^2}{(n_{i,k}^\star + T_k)(-n_{i,k}^\star T_k + 4n_{i,k}^\star \alpha_{i,k}^2 + 4T_k\alpha_{i,k}^2)\sigma^2}$$
$$= \frac{2T_k\left(c_{i,k}n_{i,k}^\star\left(4n_{i,k}^\star + 11T_k\right) + \sigma^2\right)}{\left(4n_{i,k}^\star + 3T_k\right)\sigma^2} \tag{18}$$

Using the fact that $n_{i,k}^\star = \frac{\left(n_{i,k}^{\mathrm{IND}}\right)^2}{T_k} = \frac{\sigma^2}{c_{i,k}T_k}$, (18) becomes

$$\frac{2T_k\left(c_{i,k}n_{i,k}^\star\left(4n_{i,k}^\star + 11T_k\right) + \sigma^2\right)}{\left(4n_{i,k}^\star + 3T_k\right)\sigma^2} = \frac{2T_k\left(\sigma^2 + \frac{\sigma^2\left(11T_k + \frac{4\sigma^2}{c_{i,k}T_k}\right)}{T_k}\right)}{\sigma^2\left(3T_k + \frac{4\sigma^2}{c_{i,k}T_k}\right)}$$
$$= \frac{2T_k}{3T_k + \frac{4\sigma^2}{c_{i,k}T_k}} + 2\left(\frac{11T_k + \frac{4\sigma^2}{c_{i,k}T_k}}{3T_k + \frac{4\sigma^2}{c_{i,k}T_k}}\right)$$
$$\leq \frac{2}{3} + 2\frac{11T_k}{3T_k}$$
$$= 8$$

Therefore, in this case we can conclude that

$$\frac{p_{i,k}\left(M_{\mathrm{CBL}}, s^\star\right)}{p_{i,k}\left(M_{\mathrm{SM}}, \{(n^{\mathrm{B}}, \mathbf{id}, h^{\mathrm{ACC}})\}_j\right)} \leq 8$$

In all of the cases we have considered the penalty ratio is bounded by 8. Thus concludes the proof. $\qquad\square$

# F. Hardness results

## F.1. Proof of Theorem 2

*Proof.* We claim that each agent collects their IR amount of data for each distribution $\left\{n_{i,k}^{\mathrm{IND}}\right\}_k$. Suppose that this were not the case, and consider $s'_{-i} \in \Delta(\mathcal{S})^{m-1}$ such that $\sum_{j\neq i,k} n'_{j,k} = 0$ (i.e. when the other agents collect no data). In this case agent $i$ is working alone as they are the only one collecting data and we know from Fact 1 that when working alone the best penalty an agent can achieve is $p_i^{\mathrm{IND}}$ which implies that $p_i\left(M, \left(s_i, s'_{-i}\right)\right) < p_i^{\mathrm{IND}}$ unless $n_{i,k} = n_{i,k}^{\mathrm{IND}}$. Now since we know that $n_{i,k} = n_{i,k}^{\mathrm{IND}}$ we get $p_i\left(M, \left(s_i, s'_{-i}\right)\right) \geq \sum_k c_{i,k}n_{i,k}^{\mathrm{IND}} = \frac{p_i^{\mathrm{IND}}}{2}$. $\qquad\square$

## F.2. Proof of Theorem 3

*Proof.* We claim that each agent collects their IR amount of data for each distribution $\left\{n_{i,k}^{\mathrm{IND}}\right\}_k$. Suppose that this were not the case, and consider $s'_{-i} \in \Delta(\mathcal{S})^{m-1}$ such that $\sum_{j\neq i,k} n'_{j,k} = 0$ (i.e. when the other agents collect no data). In this situation agent $i$ could improve their penalty by collecting their IR amount of data and using the sample mean (see §A.1). But this would contradict that $s$ is a dominant strategy. Thus, $n_{i,k} = n_{i,k}^{\mathrm{IND}}$ and so $p_i\left(M, \left(s_i, s'_{-i}\right)\right) \geq \sum_k c_{i,k}n_{i,k}^{\mathrm{IND}} = \frac{p_i^{\mathrm{IND}}}{2}$. $\qquad\square$

## F.3. Proof of Theorem 4

We will now prove the hardness result given in Theorem 4.

**Construction.** We will begin by constructing a difficult problem instance. Suppose there are $m$ agents and $d = 2$ distributions. Let $\gamma \in (0, \infty)$, and let the costs $c \in (0, \infty]^{m \times 2}$ be:

$$c_{1,1} = \gamma, \quad c_{1,2} = \infty, \qquad c_{i,1} = \infty, \quad c_{i,2} = \gamma \quad \forall i \in \{2, \ldots, m\}. \tag{19}$$

Next, let us compute the minimizer of the social penalty in $\Theta$. Recall that for a given $n \in \mathbb{N}^{m \times d}$, we have that $(M_{\text{SM}}, \{(n_j, \textbf{id}, h^{\text{ACC}})\}_j)$ simultaneously minimizes the individual penalties of all the agents, and hence the social penalty (see (5)). Moreover, as no agent can sample both distributions, we have $p_i^{\text{IND}} = \infty$ for all agents $i \in [m]$. Hence, we can simply minimize the social penalty without any constraints to obtain $\inf_{(M', s') \in \Theta} P(M', s')$ (see (5), (7), and (10)).

To do so, let us write the social penalty $P(M_{\text{SM}}, s_n^{\text{COM}})$ as follows,

$$P(M_{\text{SM}}, s_n^{\text{COM}}) = \sum_{i=1}^m p_i(M_{\text{SM}}, \{(n_j, \textbf{id}, h^{\text{ACC}})\}_j) = \sum_{i=1}^m \left( \sum_{k=1}^2 \left( \frac{\sigma^2}{\sum_{j=1}^m n_{j,k}} + c_{i,k} n_{i,k} \right) \right)$$

$$= \sum_{k=1}^2 \left( \frac{m\sigma^2}{\sum_{j=1}^m n_{j,k}} + \sum_{i=1}^m c_{i,k} n_{i,k} \right)$$

As $c_{1,2} = \infty$, we can set $n_{1,2} = 0$; otherwise the social penalty would be infinite. Similarly, we can set $n_{i,1} = 0$ for all $i \in \{2, \ldots, m\}$. This gives us,

$$P(M_{\text{SM}}, \{(n_j, \textbf{id}, h^{\text{ACC}})\}_j) = \frac{m\sigma^2}{n_{1,1}} + \gamma n_{1,1} + \frac{m\sigma^2}{\sum_{j=2}^m n_{j,2}} + \gamma \sum_{j=2}^m n_{j,2}.$$

Therefore, the RHS can be minimized by choosing any $n$ which satisfies

$$n_{1,1} = \sigma \sqrt{\frac{m}{\gamma}}, \quad n_{1,2} = 0, \qquad n_{i,1} = 0, \quad \sum_{i=2}^m n_{i,2} = \sigma \sqrt{\frac{m}{\gamma}} \tag{20}$$

Therefore the optimum social penalty is,

$$\inf_{(M', s') \in \Theta} P(M', s') = P(M_{\text{SM}}, s_n^{\text{COM}}) = 4\sigma\sqrt{m\gamma}. \tag{21}$$

**Proof intuition and challenges.** We will first intuitively describe the plan of attack and the associated challenges. In our construction above, to minimize the social penalty, we need to collect $\sigma\sqrt{m/\gamma}$ points from both distributions. Since agent 1 is the only agent collecting data from the first distribution $\mathcal{N}(\mu_1, \sigma^2)$, she is responsible for collecting all of that data. Though costly, this is still IR for agent 1 as she would have a penalty of $\infty$ working alone. However, no mechanism can validate agent 1's submission because she is the only one with access to distribution 1. Therefore, agent 1 will only collect an amount of data to minimize her own penalty, leaving the others with significantly less data than they would have in $n^{\text{B}}$, leading to our hardness result.

In particular, suppose a mechanism explicitly penalizes agent 1 for submitting only a small amount of data, say, by adding large noise to the estimate for $\mu_2$, or even not returning any information about $\mu_2$ at all. Then, agent 1 can fabricate a large amount of data by sampling from some normal distribution $\mathcal{N}(\widetilde{\mu}_1, \sigma^2)$ at no cost, and submit this fabricated dataset instead. Intuitively, we should not expect a mechanism to be able to determine if the agent is submitting data from the true distribution $\mathcal{N}(\mu_1, \sigma^2)$ or the fabricated one $\mathcal{N}(\widetilde{\mu}_1, \sigma^2)$ as only agent 1 has access to distribution 1. Hence, the best that a mechanism could hope for is to simply ask agent 1 to collect an amount of data to minimize her own penalty and submit it truthfully.

Formalizing this intuition however is technically challenging. In particular, our proof cannot arbitrarily choose $\widetilde{\mu}_1$ when fabricating data for agent 1. For instance, if she arbitrarily chose $\widetilde{\mu}_1 = 1.729$, this would work poorly against a mechanism which explicitly penalizes agent 1 for submitting data whose mean is close to $1.729$. The key challenge is in showing that for *any* mechanism, there is always fabricated data which agent 1 can submit to the mechanism so that the information she receives about $\mu_2$ is not compromised. We will now formally prove Theorem 4 using this intuition.

**Notation.** Unless explicitly stated otherwise, throughout this proof, we will treat $n_{1,1}$, $n_{1,2}$, $f_1$, and $h_1 = (h_{1,1}, h_{1,2})$ as random quantities sampled from a mixed strategy for agent 1. For instance, if we write $\mathbb{P}_{s_1}(n_{1,1} = 5)$, we are referring to the probability that, when agent 1 follows mixed strategy $s_1$, she collects 5 samples from the first distribution.

*Proof of Theorem 4.* Let $M \in \mathcal{M}$ be a mechanism and let $s^\star = (s_1^\star, \ldots, s_m^\star) \in \Delta(\mathcal{S})^m$ be a (possibly mixed) Nash equilibrium of $M$. First observe that if $\mathbb{P}_{s_1^\star}(n_{1,2} > 0) > 0$, i.e if agent 1 were to collect samples from the second distribution with nonzero probability, then $p_1(M, s^\star) = \infty$ which implies $P(M, s^\star) = \infty$, and we are done. Hence, going forward, we will assume $\mathbb{P}_{s_1^\star}(n_{1,2} = 0) = 1$. Similarly, we can assume $\mathbb{P}_{s_i^\star}(n_{i,1} = 0) = 1$. Note that we can also write any estimator $h_1 : \mathcal{D} \times \mathcal{D} \times \mathcal{I} \to \mathbb{R}^2$ for agent 1 as $h_1 = (h_{1,1}, h_{1,2})$ where $h_{1,k} : \mathcal{D} \times \mathcal{D} \times \mathcal{I} \to \mathbb{R}$ is her estimate for $\mu_k$.

The key technical ingredient of this proof is Lemma 6, stated below. It states that instead of following the recommended Nash strategy $s_1^\star$, agent 1 can follow an alternative strategy $\widetilde{s}_1$ in which she submits a fabricated dataset and not be worse off. In particular, even if $s_1^\star$ is pure, $\widetilde{s}_1$ may be a mixed (randomized) strategy where the randomness comes from sampling (fabricating) data. She can then use the legitimate data she collected to estimate $\mu_1$ and the information set she receives from submitting the fabricated data to estimate $\mu_2$. In $\widetilde{s}_1$, her submission function $\widetilde{f}_1$ and estimator $\widetilde{h}_{1,2}$ for $\mu_2$ may be random quantities which are determined by the fabricated data and do not depend on the data $X_1 = (X_{1,1}, \varnothing)$ she collects. We have proved Lemma 6 in Appendix F.3.1.

**Lemma 6.** *Let $M \in \mathcal{M}$ and $s^\star$ be a Nash equilibrium in $M$. Then, there exists a strategy $\widetilde{s}_1 \in \Delta(\mathcal{S})$ for agent 1, such that*

$$p_1(M, s^\star) = p_1(M, (\widetilde{s}_1, s_{-1}^\star))$$

$$= \mathop{\mathbb{E}}_{\widetilde{s}_1}\left[\frac{\sigma^2}{n_{1,1}} + \gamma n_{1,1}\right] + \sup_{\mu_2} \mathop{\mathbb{E}}_{(\widetilde{s}_1, s_{-1}^\star)}\left[\mathop{\mathbb{E}}_{\substack{X_2 \ldots X_m \\ M}}\left[(h_{1,2}(D, f_1(D), A_1) - \mu_2)^2\right]\right]$$

*for any $D \in \mathcal{D}$. Additionally, let $\left((\widetilde{n}_{1,1}, 0), \widetilde{f}_1, (\widetilde{h}_{1,1}, \widetilde{h}_{1,2})\right)$ be drawn from $\widetilde{s}$ and let $((n_{1,1}^\star, 0), f_1^\star, (h_{1,1}^\star, h_{1,2}^\star))$ be drawn from $s_1^\star$. The following statements about these random variables are true*

1. *$\widetilde{n}_{1,1} = n_{1,1}^\star$, i.e. the amount of data the agent collects in both strategies is the same.*

2. *$\widetilde{h}_{1,1}(X_1, \cdot, \cdot) = h_1^{\mathrm{SM}}(X_1)$, i.e. agent 1 uses the sample mean of the data she collected from the first distribution to estimate $\mu_1$, regardless of what she submits and the information set she receives from the mechanism.*

3. *$\widetilde{h}_{1,2}$ is constant with respect to (i.e does not depend on) $\mu_1$ and $X_1 = (X_{1,1}, \varnothing)$, i.e. the agent's estimator for $\mu_2$ does not use the data she collects from the first distribution.*

We will organize the remainder of the proof into two steps:

- **Step 1.** Using Lemma 6, we will show that in any Nash equilibrium $s^\star$, agent 1 always collects $\sigma/\sqrt{\gamma}$ points which is a factor $\sqrt{m}$ less than the amount in $n^{\mathrm{B}}$ (see (20)), that is $\mathbb{P}_{s_1^\star}(n_{1,1} \neq \sigma/\sqrt{\gamma}) = 0$.

- **Step 2.** Consequently, we show that all other agents will incur a large penalty, resulting in large social penalty.

**Step 1.** Let $n_{1,1}^{\mathrm{IND}} := \sigma/\sqrt{\gamma}$. In this step we will show that $\mathbb{P}_{s_1^\star}(n_{1,1} \neq n_{1,1}^{\mathrm{IND}}) = 0$.

Suppose for a contradiction that $\mathbb{P}_{s^\star}(n_{1,1} \neq n_{1,1}^{\mathrm{IND}}) > 0$. Let $\widetilde{s}_1$ be the strategy that Lemma 6 claims exists. Now define the strategy $s_1'$ to be the same as $\widetilde{s}_1$ except that agent 1 always collects $n_{1,1}^{\mathrm{IND}}$ points from distribution 1. Concretely, in $s_1'$, the agent first samples $((n_{1,1}, 0), f_1, (h_{1,1}, h_{1,2})) \sim \widetilde{s}_1$, and then executes $\left((n_{1,1}^{\mathrm{IND}}, 0), f_1, (h_{1,1}, h_{1,2})\right)$. Let $s' := (s_1', s_{-1}^\star)$ be the strategy profile where all other agents are following the recommended Nash strategies while agent 1 is following $s_1'$. Using the fact that in $s_1'$ the agent always collects $n_{1,1}^{\mathrm{IND}}$ points and uses the sample mean for $h_{1,1}$ (Lemma 6), we can write

$$p_1(M, s') = \sup_{\mu_1, \mu_2} \mathop{\mathbb{E}}_{s'}\left[\mathop{\mathbb{E}}_{\substack{X_1 \ldots X_m \\ M}}\left[\sum_{k \in \{1,2\}}(h_{1,k}(X_1, f_1(X_1), A_1) - \mu_k)^2\right] + \gamma n_{1,1}\right]$$

$$= \sup_{\mu_1, \mu_2} \mathop{\mathbb{E}}_{s'}\left[\mathop{\mathbb{E}}_{\substack{X_1 \ldots X_m \\ M}}\left[(h_{1,2}(X_1, f_1(X_1), A_1) - \mu_2)^2\right]\right] + \frac{\sigma^2}{n_{1,1}^{\mathrm{IND}}} + \gamma n_{1,1}^{\mathrm{IND}}.$$

Next, Lemma 6 also tells us that in $\widetilde{s}_1$, and hence in $s_1'$, neither $f_1$ nor $h_{1,2}$ depends on the actual dataset $X_1 = (X_{1,1}, \varnothing)$ agent 1 has collected. Hence, we can replace $X_1$ with any $D \in \mathcal{D}$ and drop the dependence on $X_1$ and consequently $\mu_1$ in

the third term of the RHS above. This leads us to,

$$p_1\left(M, s'\right) = \sup_{\mu_2} \mathbb{E}_{s'} \left[ \mathbb{E}_{\substack{X_2 \dots X_m \\ M}} \left[ \left(h_{1,2}\left(D, f_1(D), A_1\right) - \mu_2\right)^2 \right] \right] + \frac{\sigma^2}{n_{1,1}^{\text{IND}}} + \gamma n_{1,1}^{\text{IND}}.$$

Next, by property 1 of $\widetilde{s}_1$ in Lemma 6, and our assumption (for the contradiction), we have $\mathbb{P}_{\widetilde{s}_1}(n_{1,1} \neq n_{1,1}^{\text{IND}}) = \mathbb{P}_{s^\star}(n_{1,1} \neq n_{1,1}^{\text{IND}}) > 0$. Noting that $\frac{\sigma^2}{x} + \gamma x$ is strictly convex in $x$ and is minimized by $x = n_{1,1}^{\text{IND}} = \sigma/\sqrt{\gamma}$, we have

$$
\begin{aligned}
p_1\left(M, (s_1', s_{-1}^\star)\right) &= \left( \frac{\sigma^2}{n_{1,1}^{\text{IND}}} + \gamma n_{1,1}^{\text{IND}} \right) + \sup_{\mu_2} \mathbb{E}_{(s_1', s_{-1}^\star)} \left[ \mathbb{E}_{\substack{X_2 \dots X_m \\ M}} \left[ \left(h_{1,2}\left(D, f_1(D), A_1\right) - \mu_2\right)^2 \right] \right] \\
&< \mathbb{E}_{\widetilde{s}_1} \left[ \frac{\sigma^2}{n_{1,1}} + \gamma n_{1,1} \right] + \sup_{\mu_2} \mathbb{E}_{(s_1', s_{-1}^\star)} \left[ \mathbb{E}_{\substack{X_2 \dots X_m \\ M}} \left[ \left(h_{1,2}\left(D, f_1(D), A_1\right) - \mu_2\right)^2 \right] \right] \\
&= \mathbb{E}_{\widetilde{s}_1} \left[ \frac{\sigma^2}{n_{1,1}} + \gamma n_{1,1} \right] + \sup_{\mu_2} \mathbb{E}_{(\widetilde{s}_1, s_{-i}^\star)} \left[ \mathbb{E}_{\substack{X_2 \dots X_m \\ M}} \left[ \left(h_{1,2}\left(D, f_1(D), A_1\right) - \mu_2\right)^2 \right] \right].
\end{aligned}
$$

Above, the final step simply observes that $s' = (s_1', s_{-1}^\star)$ and that $f_1$ and $h_{1,2}$ are almost surely equal in both $s_1'$ and $\widetilde{s}_1$. By Lemma 6, the last expression above is equal to $p_1(M, s^\star)$, which gives us $p_1(M, (s_1', s_{-1}^\star)) < p_1(M, s^\star)$ which contradicts the fact that $s^\star$ is a Nash equilibrium, as agent 1 can improve their penalty by switching to $s_1'$. Therefore, we conclude $\mathbb{P}_{s_1^\star}(n_{1,1} \neq n_{1,1}^{\text{IND}}) = 0$.

**Step 2.** We will now show that agents $2 \cdots m$ will have a large penalty, which in turn results in large social penalty. Consider the penalty $p_i(M, s^\star)$ for any agent $i \in \{2, \dots, m\}$ under $s^\star$. Recall that $\mathbb{P}_{s_i^\star}(n_{i,1} = 0) = 1$ for all such agents (otherwise their individual penalties, and hence the social penalty would be infinite). We can write,

$$
\begin{aligned}
p_i\left(M, s^\star\right) &= \sup_{\mu_1, \mu_2} \mathbb{E}_{s^\star} \left[ \mathbb{E}_{\substack{X_1 \dots X_m \\ M}} \left[ \sum_{k \in \{1,2\}} \left(h_{i,k}\left(X_i, f_i(X_i), I_i\right) - \mu_k\right)^2 \right] + \gamma n_{i,2} \right] \\
&\geq \sup_{\mu_1, \mu_2} \mathbb{E}_{s^\star} \left[ \mathbb{E}_{\substack{X_1 \dots X_m \\ M}} \left[ \left(h_{i,1}\left(X_i, f_i(X_i), I_i\right) - \mu_1\right)^2 \right] \right] \\
&\geq \frac{\sigma^2}{n_{1,1}^{\text{IND}}} = \sigma\sqrt{\gamma}.
\end{aligned}
$$

Above we have dropped some non-negative terms in the second step and then in the third step used the fact that we cannot do better than the (minimax optimal) sample mean of agent 1's submission $X_{1,1}$ to estimate $\mu_1$. Similarly, agent 1's penalty will be at least $\sigma\sqrt{\gamma}$ as she collects $\sigma/\sqrt{\gamma}$ points.

Therefore, the social penalty $p_i(M, s^\star)$ is at least $m\sigma\sqrt{\gamma}$. On the other hand, from (21), we have, $P(M_{\text{SM}}, \{(n_j, \text{id}, h^{\text{ACC}})\}_j) = 4\sigma\sqrt{m\gamma}$. This gives us

$$\frac{P(M, s^\star)}{P(M_{\text{SM}}, \{(n_j, \text{id}, h^{\text{ACC}})\}_j)} \geq \frac{\sqrt{m}}{4},$$

for any mechanism $M$ and Nash equilibrium $s^\star$ of $M$. $\qquad\square$

It is worth mentioning that our anlaysis in step 2 to bound the social penalty at $(M, s^\star)$ is quite loose in the constants as we have dropped the penalty due to the estimation error and data collections of the second distribution.

F.3.1. PROOF OF LEMMA 6

At a high level, Lemma 6 says that we can decompose agent 1's penalty into two parts, one for each distribution. The first term is the sum of the estimation cost for distribution 1 plus to cost to collect the data from distribution 1. The second term

is the estimation error for the second distribution based on the information set provided by the mechanism. Notice that the second term only involves a supremum over $\mu_2$ and an expectation over the data drawn from distribution 2 as opposed to a supremum over both of $\mu_1, \mu_2$ and an expectation over all of the data. This is because the submission and estimation functions sampled from $\widetilde{s}$ will be constant with respect to $\mu_1$ and the data drawn from distribution 1.

*Proof of Lemma 6.* We will break this proof down into four steps:

- **Step 1.** Using a sequence of priors on $\mu_1, \{\Lambda_\tau\}_{\tau \geq 1}$, we will construct a sequence of lower bounds on $p_1(M, s^\star)$. Furthermore, the limit as $\tau \to \infty$ of these lower bounds will also be a lower bound on $p_1(M, s^\star)$.

- **Step 2.** We will show that this lower bound is achieved for a careful choice of $\widetilde{s} = (\widetilde{s}_1, s^\star_{-1})$ which satisfies the properties listed in Lemma 6.

- **Step 3.** Using the fact that $s^\star_1$ is the optimal strategy for agent 1 (assuming the other agents follow $s^\star_{-1}$), we can upper bound $p_1(M, s^\star)$ by $p_1(M, (\widetilde{s}_1, s^\star_{-1}))$.

- **Step 4.** We will show that our upper bound for $p_1(M, s^\star)$ matches our lower bound for $p_1(M, s^\star)$. Therefore, $p_1(M, s^\star)$ equals our upper/lower bound.

**Step 1.** For the sake of convenience, let $\Delta(\mathcal{S})' := \left\{ s \in \Delta(\mathcal{S}) : n_{1,1} = n^\star_{1,1} \right\}$. By the definition of $p_1(M, s^\star)$ and the fact that $s^\star$ is a Nash equilibrium, we have

$$
p_1(M, s^\star) = \sup_{\mu_1, \mu_2} \mathbb{E}_{s^\star} \left[ \mathbb{E}_{\substack{X_1 \ldots X_m \\ M}} \left[ \sum_{k \in \{1,2\}} (h_{1,k}(X_1, f_1(X_1), I_1) - \mu_k)^2 \right] + \gamma n_{1,1} \right]
$$

$$
= \inf_{s_1 \in \Delta(\mathcal{S})'} \sup_{\mu_1, \mu_2} \mathbb{E}_{(s_1, s^\star_{-1})} \left[ \mathbb{E}_{\substack{X_1 \ldots X_m \\ M}} \left[ \sum_{k \in \{1,2\}} (h_{1,k}(X_1, f_1(X_1), I_1) - \mu_k)^2 \right] + \gamma n_{1,1} \right]. \tag{22}
$$

If the second equality did not hold, then agent 1 could deviate from $s^\star_1$ by choosing a strategy with a different distribution over the set of submission/estimation functions and achieve a lower penalty (which would contradict that $s^\star$ is a Nash equilibrium).

We will now construct a sequence of priors parameterized by $\tau \geq 1$ and use them to lower bound (22) and thus $p_1(M, s^\star)$. Let $\Lambda_\tau := \mathcal{N}(0, \tau^2)$. Since the supremum is larger than the average we have

$$
\inf_{s_1 \in \Delta(\mathcal{S})'} \sup_{\mu_1, \mu_2} \mathbb{E}_{(s_1, s^\star_{-1})} \left[ \mathbb{E}_{\substack{X_1 \ldots X_m \\ M}} \left[ \sum_{k \in \{1,2\}} (h_{1,k}(X_1, f_1(X_1), I_1) - \mu_k)^2 \right] + \gamma n_{1,1} \right]
$$

$$
\geq \inf_{s_1 \in \Delta(\mathcal{S})'} \sup_{\mu_2} \mathbb{E}_{\mu_1 \sim \Lambda_\tau} \left[ \mathbb{E}_{(s_1, s^\star_{-1})} \left[ \mathbb{E}_{\substack{X_1 \ldots X_m \\ M}} \left[ \sum_{k \in \{1,2\}} (h_{1,k}(X_1, f_1(X_1), I_1) - \mu_k)^2 \right] + \gamma n_{1,1} \right] \right]. \tag{23}
$$

We can switch the order of the expectation for $\mu_1$ and $X_1$ by rewriting $\mu_1$ conditioned on the marginal distribution of $X_1$. For convenience, let $D(\tau)$ denote the marginal distribution for $X_1$. Changing the order, (23) becomes

$$
\inf_{s_1 \in \Delta(\mathcal{S})'} \sup_{\mu_2} \mathbb{E}_{(s_1, s^\star_{-1})} \left[ \mathbb{E}_{\substack{X_2 \ldots X_m \\ M}} \left[ \mathbb{E}_{X_1 \sim D(\tau)} \left[ \mathbb{E}_{\mu_1} \left[ \sum_{k \in \{1,2\}} (h_{1,k}(X_1, f_1(X_1), I_1) - \mu_k)^2 \Big| X_1 \right] \right] \right] + \gamma n_{1,1} \right]. \tag{24}
$$

Consider the term $\mathbb{E}_{\mu_1} \left[ (h_{1,1}(X_1, f_1(X_1), I_1) - \mu_1)^2 | X_1 \right]$. It is well known (Lehmann & Casella, 2006) that for a normal distribution, the Bayes' risk is minimized by the Bayes estimator and and that the Bayes' risk under this estimator is simply the posterior variance, i.e.

$$
\mathbb{E}_{\mu_1} \left[ (h_{1,1}(X_1, f_1(X_1), I_1) - \mu_1)^2 | X_1 \right] = \frac{\sigma^2}{n_{1,1} + \frac{\sigma^2}{\tau^2}} .
$$

Following this observation, we rewrite (24) as

$$
\inf_{s_1 \in \Delta(\mathcal{S})'} \sup_{\mu_2} \mathop{\mathbb{E}}_{(s_1, s_{-1}^\star)} \left[ \mathop{\mathbb{E}}_{\substack{X_2 \ldots X_m \\ M}} \left[ \mathop{\mathbb{E}}_{X_1 \sim D(\tau)} \left[ \mathop{\mathbb{E}}_{\mu_1} \left[ \frac{\sigma^2}{n_{1,1} + \frac{\sigma^2}{\tau^2}} + (h_{1,2}(X_1, f_1(X_1), I_1) - \mu_2)^2 \, \middle| \, X_1 \right] \right] \right] \right] + \gamma n_{1,1} \right]
$$

$$
= \inf_{s_1 \in \Delta(\mathcal{S})'} \sup_{\mu_2} \mathop{\mathbb{E}}_{(s_1, s_{-1}^\star)} \left[ \mathop{\mathbb{E}}_{\substack{X_2 \ldots X_m \\ M}} \left[ \mathop{\mathbb{E}}_{X_1 \sim D(\tau)} \left[ \mathop{\mathbb{E}}_{\mu_1} \left[ (h_{1,2}(X_1, f_1(X_1), I_1) - \mu_2)^2 \, \middle| \, X_1 \right] \right] \right] + \frac{\sigma^2}{n_{1,1} + \frac{\sigma^2}{\tau^2}} + \gamma n_{1,1} \right]. \quad (25)
$$

We now observe that when we condition on the data from distribution 1, $(h_{1,2}(X_1, f_1(X_1), I_1) - \mu_2)^2 \mid X_1$ is constant with respect to $\mu_1$. This allows us to drop the expectation over $\mu_1$ which let us rewrite (25) as

$$
\inf_{s_1 \in \Delta(\mathcal{S})'} \sup_{\mu_2} \mathop{\mathbb{E}}_{(s_1, s_{-1}^\star)} \left[ \mathop{\mathbb{E}}_{\substack{X_2 \ldots X_m \\ M}} \left[ \mathop{\mathbb{E}}_{X_1 \sim D(\tau)} \left[ (h_{1,2}(X_1, f_1(X_1), I_1) - \mu_2)^2 \right] \right] + \frac{\sigma^2}{n_{1,1} + \frac{\sigma^2}{\tau^2}} + \gamma n_{1,1} \right]. \quad (26)
$$

If we take the limit as $\tau \to \infty$ of (26), using monotone convergence and that $s_1 \in \Delta(\mathcal{S})'$, we obtain the following lower bound on equation $p_1(M, s^\star)$

$$
p_1(M, s^\star) \geq \mathop{\mathbb{E}}_{s_1^\star} \left[ \frac{\sigma^2}{n_{1,1}} + \gamma n_{1,1} \right]
$$

$$
+ \lim_{\tau \to \infty} \inf_{s_1 \in \Delta(\mathcal{S})'} \sup_{\mu_2} \mathop{\mathbb{E}}_{(s_1, s_{-1}^\star)} \left[ \mathop{\mathbb{E}}_{\substack{X_2 \ldots X_m \\ M}} \left[ \mathop{\mathbb{E}}_{X_1 \sim D(\tau)} \left[ (h_{1,2}(X_1, f_1(X_1), I_1) - \mu_2)^2 \right] \right] \right]. \quad (27)
$$

**Step 2.** Lemma 7 shows that the right hand term in (27) is constant in $\tau$ which allows us to drop the limit. Additionally, Lemma 7, and the remarks that precede its proof, describe and show how sampling $X_1$ from $D(\tau)$ can be incorporated into the strategy $s_1$ because $D(\tau)$ a probability distribution that does not depend on $\mu_1, \mu_2$ or $X_2, \ldots, X_m$. Therefore, the submission/estimation functions used in estimating distribution 2 do not depend on the data collected from distribution 1. Thus, without loss of generality we can write the lower bound in (27) as

$$
\mathop{\mathbb{E}}_{s_1^\star} \left[ \frac{\sigma^2}{n_{1,1}} + c_{1,1} n_{1,1} \right] + \underbrace{\inf_{s_1 \in \Delta(\mathcal{S})'} \sup_{\mu_2} \mathop{\mathbb{E}}_{(s_1, s_{-1}^\star)} \left[ \mathop{\mathbb{E}}_{\substack{X_2 \ldots X_m \\ M}} \left[ (h_{1,2}(D, f_1(D), I_1) - \mu_2)^2 \right] \right]}_{L :=} \quad (28)
$$

where $D \in \mathcal{D}$ is any dataset. Therefore, we have

$$
p_1(M, s^\star) \geq \mathop{\mathbb{E}}_{s_1^\star} \left[ \frac{\sigma^2}{n_{1,1}} + \gamma n_{1,1} \right] + L. \quad (29)
$$

However, a priori, we do not know that there exists a single strategy $s_1 \in \Delta(\mathcal{S})'$ which actually achieves estimation error of $L$ on distribution 2 (i.e. the error corresponding to plugging the strategy into the right hand term of (28)). Suppose for a contradiction that this is not the case. We know that for all $\varepsilon > 0$, we can find a strategy whose estimation error for distribution 2 comes within $\varepsilon$ of $L$. Furthermore, for each of these strategies, the distribution over $h_{1,1}$ does not affect the estimation error of distribution 2 since the right hand term does not involve $h_{1,1}$. Therefore, we can suppose that these strategies all select $h_{1,1}^\varepsilon = h_1^{\mathrm{SM}}(X_1)$. More concretely, $\forall \varepsilon > 0, \exists s_1^\varepsilon \in \Delta(\mathcal{S})'$ such that $h_{1,1}^\varepsilon = h_1^{\mathrm{SM}}(X_1)$ and

$$
\sup_{\mu_2} \mathop{\mathbb{E}}_{(s_1^\varepsilon, s_{-1}^\star)} \left[ \mathop{\mathbb{E}}_{\substack{X_2 \ldots X_m \\ M}} \left[ (h_{1,2}(D, f_1(D), I_1) - \mu_2)^2 \right] \right] < L + \varepsilon.
$$

If we plug the strategy $(s_1^\varepsilon, s_{-1}^\star)$ into equation (22) we obtain the following upper bound on $p_1(M, s^\star)$

$$
p_1(M, s^\star) \leq \sup_{\mu_1, \mu_2} \mathop{\mathbb{E}}_{(s_1^\varepsilon, s_{-1}^\star)} \left[ \mathop{\mathbb{E}}_{\substack{X_1 \ldots X_m \\ M}} \left[ \sum_{k \in \{1, 2\}} (h_{1,k}(X_1, f_1(X_1), I_1) - \mu_k)^2 \right] \right] + \mathop{\mathbb{E}}_{s_1^\varepsilon} [\gamma n_{1,1}]
$$

$$
= \mathop{\mathbb{E}}_{s_1^\varepsilon}\left[\frac{\sigma^2}{n_{1,1}}\right] + \sup_{\mu_2}\mathop{\mathbb{E}}_{(s_1^\varepsilon, s_{-1}^\star)}\left[\mathop{\mathbb{E}}_{\substack{X_2 \dots X_m \\ M}}\left[(h_{1,2}(D, f_1(D), I_1) - \mu_2)^2\right]\right] + \mathop{\mathbb{E}}_{s_1^\varepsilon}\left[\gamma n_{1,1}\right]
$$

$$
= \mathop{\mathbb{E}}_{s_1^\star}\left[\frac{\sigma^2}{n_{1,1}} + \gamma n_{1,1}\right] + \sup_{\mu_2}\mathop{\mathbb{E}}_{(s_1^\varepsilon, s_{-1}^\star)}\left[\mathop{\mathbb{E}}_{\substack{X_2 \dots X_m \\ M}}\left[(h_{1,2}(D, f_1(D), I_1) - \mu_2)^2\right]\right]
$$

$$
= \mathop{\mathbb{E}}_{s_1^\star}\left[\frac{\sigma^2}{n_{1,1}} + \gamma n_{1,1}\right] + L + \varepsilon \; . \tag{30}
$$

As we are supposing that the lower bound in inequality (29) is not actually achieved for any $s_1 \in \Delta(\mathcal{S})'$, the inequality is strict. Combining inequalities (29) and (30), we get

$$
\mathop{\mathbb{E}}_{s_1^\star}\left[\frac{\sigma^2}{n_{1,1}} + \gamma n_{1,1}\right] + L + \varepsilon > p_1(M, s^\star) > \mathop{\mathbb{E}}_{s_1^\star}\left[\frac{\sigma^2}{n_{1,1}} + \gamma n_{1,1}\right] + L \; .
$$

However, this yields a contradiction since these inequalities hold $\forall \varepsilon > 0$. Therefore, we conclude that $\exists (\widetilde{n}_1, \widetilde{f}_1, \widetilde{h}_1) = \widetilde{s}_1 \in \Delta(\mathcal{S})'$ such that

$$
\sup_{\mu_2}\mathop{\mathbb{E}}_{(\widetilde{s}_1, s_{-1}^\star)}\left[\mathop{\mathbb{E}}_{\substack{X_2 \dots X_m \\ M}}\left[(h_{1,2}(D, f_1(D), I_1) - \mu_2)^2\right]\right] = L \; .
$$

Furthermore, for this strategy, the distribution over $h_{1,1}$ does not affect the above term (the estimation error of distribution 2) since $h_{1,1}$ doesn't appear in the expression. Therefore, we can suppose that $\widetilde{s}_1$ always selects $\widetilde{h}_{1,1} = h_1^{\mathrm{SM}}(X_1)$. Also observe that since we were able to replace $X_1$ with $D$ in step 2, we have that $\widetilde{f}_1, \widetilde{h}_{1,2}$ are constant with respect to $\mu_1$ and $X_1$. Additionally, since $\widetilde{s}_1 \in \Delta(\mathcal{S})'$ we know that $\widetilde{n}_{1,1} = n_{1,1}^\star$. Therefore, $\widetilde{s}_1$ satisfies our three desired properties which are listed below.

    1. $\widetilde{n}_{1,1} = n_{1,1}^\star$.      2. $\widetilde{h}_{1,1} = h_1^{\mathrm{SM}}(X_1)$.      3. $\widetilde{h}_{1,2}$ is constant with respect to $\mu_1$ and $X_1$.

Let $\widetilde{s}$ be defined as $\widetilde{s} := (\widetilde{s}_1, s_{-1}^\star)$. We can now write the lower bound in equation (29) as

$$
p_1(M, s^\star) \geq \mathop{\mathbb{E}}_{\widetilde{s}_1}\left[\frac{\sigma^2}{n_{1,1}} + \gamma n_{1,1}\right] + \sup_{\mu_2}\mathop{\mathbb{E}}_{\widetilde{s}}\left[\mathop{\mathbb{E}}_{\substack{X_2 \dots X_m \\ M}}\left[(h_{1,2}(D, f_1(D), I_1) - \mu_2)^2\right]\right] \; . \tag{31}
$$

**Step 3.** We can now take $\widetilde{s}_1$, plug it into equation (22), and use properties two and three of $\widetilde{s}_1$ to get the following upper bound on $p_1(M, s^\star)$

$$
p_1(M, s^\star) \leq \sup_{\mu_1, \mu_2}\mathop{\mathbb{E}}_{\widetilde{s}}\left[\mathop{\mathbb{E}}_{\substack{X_1 \dots X_m \\ M}}\left[\sum_{k \in \{1,2\}}(h_{i,k}(X_1, f_1(X_1), I_1) - \mu_k)^2\right]\right] + \mathop{\mathbb{E}}_{\widetilde{s}_1}\left[\gamma n_{1,1}\right]
$$

$$
= \mathop{\mathbb{E}}_{\widetilde{s}_1}\left[\frac{\sigma^2}{n_{1,1}} + \gamma n_{1,1}\right] + \sup_{\mu_2}\mathop{\mathbb{E}}_{\widetilde{s}}\left[\mathop{\mathbb{E}}_{\substack{X_2 \dots X_m \\ M}}\left[(h_{1,2}(D, f_1(D), I_1) - \mu_2)^2\right]\right] \; . \tag{32}
$$

**Step 4.** But notice that (31) and (32) provide matching lower and upper bounds for $p_1(M, s^\star)$. We can therefore conclude that

$$
p_1(M, s^\star) = \mathop{\mathbb{E}}_{\widetilde{s}_1}\left[\frac{\sigma^2}{n_{1,1}} + \gamma n_{1,1}\right] + \sup_{\mu_2}\mathop{\mathbb{E}}_{\widetilde{s}}\left[\mathop{\mathbb{E}}_{\substack{X_2 \dots X_m \\ M}}\left[(h_{1,2}(D, f_1(D), I_1) - \mu_2)^2\right]\right] \; .
$$

This completes the proof. $\qquad\qquad\qquad\qquad\qquad\qquad\qquad\qquad\qquad\qquad\qquad\qquad\qquad\qquad\quad\square$

**Lemma 7.** *The expression introduced in Step 1. of Lemma 6 and given below is constant in $\tau$.*

$$z^1(\tau) := \inf_{s_1 \in \Delta(\mathcal{S})'} \underbrace{\sup_{\mu_2} \mathbb{E}_{(s_1, s_{-1}^\star)} \left[ \mathbb{E}_{\substack{X_2 \ldots X_m \\ M}} \left[ \mathbb{E}_{X_1 \sim D(\tau)} \left[ (h_{1,2}(X_1, f_1(X_1), I_1) - \mu_2)^2 \right] \right] \right]}_{z^2(\tau, s_1) :=}$$

**Remark 1.** *For convenience, we have defined the quantities $z^1(\tau)$ and $z^2(\tau, s_1)$ above. We can think of $z^1(\tau)$ as the best possible estimation error agent 1 can achieve on distribution 2 for a particular $\tau$ and any choice of $s_1 \in \Delta(\mathcal{S})'$. $z^2(\tau, s_1)$ can be thought of as the estimation error agent 1 achieves on distribution 2 for a particular $\tau$ and choice of strategy $s_1 \in \Delta(\mathcal{S})'$.*

**Remark 2.** *Here is an intuitive justification for why we should believe Lemma 7 is true. For any choice of $\tau$, $D(\tau)$ is just some probability distribution that does not depend on $\mu_1, \mu_2$ or the data drawn from these distributions. If $z^1(\tau_a) < z^1(\tau_b)$ for two different values of of tau, it means that we were able to achieve a lower estimation error by using $X_1 \sim D(\tau_a)$ than using $X_1 \sim D(\tau_b)$. However, because agent 1 can use a randomized strategy, they could draw and use data from $D(\tau_a)$ while ignoring any data drawn from $D(\tau_b)$. Therefore, the error we achieve should not be subject to the value of $\tau$. We now use this idea to prove Lemma 7.*

*Proof of Lemma 7.* Suppose for a contradiction that $\exists \tau_a \neq \tau_b$ such that $z^1(\tau_a) < z^1(\tau_b)$. A priori we do not know that $z^1(\tau_a)$ is achieved by a single choice of $s_1 \in \Delta(\mathcal{S})'$ so let $(s_1^{a,n})_{n \in \mathbb{N}}$ be a sequence such that $\lim_{n \to \infty} z^2(\tau_a, s_1^{a,n}) = z^1(\tau_b)$. But using this sequence, we can construct a different sequence $(s_1^{b,n})_{n \in \mathbb{N}}$ defined via $(n_1^{b,n}, f_1^{b,n}, h_1^{b,n}) \sim s_1^{b,n}$ such that

$$(n_1^{b,n}, f_1^{b,n}, h_1^{b,n}) = \left( n_1^{a,n}, \ f_1^{a,n}(X_1^{\tau_a}), \left( h_{1,1}^{a,n}, \ h_{1,2}^{a,n}(X_1^{\tau_a}, f_1^{a,n}(X_1^{\tau_a}), A_1) \right) \right)$$

$$\text{where} \quad (n_1^{a,n}, f_1^{a,n}, h_1^{a,n}) \sim s_1^{a,n} \quad \text{and} \quad X_1^{\tau_a} \sim D(\tau_a)$$

This construction samples a strategy from $s_1^{a,n}$ and then uses data sampled from $D(\tau_a)$ in place of the data sampled from $D(\tau_b)$. But notice that via this construction

$$z^2(\tau_a, s_1^{b,n}) = \sup_{\mu_2} \mathbb{E}_{\substack{X_2 \ldots X_m \\ M}} \left[ \mathbb{E}_{(s_1^{b,n}, s_{-1}^\star)} \left[ \mathbb{E}_{X_1 \sim D(\tau_b)} \left[ (h_{1,2}(X_1, f_1 X_1, A_1) - \mu_2)^2 \right] \right] \right]$$

$$= \sup_{\mu_2} \mathbb{E}_{\substack{X_2 \ldots X_m \\ M}} \left[ \mathbb{E}_{(s_1^{a,n}, s_{-1}^\star)} \left[ \mathbb{E}_{X_1^{\tau_a} \sim D(\tau_a)} \left[ \mathbb{E}_{X_1 \sim D(\tau_b)} \left[ (h_{1,2}(X_1^{\tau_a}, f_1 X_1^{\tau_a}, A_1) - \mu_2)^2 \right] \right] \right] \right]$$

$$= \sup_{\mu_2} \mathbb{E}_{\substack{X_2 \ldots X_m \\ M}} \left[ \mathbb{E}_{(s_1^{a,n}, s_{-1}^\star)} \left[ \mathbb{E}_{X_1^{\tau_a} \sim D(\tau_a)} \left[ (h_{1,2}(X_1^{\tau_a}, f_1 X_1^{\tau_a}, A_1) - \mu_2)^2 \right] \right] \right]$$

$$= \sup_{\mu_2} \mathbb{E}_{\substack{X_2 \ldots X_m \\ M}} \left[ \mathbb{E}_{(s_1^{a,n}, s_{-1}^\star)} \left[ \mathbb{E}_{X_1 \sim D(\tau_a)} \left[ (h_{1,2}(X_1, f_1 X_1, A_1) - \mu_2)^2 \right] \right] \right]$$

$$= z^2(\tau_a, s_1^{a,n})$$

This implies that $\lim_{n \to \infty} z^2(\tau_b, s_1^{b,n}) = \lim_{n \to \infty} z^2(\tau_a, s_1^{a,n})$. We can therefore say that $z^1(\tau_a) \geq z^1(\tau_b)$ which is a contradiction. Therefore, $z^1$ is constant in $\tau$. $\square$

## G. Technical Results

This section is comprised of technical results that are not relevant to the ideas of the proofs they are used in.

**Lemma 8.** *For all $x > 0$*

$$\text{Erfc}(x) \geq \frac{1}{\sqrt{\pi}} \left( \frac{\exp(-x^2)}{x} - \frac{\exp(-x^2)}{2x^3} \right)$$

*Proof.* First set

$$g(x) := \mathrm{Erfc}(x) - \frac{1}{\sqrt{\pi}} \left( \frac{\exp(-x^2)}{x} - \frac{\exp(-x^2)}{2x^3} \right).$$

We find that $g'(x) = \frac{-3 \exp(-x^2)}{4x^4} < 0$. Furthermore, since

$$\lim_{x \to \infty} \left( \mathrm{Erfc}(x) - \frac{1}{\sqrt{\pi}} \left( \frac{\exp(-x^2)}{x} - \frac{\exp(-x^2)}{2x^3} \right) \right) = 0$$

we conclude that $g(x) \geq 0$. $\hfill\square$

**Lemma 9.** *Let $n'_{i,k}$ be the value of $\widetilde{n}_{i,k}$ after the first while loop but before the second in Algorithm 1. Also let $T_k = \sum_{i=1}^{m} n'_{i,k}$ then*

$$\frac{\sigma^2}{n_{i,k}^{\mathrm{IND}}} + c_{i,k} n_{i,k}^{\mathrm{IND}} > 4 \left( \frac{\sigma^2}{\sum_{j=1}^{m} n'_{j,k}} + c_{i,k} n'_{i,k} \right) \Rightarrow T_k > 2 n_{i,k}^{\mathrm{IND}}$$

*Proof.* We will prove the contrapositive. Since $n_{i,k}^{\mathrm{IND}} = \frac{\sigma}{\sqrt{c_{i,k}}}$

$$
\begin{aligned}
4 \left( \frac{\sigma^2}{\sum_{j=1}^{m} n'_{j,k}} + c_{i,k} n'_{i,k} \right) &= 4 \left( \frac{\sigma^2}{T_k} + c_{i,k} n'_{i,k} \right) \\
&\geq 4 \left( \frac{\sigma^2}{2 n_{i,k}^{\mathrm{IND}}} + c_{i,k} n'_{i,k} \right) \\
&\geq 2 \frac{\sigma^2}{n_{i,k}^{\mathrm{IND}}} \\
&= \frac{\sigma^2}{n_{i,k}^{\mathrm{IND}}} + c_{i,k} n_{i,k}^{\mathrm{IND}}
\end{aligned}
$$

Therefore $T_k > 2 n_{i,k}^{\mathrm{IND}}$. $\hfill\square$

**Lemma 10.** *Suppose that $n^{\mathrm{B}}$ satisfies (8). If $i \notin V_k$ then $\frac{n_{i,k}^{\mathrm{IND}}}{2} > n_{i,k}^{\star}$.*

*Proof.* Let $\widetilde{n}$ be the corresponding vector returned in Algorithm 1, and let $n'_{i,k}$ denote the value of $\widetilde{n}$ after the first while loop but before the second while loop in Algorithm 1. Since $i \notin V_k$, we know that $\frac{\sigma^2}{n_{i,k}^{\mathrm{IND}}} + c_{i,k} n_{i,k}^{\mathrm{IND}} > 4 \left( \frac{\sigma^2}{\sum_{j=1}^{m} n'_{j,k}} + c_{i,k} n'_{i,k} \right)$. We now consider two cases:

If $n'_{i,k} = \widetilde{n}_{i,k}$: Since $n_{i,k}^{\mathrm{IND}} = \frac{\sigma}{\sqrt{c_{i,k}}}$ we have that

$$2 n_{i,k}^{\mathrm{IND}} c_{i,k} = \frac{\sigma^2}{n_{i,k}^{\mathrm{IND}}} + c_{i,k} n_{i,k}^{\mathrm{IND}} > 4 \left( \frac{\sigma^2}{\sum_{j=1}^{m} n'_{j,k}} + c_{i,k} n'_{i,k} \right) > 4 c_{i,k} n'_{i,k} \quad \Rightarrow \quad \frac{n_{i,k}^{\mathrm{IND}}}{2} > \widetilde{n}_{i,k}$$

If $n'_{i,k} \neq \widetilde{n}_{i,k}$: We know that $\widetilde{n}_{i,k}$ was changed in loop two, otherwise $n'_{i,k}$ and $\widetilde{n}_{i,k}$ would be equal since they are equal after loop one by definition. Because $\widetilde{n}_{i,k}$ was updated in the second loop, we know that $\widetilde{n}_{i,k} = \frac{(n_{i,k}^{\mathrm{IND}})^2}{T_k}$ where $T_k = \sum_{i=1}^{m} n'_{i,k}$ by the definition of Algorithm 1. Lemma 9 tells us that $T_k > 2 n_{i,k}^{\mathrm{IND}}$. Therefore $\widetilde{n}_{i,k} = \frac{(n_{i,k}^{\mathrm{IND}})^2}{T_k} < \frac{n_{i,k}^{\mathrm{IND}}}{2}$.

In both cases $\frac{n_{i,k}^{\mathrm{IND}}}{2} > \widetilde{n}_{i,k}$. Therefore $\frac{n_{i,k}^{\mathrm{IND}}}{2} > n_{i,k}^{\star}$ as $n^{\star}$ is defined in Algorithm 2 to be the value of $\widetilde{n}$ returned by Algorithm 1. $\hfill\square$

**Lemma 11.** *Suppose that $n^{\mathrm{B}}$ satisfies* (8). *Then $\frac{n_{i,k}^{\mathrm{IND}}}{n_{i,k}^{\mathrm{B}}} > \frac{1}{2}$.*

*Proof.* Let $\overline{n}_{i,k}$ be the maximum amount of data that is individually rational for agent $i$ to collect assuming that $k$ is the only distribution that $i$ is trying to estimate. By this definition we have

$$\frac{\sigma^2}{\overline{n}_{i,k} + \sum_{j \neq i} n_{j,k}^{\mathrm{B}}} + c_{i,k}\overline{n}_{i,k} = \frac{\sigma^2}{n_{i,k}^{\mathrm{IND}}} + c_{i,k}n_{i,k}^{\mathrm{IND}} = 2\sigma\sqrt{c_{i,k}}$$

Therefore $c_{i,k}\overline{n}_{i,k} < 2\sigma\sqrt{c_{i,k}} \Rightarrow \overline{n}_{i,k} < \frac{2\sigma}{\sqrt{c_{i,k}}}$. We can now write

$$\frac{n_{i,k}^{\mathrm{IND}}}{\overline{n}_{i,k}} > \frac{\frac{\sigma}{\sqrt{c_{i,k}}}}{\frac{2\sigma}{\sqrt{c_{i,k}}}} = \frac{1}{2} \ .$$

By definition $\overline{n}_{i,k} \geq n_{i,k}^{\mathrm{B}}$ and so $\frac{n_{i,k}^{\mathrm{IND}}}{n_{i,k}^{\mathrm{B}}} \geq \frac{n_{i,k}^{\mathrm{IND}}}{\overline{n}_{i,k}} > \frac{1}{2}$ which proves the claim. $\qquad\square$

**Lemma 12.** *Suppose that $n^{\mathrm{B}}$ satisfies* (8). *Let $n'$ be the value of $\widetilde{n}$ in Algorithm 1 after the first while loop but before the second while loop. Then $\frac{n_{i,k}'}{n_{i,k}^{\mathrm{B}}} > \frac{1}{2}$.*

*Proof.* Initially $\widetilde{n}_{i,k} = n_{i,k}^{\mathrm{B}}$ but may be modified by the first while loop. If $\widetilde{n}_{i,k}$ is unchanged by the first loop then $n_{i,k}' = n_{i,k}^{\mathrm{B}}$ and so $\frac{n_{i,k}'}{n_{i,k}^{\mathrm{B}}} > \frac{1}{2}$. If $\widetilde{n}_{i,k}$ is changed by the first loop then $\widetilde{n}_{i,k}$ becomes $n_{i,k}^{\mathrm{IND}}$. But since $n^{\mathrm{B}}$ satisfies (8), we know by Lemma 11 that $\frac{n_{i,k}^{\mathrm{IND}}}{n_{i,k}^{\mathrm{B}}} > \frac{1}{2}$ and $\frac{n_{i,k}'}{n_{i,k}^{\mathrm{B}}} > \frac{1}{2}$. In both cases the desired inequality holds. $\qquad\square$

**Lemma 13.** *Suppose that $n^{\mathrm{B}}$ satisfies* (8). *Let $n^\star$ be the value returned by Algorithm 1. Then $\frac{n_{i,k}^\star}{n_{i,k}^{\mathrm{B}}} > \frac{1}{2}$.*

*Proof.* The final value of $\widetilde{n}$ will be the value returned for $n^\star$ in Algorithm 1. Initially $\widetilde{n}_{i,k} = n_{i,k}^{\mathrm{B}}$ and is modified (if at all) in either the first or the second while loop. The second while loop can only increase $\widetilde{n}_{i,k}$ from $n_{i,k}^{\mathrm{B}}$ to $\frac{\left(n_{i,k}^{\mathrm{IND}}\right)^2}{T_k}$ so if $\widetilde{n}_{i,k}$ is modified by the second while loop we know that $\frac{n_{i,k}^\star}{n_{i,k}^{\mathrm{B}}} > \frac{1}{2}$. If $\widetilde{n}_{i,k}$ is unchanged by either loop then $\widetilde{n}_{i,k} = n_{i,k}^{\mathrm{B}}$ and so $\frac{n_{i,k}^\star}{n_{i,k}^{\mathrm{B}}} > \frac{1}{2}$. If $\widetilde{n}_{i,k}$ is changed by the first loop then $\widetilde{n}_{i,k}$ becomes $n_{i,k}^{\mathrm{IND}}$. But since $n^{\mathrm{B}}$ satisfies (8), we know by Lemma 11 that $\frac{n_{i,k}^{\mathrm{IND}}}{n_{i,k}^{\mathrm{B}}} > \frac{1}{2}$ and $\frac{n_{i,k}^\star}{n_{i,k}^{\mathrm{B}}} > \frac{1}{2}$. In every case the desired inequality holds. $\qquad\square$

**Lemma 14.** *Let $G_{i,k}\left(\alpha_{i,k}\right)_{\mathrm{UB}}$ be as defined in* (12). *Then we have that*

$$G_{i,k}\left(\sqrt{n_{i,k}^\star}\right)_{\mathrm{UB}} = -\frac{64(n_{i,k}^\star)^{3/2}(c_{i,k}n_{i,k}^\star(T_k)^3 + (2(n_{i,k}^\star)^2 - (T_k)^2)\sigma^2)}{(T_k)^{5/2}(T_k - 2n_{i,k}^\star)\sigma^2} \ .$$

*Proof.* The exponential terms in $G_{i,k}\left(\sqrt{n_{i,k}^\star}\right)_{\mathrm{UB}}$ cancel giving

$$G_{i,k}\left(\sqrt{n_{i,k}^\star}\right)_{\mathrm{UB}} = \overbrace{\frac{4\sqrt{n_{i,k}^\star}}{\sqrt{T_k}}\left(-1 + \frac{4T_k}{T_k - 2n_{i,k}^\star} - \frac{16c_{i,k}(n_{i,k}^\star)^2 T_k}{(T_k - 2n_{i,k}^\star)\sigma^2}\right)}^{(a)}$$
$$\underbrace{-\sqrt{2}\left(-1 + 4\left(1 + \frac{T_k}{n_{i,k}^\star}\right)\frac{n_{i,k}^\star}{T_k}\right)\left(\left(\frac{8n_{i,k}^\star}{T_k}\right)^{1/2} - \frac{1}{2}\left(\frac{8n_{i,k}^\star}{T_k}\right)^{3/2}\right)}_{(b)}$$

Now notice that

$$(a) = -\frac{4(n_{i,k}^\star)^{1/2}(T_k)^2\sigma^2}{(T_k)^{5/2}(T_k - 2n_{i,k}^\star)\sigma^2} + \frac{8(n_{i,k}^\star)^{3/2}(T_k)^2\sigma^2}{(T_k)^{5/2}(T_k - 2n_{i,k}^\star)\sigma^2}$$
$$+ \frac{16(n_{i,k}^\star)^{1/2}(T_k)^3\sigma^2}{(T_k)^{5/2}(T_k - 2n_{i,k}^\star)\sigma^2} - \frac{64c_{i,k}(n_{i,k}^\star)^{5/2}(T_k)^3}{(T_k)^{5/2}(T_k - 2n_{i,k}^\star)\sigma^2}$$

$$(b) = -\sqrt{2}\left(3 + \frac{4n_{i,k}^\star}{T_k}\right)\left(\left(\frac{8n_{i,k}^\star}{T_k}\right)^{1/2} - \frac{1}{2}\left(\frac{8n_{i,k}^\star}{T_k}\right)^{3/2}\right)$$
$$= -\sqrt{2}\left(3\left(\frac{8n_{i,k}^\star}{T_k}\right)^{1/2} - \frac{3}{2}\left(\frac{8n_{i,k}^\star}{T_k}\right)^{3/2} + \frac{8\sqrt{2}(n_{i,k}^\star)^{3/2}}{(T_k)^{3/2}} - \frac{32\sqrt{2}(n_{i,k}^\star)^{5/2}}{(T_k)^{5/2}}\right)$$
$$= -\frac{12(n_{i,k}^\star)^{1/2}(T_k)^2(T_k - 2n_{i,k}^\star)\sigma^2}{(T_k)^{5/2}(T_k - 2n_{i,k}^\star)\sigma^2} + \frac{48(n_{i,k}^\star)^{3/2}T_k(T_k - 2n_{i,k}^\star)\sigma^2}{(T_k)^{5/2}(T_k - 2n_{i,k}^\star)\sigma^2}$$
$$- \frac{16(n_{i,k}^\star)^{3/2}T_k(T_k - 2n_{i,k}^\star)\sigma^2}{(T_k)^{5/2}(T_k - 2n_{i,k}^\star)\sigma^2} + \frac{64(n_{i,k}^\star)^{5/2}(T_k - 2n_{i,k}^\star)\sigma^2}{(T_k)^{5/2}(T_k - 2n_{i,k}^\star)\sigma^2}$$
$$= -\frac{12(n_{i,k}^\star)^{1/2}(T_k)^3\sigma^2}{(T_k)^{5/2}(T_k - 2n_{i,k}^\star)\sigma^2} + \frac{24(n_{i,k}^\star)^{3/2}(T_k)^2\sigma^2}{(T_k)^{5/2}(T_k - 2n_{i,k}^\star)\sigma^2}$$
$$+ \frac{48(n_{i,k}^\star)^{3/2}(T_k)^2\sigma^2}{(T_k)^{5/2}(T_k - 2n_{i,k}^\star)\sigma^2} - \frac{96(n_{i,k}^\star)^{5/2}T_k\sigma^2}{(T_k)^{5/2}(T_k - 2n_{i,k}^\star)\sigma^2}$$
$$- \frac{16(n_{i,k}^\star)^{3/2}T_k(T_k)^2\sigma^2}{(T_k)^{5/2}(T_k - 2n_{i,k}^\star)\sigma^2} + \frac{32(n_{i,k}^\star)^{5/2}T_k\sigma^2}{(T_k)^{5/2}(T_k - 2n_{i,k}^\star)\sigma^2}$$
$$+ \frac{64(n_{i,k}^\star)^{5/2}T_k\sigma^2}{(T_k)^{5/2}(T_k - 2n_{i,k}^\star)\sigma^2} - \frac{128(n_{i,k}^\star)^{7/2}\sigma^2}{(T_k)^{5/2}(T_k - 2n_{i,k}^\star)\sigma^2}$$

The result now follows from

$$(a) + (b) = -\frac{64(n_{i,k}^\star)^{3/2}(c_{i,k}n_{i,k}^\star(T_k)^3 + (2(n_{i,k}^\star)^2 - (T_k)^2)\sigma^2)}{(T_k)^{5/2}(T_k - 2n_{i,k}^\star)\sigma^2}$$

$\square$

**Lemma 15.** *In the context of §H.2 we have that*

$$\frac{\partial p_i}{\partial n_{i,k}}(n_{i,k}^\star) = = -\frac{\sigma^2 D_{i,k}}{64\alpha_{i,k}^3\sqrt{T_k}n_{i,k}^\star}\left(\frac{4\alpha_{i,k}}{\sqrt{T_k}}\left(\frac{4\alpha_{i,k}^2 T_k}{D_{i,k}n_{i,k}^\star} - 1 - c_{i,k}\frac{16\alpha_{i,k}^2 T_k n_{i,k}^\star}{\sigma^2 D_{i,k}}\right)\right.$$
$$\left. - \exp\left(\frac{T_k}{8\alpha_{i,k}^2}\right)\left(\frac{4\alpha_{i,k}^2}{T_k}\left(\frac{T_k}{n_{i,k}^\star} + 1\right) - 1\right)\sqrt{2\pi}Erfc\left(\sqrt{\frac{T_k}{8\alpha_{i,k}^2}}\right)\right)$$

*Proof.* By the dominated convergence theorem we have

$$\frac{\partial p_i}{\partial n_{i,k}}(n_{i,k}^\star) = \mathop{\mathbb{E}}_{x\sim\mathcal{N}(0,1)}\left[-\sigma^2\frac{1 + \frac{D_{i,k}}{\left(1+\alpha_{i,k}^2\left(\frac{1}{n_{i,k}^\star} + \frac{1}{n_{i,k}^\star}\right)x^2\right)^2}\frac{\alpha_{i,k}^2 x^2}{(n_{i,k}^\star)^2}}{\left(\frac{D_{i,k}}{1+\alpha_{i,k}^2\left(\frac{1}{n_{i,k}^\star} + \frac{1}{n_{i,k}^\star}\right)x^2} + n_{i,k}^\star + n_{i,k}^\star\right)^2} + c_{i,k}\right]$$

$$= -\sigma^2 \mathop{\mathbb{E}}_{x\sim\mathcal{N}(0,1)} \left[ \frac{1 + \frac{D_{i,k}}{\left(1+\frac{2\alpha_{i,k}^2}{n_{i,k}^\star}x^2\right)^2} \frac{\alpha_{i,k}^2 x^2}{(n_{i,k}^\star)^2}}{\left(\frac{D_{i,k}}{1+\frac{2\alpha_{i,k}^2}{n_{i,k}^\star}x^2} + 2n_{i,k}^\star\right)^2} \right] + c_{i,k}$$

$$= -\frac{\sigma^2}{(n_{i,k}^\star)^2} \mathop{\mathbb{E}}_{x\sim\mathcal{N}(0,1)} \left[ \frac{1 + \frac{D_{i,k}\alpha_{i,k}^2 x^2}{(n_{i,k}^\star + 2\alpha_{i,k}^2 x^2)^2}}{\left(\frac{D_{i,k}}{n_{i,k}^\star + 2\alpha_{i,k}^2 x^2} + 2\right)^2} \right] + c_{i,k}$$

$$= -\frac{\sigma^2}{4(n_{i,k}^\star)^2} \mathop{\mathbb{E}}_{x\sim\mathcal{N}(0,1)} \left[ \frac{4\left(n_{i,k}^\star + 2\alpha_{i,k}^2 x^2\right)^2 + 4D_{i,k}\alpha_{i,k}^2 x^2}{\left(D_{i,k} + 2\left(n_{i,k}^\star + 2\alpha_{i,k}^2 x^2\right)\right)^2} \right] + c_{i,k}$$

$$= -\frac{\sigma^2}{4(n_{i,k}^\star)^2} \mathop{\mathbb{E}}_{x\sim\mathcal{N}(0,1)} \left[ 1 + \frac{-(D_{i,k})^2 - 4D_{i,k}(n_{i,k}^\star + 2\alpha_{i,k}^2 x^2) + 4D_{i,k}\alpha_{i,k}^2 x^2}{\left(D_{i,k} + 2\left(n_{i,k}^\star + 2\alpha_{i,k}^2 x^2\right)\right)^2} \right] + c_{i,k}$$

$$= -\frac{\sigma^2}{4(n_{i,k}^\star)^2} \mathop{\mathbb{E}}_{x\sim\mathcal{N}(0,1)} \left[ 1 + D_{i,k}\frac{-D_{i,k} - 4(n_{i,k}^\star + 2\alpha_{i,k}^2 x^2) + 4\alpha_{i,k}^2 x^2}{\left(4\alpha_{i,k}^2 x^2 + T_k\right)^2} \right] + c_{i,k}$$

$$= -\frac{\sigma^2}{4(n_{i,k}^\star)^2} \mathop{\mathbb{E}}_{x\sim\mathcal{N}(0,1)} \left[ 1 + D_{i,k}\frac{-T_k - 2n_{i,k}^\star - 4\alpha_{i,k}^2 x^2}{\left(4\alpha_{i,k}^2 x^2 + T_k\right)^2} \right] + c_{i,k}$$

$$= -\frac{\sigma^2}{4(n_{i,k}^\star)^2} + \frac{\sigma^2}{4(n_{i,k}^\star)^2}D_{i,k} \mathop{\mathbb{E}}_{x\sim\mathcal{N}(0,1)} \left[ \frac{1}{\left(4\alpha_{i,k}^2 x^2 + T_k\right)} + \frac{2n_{i,k}^\star}{\left(4\alpha_{i,k}^2 x^2 + T_k\right)^2} \right] + c_{i,k}$$

$$= -\frac{\sigma^2}{4(n_{i,k}^\star)^2} + \frac{\sigma^2}{4(n_{i,k}^\star)^2}D_{i,k} \mathop{\mathbb{E}}_{x\sim\mathcal{N}(0,1)} \left[ \frac{1}{4\alpha_{i,k}^2} \frac{1}{\left(x^2 + \frac{T_k}{4\alpha_{i,k}^2}\right)} + \frac{2n_{i,k}^\star}{16\alpha_{i,k}^4} \frac{1}{\left(x^2 + \frac{T_k}{4\alpha_{i,k}^2}\right)^2} \right] + c_{i,k}$$

$$= c_{i,k} - \frac{\sigma^2}{4(n_{i,k}^\star)^2} + \frac{\sigma^2}{4(n_{i,k}^\star)^2}D_{i,k}\frac{2n_{i,k}^\star}{16\alpha_{i,k}^4}\frac{1}{\frac{T_k}{2\alpha_{i,k}^2}}$$

$$+ \frac{\sigma^2}{4(n_{i,k}^\star)^2}D_{i,k} \left( \frac{1}{4\alpha_{i,k}^2} + \frac{2n_{i,k}^\star}{16\alpha_{i,k}^4}\frac{1 - \frac{T_k}{4\alpha_{i,k}^2}}{\frac{T_k}{2\alpha_{i,k}^2}} \right) \exp\left(\frac{T_k}{8\alpha_{i,k}^2}\right) \mathrm{Erfc}\left(\sqrt{\frac{T_k}{8\alpha_{i,k}^2}}\right) \sqrt{\frac{\pi}{\frac{T_k}{2\alpha_{i,k}^2}}}$$

$$= c_{i,k} - \frac{\sigma^2}{4(n_{i,k}^\star)^2} \left( 1 - \frac{D_{i,k}n_{i,k}^\star}{4\alpha_{i,k}^2 T_k} \right)$$

$$+ \frac{\sigma^2}{4(n_{i,k}^\star)^2}D_{i,k} \left( \frac{1}{4\alpha_{i,k}^2} + \frac{n_{i,k}^\star}{4\alpha_{i,k}^2 T_k} - \frac{n_{i,k}^\star}{16\alpha_{i,k}^4} \right) \exp\left(\frac{T_k}{8\alpha_{i,k}^2}\right) \mathrm{Erfc}\left(\sqrt{\frac{D_{i,k}}{8\alpha_{i,k}^2}}\right) \sqrt{\frac{\pi}{\frac{T_k}{2\alpha_{i,k}^2}}}$$

$$= c_{i,k} - \frac{\sigma^2}{4(n_{i,k}^\star)^2} \left( 1 - \frac{D_{i,k}n_{i,k}^\star}{4\alpha_{i,k}^2 T_k} \right)$$

$$+ \frac{\sigma^2}{4(n_{i,k}^\star)^2}D_{i,k}\frac{\alpha_{i,k}\sqrt{2\pi}}{\sqrt{T_k}}\frac{n_{i,k}^\star}{16\alpha_{i,k}^4} \left( \frac{4\alpha_{i,k}^2}{T_k}\left(\frac{T_k}{n_{i,k}^\star} + 1\right) - 1 \right) \exp\left(\frac{T_k}{8\alpha_{i,k}^2}\right) \mathrm{Erfc}\left(\sqrt{\frac{T_k}{8\alpha_{i,k}^2}}\right)$$

$$= -\frac{\sigma^2 D_{i,k}}{64\alpha_{i,k}^3\sqrt{T_k}n_{i,k}^\star} \left( \frac{4\alpha_{i,k}}{\sqrt{T_k}}\left(\frac{4\alpha_{i,k}^2 T_k}{D_{i,k}n_{i,k}^\star} - 1 - c_{i,k}\frac{16\alpha_{i,k}^2 T_k n_{i,k}^\star}{\sigma^2 D_{i,k}}\right) \right)$$

$$- \exp\left(\frac{T_k}{8\alpha_{i,k}^2}\right)\left(\frac{4\alpha_{i,k}^2}{T_k}\left(\frac{T_k}{n_{i,k}^\star}+1\right)-1\right)\sqrt{2\pi}\mathrm{Erfc}\left(\sqrt{\frac{T_k}{8\alpha_{i,k}^2}}\right)\right)$$

$\square$

**Lemma 16.** *Suppose that* (8) *is satisfied. Let $T_k$ be as defined in Algorithm 1 and $D_{i,k} = T_k - 2n_{i,k}^\star$. If $i \notin V_k$ and $n_{i,k}^\star > 0$ then*

$$p_{i,k}\left(M_{\text{CBL}}, s^\star\right) = \frac{\sigma^2}{2n_{i,k}^\star} - \frac{\sigma^2}{4\alpha_{i,k}^2}\frac{\left(\frac{D_{i,k}}{n_{i,k}^\star}\right)}{2}\exp\left(\frac{T_k}{8\alpha_{i,k}^2}\right)\mathrm{Erfc}\left(\sqrt{\frac{T_k}{8\alpha_{i,k}^2}}\right)\sqrt{\frac{\pi}{\left(\frac{T_k}{2\alpha_{i,k}^2}\right)}} + c_{i,k}n_{i,k}^\star .$$

*Proof.* By the expression for agent $i$'s penalty for distribution $k$ given in the proof of Lemma 3 and the result of Lemma 24, we have that

$$p_{i,k}\left(M_{\text{CBL}}, s^\star\right)$$

$$= \mathop{\mathbb{E}}_{x\sim\mathcal{N}(0,1)}\left[\frac{1}{\frac{|Z_{i,k}'|}{\sigma^2+\alpha_{i,k}^2\left(\frac{\sigma^2}{|X_{i,k}|}+\frac{\sigma^2}{|Z_{i,k}|}\right)x^2}+\frac{|X_{i,k}|+|Z_{i,k}|}{\sigma^2}}\right] + c_{i,k}n_{i,k}^\star$$

$$= \mathop{\mathbb{E}}_{x\sim\mathcal{N}(0,1)}\left[\frac{1}{\frac{D_{i,k}}{\sigma^2+\alpha_{i,k}^2\left(\frac{\sigma^2}{n_{i,k}^\star}+\frac{\sigma^2}{n_{i,k}^\star}\right)x^2}+\frac{n_{i,k}^\star+n_{i,k}^\star}{\sigma^2}}\right] + c_{i,k}n_{i,k}^\star$$

$$= \mathop{\mathbb{E}}_{x\sim\mathcal{N}(0,1)}\left[\frac{1}{\frac{D_{i,k}}{\sigma^2+\alpha_{i,k}^2\left(\frac{2\sigma^2}{n_{i,k}^\star}x^2\right)}+\frac{2n_{i,k}^\star}{\sigma^2}}\right] + c_{i,k}n_{i,k}^\star$$

$$= \frac{\sigma^2}{n_{i,k}^\star}\mathop{\mathbb{E}}_{x\sim\mathcal{N}(0,1)}\left[\frac{1}{\frac{D_{i,k}/n_{i,k}^\star}{1+\frac{2\alpha_{i,k}^2 x^2}{n_{i,k}^\star}}+2}\right] + c_{i,k}n_{i,k}^\star$$

$$= \frac{\sigma^2}{n_{i,k}^\star}\mathop{\mathbb{E}}_{x\sim\mathcal{N}(0,1)}\left[\frac{1}{\frac{D_{i,k}/n_{i,k}^\star+2+\frac{4\alpha_{i,k}^2 x^2}{n_{i,k}^\star}}{1+\frac{2\alpha_{i,k}^2 x^2}{n_{i,k}^\star}}}\right] + c_{i,k}n_{i,k}^\star$$

$$= \frac{\sigma^2}{n_{i,k}^\star}\mathop{\mathbb{E}}_{x\sim\mathcal{N}(0,1)}\left[\frac{1+\frac{2\alpha_{i,k}^2 x^2}{n_{i,k}^\star}}{D_{i,k}/n_{i,k}^\star+2+\frac{4\alpha_{i,k}^2 x^2}{n_{i,k}^\star}}\right] + c_{i,k}n_{i,k}^\star$$

$$= \frac{\sigma^2}{n_{i,k}^\star}\mathop{\mathbb{E}}_{x\sim\mathcal{N}(0,1)}\left[\frac{1}{2}-\frac{\left(\frac{D_{i,k}}{n_{i,k}^\star}\right)}{2}\frac{1}{D_{i,k}/n_{i,k}^\star+2+\frac{4\alpha_{i,k}^2 x^2}{n_{i,k}^\star}}\right] + c_{i,k}n_{i,k}^\star$$

$$= \frac{\sigma^2}{2n_{i,k}^\star}-\frac{\sigma^2}{n_{i,k}^\star}\frac{D_{i,k}/n_{i,k}^\star}{2}\mathop{\mathbb{E}}_{x\sim\mathcal{N}(0,1)}\left[\frac{1}{D_{i,k}/n_{i,k}^\star+2+\frac{4\alpha_{i,k}^2 x^2}{n_{i,k}^\star}}\right] + c_{i,k}n_{i,k}^\star$$

$$
= \frac{\sigma^2}{2n^\star_{i,k}} - \frac{\sigma^2}{n^\star_{i,k}} \frac{D_{i,k}/n^\star_{i,k}}{2} \frac{1}{\left(\frac{4\alpha^2_{i,k}}{n^\star_{i,k}}\right)} \mathop{\mathbb{E}}_{x \sim \mathcal{N}(0,1)} \left[ \frac{1}{\frac{D_{i,k}/n^\star_{i,k}+2}{\left(\frac{4\alpha^2_{i,k}}{n^\star_{i,k}}\right)} + x^2} \right] + c_{i,k} n^\star_{i,k}
$$

$$
= \frac{\sigma^2}{2n^\star_{i,k}} + c_{i,k} n^\star_{i,k}
$$

$$
- \frac{\sigma^2}{4\alpha^2_{i,k}} \frac{D_{i,k}/n^\star_{i,k}}{2} \exp\left( \frac{D_{i,k}/n^\star_{i,k}+2}{2\left(\frac{4\alpha^2_{i,k}}{n^\star_{i,k}}\right)} \right) \mathrm{Erfc}\left( \sqrt{\frac{D_{i,k}/n^\star_{i,k}+2}{2\left(\frac{4\alpha^2_{i,k}}{n^\star_{i,k}}\right)}} \right) \sqrt{\frac{\pi}{2\left(\frac{D_{i,k}/n^\star_{i,k}+2}{\left(\frac{4\alpha^2_{i,k}}{n^\star_{i,k}}\right)}\right)}}
$$

$$
= \frac{\sigma^2}{2n^\star_{i,k}} - \frac{\sigma^2}{4\alpha^2_{i,k}} \frac{D_{i,k}/n^\star_{i,k}}{2} \exp\left( \frac{T_k}{8\alpha^2_{i,k}} \right) \mathrm{Erfc}\left( \sqrt{\frac{T_k}{8\alpha^2_{i,k}}} \right) \sqrt{\frac{\pi}{\frac{T_k}{2\alpha^2_{i,k}}}} + c_{i,k} n^\star_{i,k} \; .
$$

$\square$

**Lemma 17.** *Suppose that* (8) *is satisfied. Let $T_k$ be as defined in Algorithm 1. If $i \notin V_k$ and $n^\star_{i,k} > 0$ then*

$$
p_{i,k}\left(M_{\text{CBL}}, s^\star\right) = \frac{2c_{i,k}(n^\star_{i,k})^2 + \sigma^2 + \frac{\alpha_{i,k}(16c_{i,k}(n^\star_{i,k})^2 T_k \alpha^2_{i,k} + (n^\star_{i,k}(-2n^\star_{i,k}+T_k)-4T_k\alpha^2_{i,k})\sigma^2)}{-n^\star_{i,k} T_k \alpha_{i,k} + 4(n^\star_{i,k}+T_k)\alpha^3_{i,k}}}{2n^\star_{i,k}}
$$

*Proof.* Let $D_{i,k} = T_k - 2n^\star_{i,k}$. We have already shown that $G_{i,k}(\alpha_{i,k}) = 0$ has a solution. Therefore

$$
\exp\left( \frac{T_k}{8\alpha^2_{i,k}} \right) \mathrm{Erfc}\left( \sqrt{\frac{T_k}{8\alpha^2_{i,k}}} \right) = \frac{2\sqrt{\frac{2}{\pi}}\alpha_{i,k}\left(-1 + \frac{4T_k\alpha^2_{i,k}}{D_{i,k}n^\star_{i,k}} - \frac{16c_{i,k}n^\star_{i,k}T_k\alpha^2_{i,k}}{D_{i,k}\sigma^2}\right)}{\sqrt{T_k}\left(-1 + \frac{4\left(1+\frac{T_k}{n^\star_{i,k}}\right)\alpha^2_{i,k}}{T_k}\right)}
$$

Substituting this expression into Lemma 16 gives us

$$
p_{i,k}\left(M_{\text{CBL}}, s^\star\right)
$$

$$
= c_{i,k}n^\star_{i,k} + \frac{\sigma^2}{2n^\star_{i,k}} - \frac{D_{i,k}\sqrt{\frac{\alpha^2_{i,k}}{T_k}}\left(-1 + \frac{4T_k\alpha^2_{i,k}}{D_{i,k}n^\star_{i,k}} - \frac{16c_{i,k}n^\star_{i,k}T_k\alpha^2_{i,k}}{D_{i,k}\sigma^2}\right)\sigma^2}{2n^\star_{i,k}\sqrt{T_k}\alpha_{i,k}\left(-1 + \frac{4\left(1+\frac{T_k}{n^\star_{i,k}}\right)\alpha^2_{i,k}}{T_k}\right)}
$$

$$
= \frac{1}{2n^\star_{i,k}}\left( 2c_{i,k}(n^\star_{i,k})^2 + \sigma^2 - \underbrace{\frac{D_{i,k}\sqrt{\frac{\alpha^2_{i,k}}{T_k}}\left(-1 + \frac{4T_k\alpha^2_{i,k}}{D_{i,k}n^\star_{i,k}} - \frac{16c_{i,k}n^\star_{i,k}T_k\alpha^2_{i,k}}{D_{i,k}\sigma^2}\right)\sigma^2}{\sqrt{T_k}\alpha_{i,k}\left(-1 + \frac{4\left(1+\frac{T_k}{n^\star_{i,k}}\right)\alpha^2_{i,k}}{T_k}\right)}}_{(*)} \right)
$$

where we have the following expression for $(*)$

$$
(*) = \frac{-\sqrt{\frac{\alpha^2_{i,k}}{T_k}}\left(-D_{i,k}\sigma^2 + \frac{4T_k\alpha^2_{i,k}\sigma^2}{n^\star_{i,k}} - 16c_{i,k}n^\star_{i,k}T_k\alpha^2_{i,k}\right)}{-\sqrt{T_k}\alpha_{i,k} + \frac{4\left(1+\frac{T_k}{n^\star_{i,k}}\right)\alpha^3_{i,k}}{\sqrt{T_k}}}
$$

$$
= \frac{-n_{i,k}^{\star}\sqrt{\frac{\alpha_{i,k}^2}{T_k}}\left(-D_{i,k}\sigma^2 + \frac{4T_k\alpha_{i,k}^2\sigma^2}{n_{i,k}^{\star}} - 16c_{i,k}n_{i,k}^{\star}T_k\alpha_{i,k}^2\right)}{\frac{1}{\sqrt{T_k}}\left(-T_k\alpha_{i,k}n_{i,k}^{\star} + 4(n_{i,k}^{\star}+T_k)\alpha_{i,k}^3\right)}
$$

$$
= \frac{-\alpha_{i,k}\left(-D_{i,k}\sigma^2 n_{i,k}^{\star} + 4T_k\alpha_{i,k}^2\sigma^2 - 16c_{i,k}(n_{i,k}^{\star})^2T_k\alpha_{i,k}^2\right)}{-T_k\alpha_{i,k}n_{i,k}^{\star} + 4(n_{i,k}^{\star}+T_k)\alpha_{i,k}^3}
$$

$$
= \frac{\alpha_{i,k}\left(16c_{i,k}(n_{i,k}^{\star})^2T_k\alpha_{i,k}^2 + (D_{i,k}n_{i,k}^{\star} - 4T_k\alpha_{i,k}^2)\sigma^2\right)}{-n_{i,k}^{\star}T_k\alpha_{i,k} + 4(n_{i,k}^{\star}+T_k)\alpha_{i,k}^3}
$$

$$
= \frac{\alpha_{i,k}\left(16c_{i,k}(n_{i,k}^{\star})^2T_k\alpha_{i,k}^2 + (n_{i,k}^{\star}(-2n_{i,k}^{\star}+T_k) - 4T_k\alpha_{i,k}^2)\sigma^2\right)}{-n_{i,k}^{\star}T_k\alpha_{i,k} + 4(n_{i,k}^{\star}+T_k)\alpha_{i,k}^3}
$$

Substituting in this expression for $(*)$ gives the desired result. $\qquad\square$

**Lemma 18.** *The following two expressions are equal*

$$
\frac{T_k}{n_{i,k}^{\star}(2c_{i,k}n_{i,k}^{\star}T_k + \sigma^2)}\left(2c_{i,k}(n_{i,k}^{\star})^2 + \sigma^2 + \frac{\alpha_{i,k}\left(16c_{i,k}(n_{i,k}^{\star})^2T_k\alpha_{i,k}^2 + (n_{i,k}^{\star}(T_k - 2n_{i,k}^{\star}) - 4T_k\alpha_{i,k}^2)\sigma^2\right)}{-n_{i,k}^{\star}T_k\alpha_{i,k} + 4(n_{i,k}^{\star}+T_k)\alpha_{i,k}^3}\right)
$$

$$
= \frac{T_k(2c_{i,k}(n_{i,k}^{\star})^2 + 6c_{i,k}n_{i,k}^{\star}T_k + \sigma^2)}{(n_{i,k}^{\star}+T_k)(2c_{i,k}n_{i,k}^{\star}T_k + \sigma^2)} + \frac{4c_{i,k}(n_{i,k}^{\star})^2(T_k)^3 - 2(n_{i,k}^{\star})^2T_k\sigma^2 - n_{i,k}^{\star}(T_k)^2\sigma^2}{(n_{i,k}^{\star}+T_k)(-n_{i,k}^{\star}T_k + 4n_{i,k}^{\star}\alpha_{i,k}^2 + 4T_k\alpha_{i,k}^2)(2c_{i,k}n_{i,k}^{\star}T_k + \sigma^2)} \ .
$$

*Proof.* Via a sequence of algebraic manipulations, we have that

$$
\frac{T_k}{n_{i,k}^{\star}(2c_{i,k}n_{i,k}^{\star}T_k + \sigma^2)}\left(2c_{i,k}(n_{i,k}^{\star})^2 + \sigma^2 + \frac{\alpha_{i,k}\left(16c_{i,k}(n_{i,k}^{\star})^2T_k\alpha_{i,k}^2 + (n_{i,k}^{\star}(T_k - 2n_{i,k}^{\star}) - 4T_k\alpha_{i,k}^2)\sigma^2\right)}{-n_{i,k}^{\star}T_k\alpha_{i,k} + 4(n_{i,k}^{\star}+T_k)\alpha_{i,k}^3}\right)
$$

$$
= \frac{T_k\left(2c_{i,k}(n_{i,k}^{\star})^2 + \sigma^2\right)}{n_{i,k}^{\star}\left(2c_{i,k}n_{i,k}^{\star}T_k + \sigma^2\right)} + \frac{T_k\left(16c_{i,k}(n_{i,k}^{\star})^2T_k\alpha_{i,k}^2 + \left(n_{i,k}^{\star}\left(T_k - 2n_{i,k}^{\star}\right) - 4T_k\alpha_{i,k}^2\right)\sigma^2\right)}{n_{i,k}^{\star}\left(2c_{i,k}n_{i,k}^{\star}T_k + \sigma^2\right)\left(-n_{i,k}^{\star}T_k + 4\left(n_{i,k}^{\star}+T_k\right)\alpha_{i,k}^2\right)}
$$

$$
= \frac{T_k\left(\frac{n_{i,k}^{\star}+T_k}{n_{i,k}^{\star}}\right)\left(2c_{i,k}(n_{i,k}^{\star})^2 + \sigma^2\right)}{\left(n_{i,k}^{\star}+T_k\right)\left(2c_{i,k}n_{i,k}^{\star}T_k + \sigma^2\right)}
$$
$$
+ \frac{T_k\left(\frac{n_{i,k}^{\star}+T_k}{n_{i,k}^{\star}}\right)\left(16c_{i,k}(n_{i,k}^{\star})^2T_k\alpha_{i,k}^2 + \left(n_{i,k}^{\star}\left(T_k - 2n_{i,k}^{\star}\right) - 4T_k\alpha_{i,k}^2\right)\sigma^2\right)}{\left(n_{i,k}^{\star}+T_k\right)\left(-n_{i,k}^{\star}T_k + 4n_{i,k}^{\star}\alpha_{i,k}^2 + 4T_k\alpha_{i,k}^2\right)\left(2c_{i,k}n_{i,k}^{\star}T_k + \sigma^2\right)}
$$

$$
= \frac{T_k\left(2c_{i,k}(n_{i,k}^{\star})^2 + \sigma^2\right)}{\left(n_{i,k}^{\star}+T_k\right)\left(2c_{i,k}n_{i,k}^{\star}T_k + \sigma^2\right)} + \frac{\frac{T_k^2}{n_{i,k}^{\star}}\left(2c_{i,k}(n_{i,k}^{\star})^2 + \sigma^2\right)}{\left(n_{i,k}^{\star}+T_k\right)\left(2c_{i,k}n_{i,k}^{\star}T_k + \sigma^2\right)}
$$
$$
+ \frac{T_k\left(1 + \frac{T_k}{n_{i,k}^{\star}}\right)\left(16c_{i,k}(n_{i,k}^{\star})^2T_k\alpha_{i,k}^2 + \left(n_{i,k}^{\star}\left(T_k - 2n_{i,k}^{\star}\right) - 4T_k\alpha_{i,k}^2\right)\sigma^2\right)}{\left(n_{i,k}^{\star}+T_k\right)\left(-n_{i,k}^{\star}T_k + 4n_{i,k}^{\star}\alpha_{i,k}^2 + 4T_k\alpha_{i,k}^2\right)\left(2c_{i,k}n_{i,k}^{\star}T_k + \sigma^2\right)}
$$

$$
= \frac{T_k\left(2c_{i,k}(n_{i,k}^{\star})^2 + 6c_{i,k}n_{i,k}^{\star}T_k + \sigma^2\right)}{\left(n_{i,k}^{\star}+T_k\right)\left(2c_{i,k}n_{i,k}^{\star}T_k + \sigma^2\right)} + \frac{-6c_{i,k}n_{i,k}^{\star}T_k^2 + \frac{T_k^2}{n_{i,k}^{\star}}\left(2c_{i,k}(n_{i,k}^{\star})^2 + \sigma^2\right)}{\left(n_{i,k}^{\star}+T_k\right)\left(2c_{i,k}n_{i,k}^{\star}T_k + \sigma^2\right)} \tag{33}
$$

$$
+ \frac{T_k\left(1 + \frac{T_k}{n_{i,k}^{\star}}\right)\left(16c_{i,k}(n_{i,k}^{\star})^2T_k\alpha_{i,k}^2 + \left(n_{i,k}^{\star}\left(T_k - 2n_{i,k}^{\star}\right) - 4T_k\alpha_{i,k}^2\right)\sigma^2\right)}{\left(n_{i,k}^{\star}+T_k\right)\left(-n_{i,k}^{\star}T_k + 4n_{i,k}^{\star}\alpha_{i,k}^2 + 4T_k\alpha_{i,k}^2\right)\left(2c_{i,k}n_{i,k}^{\star}T_k + \sigma^2\right)} \tag{34}
$$

Notice that we can rewrite line (34) as

$$
\frac{T_k\left(1+\frac{T_k}{n_{i,k}^\star}\right)\left(16c_{i,k}(n_{i,k}^\star)^2 T_k\alpha_{i,k}^2+\left(n_{i,k}^\star\left(T_k-2n_{i,k}^\star\right)-4T_k\alpha_{i,k}^2\right)\sigma^2\right)}{\left(n_{i,k}^\star+T_k\right)\left(-n_{i,k}^\star T_k+4n_{i,k}^\star\alpha_{i,k}^2+4T_k\alpha_{i,k}^2\right)\left(2c_{i,k}n_{i,k}^\star T_k+\sigma^2\right)}
$$

$$
=\frac{\left(1+\frac{T_k}{n_{i,k}^\star}\right)\left(16c_{i,k}(n_{i,k}^\star)^2 T_k\alpha_{i,k}^2-4T_k^2\alpha_{i,k}^2\sigma^2\right)}{\left(n_{i,k}^\star+T_k\right)\left(-n_{i,k}^\star T_k+4n_{i,k}^\star\alpha_{i,k}^2+4T_k\alpha_{i,k}^2\right)\left(2c_{i,k}n_{i,k}^\star T_k+\sigma^2\right)}
$$

$$
+\frac{\left(1+\frac{T_k}{n_{i,k}^\star}\right)\left(T_k n_{i,k}^\star\left(T_k-2n_{i,k}^\star\right)\sigma^2\right)}{\left(n_{i,k}^\star+T_k\right)\left(-n_{i,k}^\star T_k+4n_{i,k}^\star\alpha_{i,k}^2+4T_k\alpha_{i,k}^2\right)\left(2c_{i,k}n_{i,k}^\star T_k+\sigma^2\right)}
$$

$$
=\frac{\left(1+\frac{T_k}{n_{i,k}^\star}\right)\left(16c_{i,k}(n_{i,k}^\star)^2 T_k\alpha_{i,k}^2-4T_k^2\alpha_{i,k}^2\sigma^2\right)}{\left(n_{i,k}^\star+T_k\right)\left(-n_{i,k}^\star T_k+4n_{i,k}^\star\alpha_{i,k}^2+4T_k\alpha_{i,k}^2\right)\left(2c_{i,k}n_{i,k}^\star T_k+\sigma^2\right)}
$$

$$
+\frac{n_{i,k}^\star T_k^2\sigma^2-2(n_{i,k}^\star)^2 T_k\sigma^2+T_k^2\left(T_k-2n_{i,k}^\star\right)\sigma^2}{\left(n_{i,k}^\star+T_k\right)\left(-n_{i,k}^\star T_k+4n_{i,k}^\star\alpha_{i,k}^2+4T_k\alpha_{i,k}^2\right)\left(2c_{i,k}n_{i,k}^\star T_k+\sigma^2\right)}
$$

$$
=\frac{\left(1+\frac{T_k}{n_{i,k}^\star}\right)\left(16c_{i,k}(n_{i,k}^\star)^2 T_k\alpha_{i,k}^2-4T_k^2\alpha_{i,k}^2\sigma^2\right)}{\left(n_{i,k}^\star+T_k\right)\left(-n_{i,k}^\star T_k+4n_{i,k}^\star\alpha_{i,k}^2+4T_k\alpha_{i,k}^2\right)\left(2c_{i,k}n_{i,k}^\star T_k+\sigma^2\right)}
$$

$$
+\frac{4c_{i,k}(n_{i,k}^\star)^2 T_k^3-2(n_{i,k}^\star)^2 T_k\sigma^2-n_{i,k}^\star T_k\sigma^2}{\left(n_{i,k}^\star+T_k\right)\left(-n_{i,k}^\star T_k+4n_{i,k}^\star\alpha_{i,k}^2+4T_k\alpha_{i,k}^2\right)\left(2c_{i,k}n_{i,k}^\star T_k+\sigma^2\right)}
$$

$$
+\frac{-4c_{i,k}(n_{i,k}^\star)^2 T_k^3+T_k^3\sigma^2}{\left(n_{i,k}^\star+T_k\right)\left(-n_{i,k}^\star T_k+4n_{i,k}^\star\alpha_{i,k}^2+4T_k\alpha_{i,k}^2\right)\left(2c_{i,k}n_{i,k}^\star T_k+\sigma^2\right)}
$$

Substituting in this expression, we have that lines (33) and (34) become

$$
\frac{T_k\left(2c_{i,k}(n_{i,k}^\star)^2+6c_{i,k}n_{i,k}^\star T_k+\sigma^2\right)}{\left(n_{i,k}^\star+T_k\right)\left(2c_{i,k}n_{i,k}^\star T_k+\sigma^2\right)}
$$

$$
+\frac{4c_{i,k}(n_{i,k}^\star)^2 T_k^3-2(n_{i,k}^\star)^2 T_k\sigma^2-n_{i,k}^\star T_k\sigma^2}{\left(n_{i,k}^\star+T_k\right)\left(-n_{i,k}^\star T_k+4n_{i,k}^\star\alpha_{i,k}^2+4T_k\alpha_{i,k}^2\right)\left(2c_{i,k}n_{i,k}^\star T_k+\sigma^2\right)}
$$

$$
+\frac{-6c_{i,k}n_{i,k}^\star T_k^2+\frac{T_k^2}{n_{i,k}^\star}\left(2c_{i,k}(n_{i,k}^\star)^2+\sigma^2\right)}{\left(n_{i,k}^\star+T_k\right)\left(2c_{i,k}n_{i,k}^\star T_k+\sigma^2\right)} \tag{35}
$$

$$
+\frac{\left(1+\frac{T_k}{n_{i,k}^\star}\right)\left(16c_{i,k}(n_{i,k}^\star)^2 T_k\alpha_{i,k}^2-4T_k^2\alpha_{i,k}^2\sigma^2\right)}{\left(n_{i,k}^\star+T_k\right)\left(-n_{i,k}^\star T_k+4n_{i,k}^\star\alpha_{i,k}^2+4T_k\alpha_{i,k}^2\right)\left(2c_{i,k}n_{i,k}^\star T_k+\sigma^2\right)} \tag{36}
$$

$$
+\frac{-4c_{i,k}(n_{i,k}^\star)^2 T_k^3+T_k^3\sigma^2}{\left(n_{i,k}^\star+T_k\right)\left(-n_{i,k}^\star T_k+4n_{i,k}^\star\alpha_{i,k}^2+4T_k\alpha_{i,k}^2\right)\left(2c_{i,k}n_{i,k}^\star T_k+\sigma^2\right)} \tag{37}
$$

Now notice that if we can show lines (35), (36), and (37) sum to 0 then we are done with the proof. We can rewrite the sum of these three lines as

$$
\frac{\left(-4c_{i,k}n_{i,k}^\star T_k^2+\frac{T_k^2\sigma^2}{n_{i,k}^\star}\right)\left(-n_{i,k}^\star T_k+4n_{i,k}^\star\alpha_{i,k}^2+4T_k\alpha_{i,k}^2\right)}{\left(n_{i,k}^\star+T_k\right)\left(-n_{i,k}^\star T_k+4n_{i,k}^\star\alpha_{i,k}^2+4T_k\alpha_{i,k}^2\right)\left(2c_{i,k}n_{i,k}^\star T_k+\sigma^2\right)}
$$

$$+ \frac{\left(1 + \frac{T_k}{n_{i,k}^\star}\right)\left(16c_{i,k}(n_{i,k}^\star)^2 T_k \alpha_{i,k}^2 - 4T_k^2 \alpha_{i,k}^2 \sigma^2\right)}{\left(n_{i,k}^\star + T_k\right)\left(-n_{i,k}^\star T_k + 4n_{i,k}^\star \alpha_{i,k}^2 + 4T_k \alpha_{i,k}^2\right)\left(2c_{i,k}n_{i,k}^\star T_k + \sigma^2\right)}$$

$$+ \frac{-4c_{i,k}(n_{i,k}^\star)^2 T_k^3 + T_k^3 \sigma^2}{\left(n_{i,k}^\star + T_k\right)\left(-n_{i,k}^\star T_k + 4n_{i,k}^\star \alpha_{i,k}^2 + 4T_k \alpha_{i,k}^2\right)\left(2c_{i,k}n_{i,k}^\star T_k + \sigma^2\right)}$$

$$= \left((n_{i,k}^\star + T_k)\left(-n_{i,k}^\star T_k + 4n_{i,k}^\star \alpha_{i,k}^2 + 4T_k \alpha_{i,k}^2\right)\left(2c_{i,k}n_{i,k}^\star T_k + \sigma^2\right)\right)^{-1}$$

$$\left(\left(-4c_{i,k}n_{i,k}^\star T_k^2 + \frac{T_k^2 \sigma^2}{n_{i,k}^\star}\right)\left(-n_{i,k}^\star T_k + 4n_{i,k}^\star \alpha_{i,k}^2 + 4T_k \alpha_{i,k}^2\right)\right. \tag{38}$$

$$\left. + \left(1 + \frac{T_k}{n_{i,k}^\star}\right)\left(16c_{i,k}(n_{i,k}^\star)^2 T_k \alpha_{i,k}^2 - 4T_k^2 \alpha_{i,k}^2 \sigma^2\right) - 4c_{i,k}(n_{i,k}^\star)^2 T_k^3 + T_k^3 \sigma^2\right) \tag{39}$$

We will conclude by showing that lines (38) and (39) sum to 0 which will complete the proof.

$$\left(-4c_{i,k}n_{i,k}^\star T_k^2 + \frac{T_k^2 \sigma^2}{n_{i,k}^\star}\right)\left(-n_{i,k}^\star T_k + 4n_{i,k}^\star \alpha_{i,k}^2 + 4T_k \alpha_{i,k}^2\right)$$

$$+ \left(1 + \frac{T_k}{n_{i,k}^\star}\right)\left(16c_{i,k}(n_{i,k}^\star)^2 T_k \alpha_{i,k}^2 - 4T_k^2 \alpha_{i,k}^2 \sigma^2\right) - 4c_{i,k}(n_{i,k}^\star)^2 T_k^3 + T_k^3 \sigma^2$$

$$= -c_{i,k}(n_{i,k}^\star)^2 T_k^3 - 16c_{i,k}(n_{i,k}^\star)^2 T_k^2 \alpha_{i,k}^2 - 16c_{i,k}n_{i,k}^\star T_k^3 \alpha_{i,k}^2$$

$$- T_k^3 \sigma^2 + 4T_k^2 \sigma^2 \alpha_{i,k}^2 + \frac{4T_k^3 \sigma^2 \alpha_{i,k}^2}{n_{i,k}^\star}$$

$$+ 16c_{i,k}(n_{i,k}^\star)^2 T_k^2 \alpha_{i,k}^2 - 4T_k^2 \alpha_{i,k}^2 \sigma^2$$

$$+ 16c_{i,k}n_{i,k}^\star T_k^3 \alpha_{i,k}^2 - \frac{4T_k^3 \alpha_{i,k}^2 \sigma^2}{n_{i,k}^\star}$$

$$- 4c_{i,k}(n_{i,k}^\star)^2 T_k^3 + T_k^3 \sigma^2$$

$$= 0$$

$\square$

**Lemma 19.** *The following two expressions are equal*

$$\frac{T_k}{n_{i,k}^\star \sigma^2}\left(2c_{i,k}(n_{i,k}^\star)^2 + \sigma^2 + \frac{\alpha_{i,k}\left(16c_{i,k}(n_{i,k}^\star)^2 T_k \alpha_{i,k}^2 + (n_{i,k}^\star(T_k - 2n_{i,k}^\star) - 4T_k \alpha_{i,k}^2)\sigma^2\right)}{-n_{i,k}^\star T_k \alpha_{i,k} + 4(n_{i,k}^\star + T_k)\alpha_{i,k}^3}\right)$$

$$= \frac{T_k(2c_{i,k}(n_{i,k}^\star)^2 + 6c_{i,k}n_{i,k}^\star T_k + \sigma^2)}{(n_{i,k}^\star + T_k)\sigma^2} + \frac{4c_{i,k}(n_{i,k}^\star)^2(T_k)^3 - 2(n_{i,k}^\star)^2 T_k \sigma^2 - n_{i,k}^\star(T_k)^2 \sigma^2}{(n_{i,k}^\star + T_k)(-n_{i,k}^\star T_k + 4n_{i,k}^\star \alpha_{i,k}^2 + 4T_k \alpha_{i,k}^2)\sigma^2} \; .$$

*Proof.* Via a sequence of algebraic manipulations, we have that

$$\frac{T_k}{n_{i,k}^\star \sigma^2}\left(2c_{i,k}(n_{i,k}^\star)^2 + \sigma^2 + \frac{\alpha_{i,k}\left(16c_{i,k}(n_{i,k}^\star)^2 T_k \alpha_{i,k}^2 + (n_{i,k}^\star(T_k - 2n_{i,k}^\star) - 4T_k \alpha_{i,k}^2)\sigma^2\right)}{-n_{i,k}^\star T_k \alpha_{i,k} + 4(n_{i,k}^\star + T_k)\alpha_{i,k}^3}\right)$$

$$= \frac{T_k\left(2c_{i,k}(n_{i,k}^\star)^2 + \sigma^2\right)}{n_{i,k}^\star \sigma^2} + \frac{T_k\left(16c_{i,k}(n_{i,k}^\star)^2 T_k \alpha_{i,k}^2 + \left(n_{i,k}^\star\left(T_k - 2n_{i,k}^\star\right) - 4T_k \alpha_{i,k}^2\right)\sigma^2\right)}{n_{i,k}^\star\left(-n_{i,k}^\star T_k + 4\left(n_{i,k}^\star + T_k\right)\alpha_{i,k}^2\right)\sigma^2}$$

$$= \frac{T_k \left(\frac{n_{i,k}^\star + T_k}{n_{i,k}^\star}\right) \left(2c_{i,k}(n_{i,k}^\star)^2 + \sigma^2\right)}{\left(n_{i,k}^\star + T_k\right)\sigma^2}$$

$$+ \frac{T_k \left(\frac{n_{i,k}^\star + T_k}{n_{i,k}^\star}\right) \left(16c_{i,k}(n_{i,k}^\star)^2 T_k \alpha_{i,k}^2 + \left(n_{i,k}^\star \left(T_k - 2n_{i,k}^\star\right) - 4T_k\alpha_{i,k}^2\right)\sigma^2\right)}{\left(n_{i,k}^\star + T_k\right)\left(-n_{i,k}^\star T_k + 4n_{i,k}^\star \alpha_{i,k}^2 + 4T_k\alpha_{i,k}^2\right)\sigma^2}$$

$$= \frac{T_k \left(2c_{i,k}(n_{i,k}^\star)^2 + \sigma^2\right)}{\left(n_{i,k}^\star + T_k\right)\sigma^2} + \frac{\frac{T_k^2}{n_{i,k}^\star}\left(2c_{i,k}(n_{i,k}^\star)^2 + \sigma^2\right)}{\left(n_{i,k}^\star + T_k\right)\sigma^2}$$

$$+ \frac{T_k \left(1 + \frac{T_k}{n_{i,k}^\star}\right)\left(16c_{i,k}(n_{i,k}^\star)^2 T_k \alpha_{i,k}^2 + \left(n_{i,k}^\star\left(T_k - 2n_{i,k}^\star\right) - 4T_k\alpha_{i,k}^2\right)\sigma^2\right)}{\left(n_{i,k}^\star + T_k\right)\left(-n_{i,k}^\star T_k + 4n_{i,k}^\star \alpha_{i,k}^2 + 4T_k\alpha_{i,k}^2\right)\sigma^2}$$

$$= \frac{T_k \left(2c_{i,k}(n_{i,k}^\star)^2 + 6c_{i,k}n_{i,k}^\star T_k + \sigma^2\right)}{\left(n_{i,k}^\star + T_k\right)\sigma^2} + \frac{-6c_{i,k}n_{i,k}^\star T_k^2 + \frac{T_k^2}{n_{i,k}^\star}\left(2c_{i,k}(n_{i,k}^\star)^2 + \sigma^2\right)}{\left(n_{i,k}^\star + T_k\right)\sigma^2} \tag{40}$$

$$+ \frac{T_k \left(1 + \frac{T_k}{n_{i,k}^\star}\right)\left(16c_{i,k}(n_{i,k}^\star)^2 T_k \alpha_{i,k}^2 + \left(n_{i,k}^\star\left(T_k - 2n_{i,k}^\star\right) - 4T_k\alpha_{i,k}^2\right)\sigma^2\right)}{\left(n_{i,k}^\star + T_k\right)\left(-n_{i,k}^\star T_k + 4n_{i,k}^\star \alpha_{i,k}^2 + 4T_k\alpha_{i,k}^2\right)\sigma^2} \tag{41}$$

Notice that we can rewrite line (41) as

$$\frac{T_k \left(1 + \frac{T_k}{n_{i,k}^\star}\right)\left(16c_{i,k}(n_{i,k}^\star)^2 T_k \alpha_{i,k}^2 + \left(n_{i,k}^\star\left(T_k - 2n_{i,k}^\star\right) - 4T_k\alpha_{i,k}^2\right)\sigma^2\right)}{\left(n_{i,k}^\star + T_k\right)\left(-n_{i,k}^\star T_k + 4n_{i,k}^\star \alpha_{i,k}^2 + 4T_k\alpha_{i,k}^2\right)\sigma^2}$$

$$= \frac{\left(1 + \frac{T_k}{n_{i,k}^\star}\right)\left(16c_{i,k}(n_{i,k}^\star)^2 T_k \alpha_{i,k}^2 - 4T_k^2\alpha_{i,k}^2\sigma^2\right)}{\left(n_{i,k}^\star + T_k\right)\left(-n_{i,k}^\star T_k + 4n_{i,k}^\star \alpha_{i,k}^2 + 4T_k\alpha_{i,k}^2\right)\sigma^2} + \frac{\left(1 + \frac{T_k}{n_{i,k}^\star}\right)\left(T_k n_{i,k}^\star\left(T_k - 2n_{i,k}^\star\right)\sigma^2\right)}{\left(n_{i,k}^\star + T_k\right)\left(-n_{i,k}^\star T_k + 4n_{i,k}^\star \alpha_{i,k}^2 + 4T_k\alpha_{i,k}^2\right)\sigma^2}$$

$$= \frac{\left(1 + \frac{T_k}{n_{i,k}^\star}\right)\left(16c_{i,k}(n_{i,k}^\star)^2 T_k \alpha_{i,k}^2 - 4T_k^2\alpha_{i,k}^2\sigma^2\right)}{\left(n_{i,k}^\star + T_k\right)\left(-n_{i,k}^\star T_k + 4n_{i,k}^\star \alpha_{i,k}^2 + 4T_k\alpha_{i,k}^2\right)\sigma^2} + \frac{n_{i,k}^\star T_k^2\sigma^2 - 2(n_{i,k}^\star)^2 T_k\sigma^2 + T_k^2\left(T_k - 2n_{i,k}^\star\right)\sigma^2}{\left(n_{i,k}^\star + T_k\right)\left(-n_{i,k}^\star T_k + 4n_{i,k}^\star \alpha_{i,k}^2 + 4T_k\alpha_{i,k}^2\right)\sigma^2}$$

$$= \frac{\left(1 + \frac{T_k}{n_{i,k}^\star}\right)\left(16c_{i,k}(n_{i,k}^\star)^2 T_k \alpha_{i,k}^2 - 4T_k^2\alpha_{i,k}^2\sigma^2\right)}{\left(n_{i,k}^\star + T_k\right)\left(-n_{i,k}^\star T_k + 4n_{i,k}^\star \alpha_{i,k}^2 + 4T_k\alpha_{i,k}^2\right)\sigma^2} + \frac{4c_{i,k}(n_{i,k}^\star)^2 T_k^3 - 2(n_{i,k}^\star)^2 T_k\sigma^2 - n_{i,k}^\star T_k\sigma^2}{\left(n_{i,k}^\star + T_k\right)\left(-n_{i,k}^\star T_k + 4n_{i,k}^\star \alpha_{i,k}^2 + 4T_k\alpha_{i,k}^2\right)\sigma^2}$$

$$+ \frac{-4c_{i,k}(n_{i,k}^\star)^2 T_k^3 + T_k^3\sigma^2}{\left(n_{i,k}^\star + T_k\right)\left(-n_{i,k}^\star T_k + 4n_{i,k}^\star \alpha_{i,k}^2 + 4T_k\alpha_{i,k}^2\right)\sigma^2}$$

Substituting in this expression, we have that lines (40) and (41) become

$$\frac{T_k \left(2c_{i,k}(n_{i,k}^\star)^2 + 6c_{i,k}n_{i,k}^\star T_k + \sigma^2\right)}{\left(n_{i,k}^\star + T_k\right)\sigma^2} + \frac{4c_{i,k}(n_{i,k}^\star)^2 T_k^3 - 2(n_{i,k}^\star)^2 T_k\sigma^2 - n_{i,k}^\star T_k\sigma^2}{\left(n_{i,k}^\star + T_k\right)\left(-n_{i,k}^\star T_k + 4n_{i,k}^\star \alpha_{i,k}^2 + 4T_k\alpha_{i,k}^2\right)\sigma^2}$$

$$+ \frac{-6c_{i,k}n_{i,k}^\star T_k^2 + \frac{T_k^2}{n_{i,k}^\star}\left(2c_{i,k}(n_{i,k}^\star)^2 + \sigma^2\right)}{\left(n_{i,k}^\star + T_k\right)\sigma^2} \tag{42}$$

$$+ \frac{\left(1 + \frac{T_k}{n_{i,k}^\star}\right)\left(16c_{i,k}(n_{i,k}^\star)^2 T_k \alpha_{i,k}^2 - 4T_k^2\alpha_{i,k}^2\sigma^2\right)}{\left(n_{i,k}^\star + T_k\right)\left(-n_{i,k}^\star T_k + 4n_{i,k}^\star \alpha_{i,k}^2 + 4T_k\alpha_{i,k}^2\right)\sigma^2} \tag{43}$$

$$+ \frac{-4c_{i,k}(n_{i,k}^\star)^2 T_k^3 + T_k^3 \sigma^2}{\left(n_{i,k}^\star + T_k\right)\left(-n_{i,k}^\star T_k + 4n_{i,k}^\star \alpha_{i,k}^2 + 4T_k \alpha_{i,k}^2\right)\sigma^2} \tag{44}$$

Now notice that if we can show lines (42), (43), and (44) sum to 0 then we are done with the proof. We can rewrite the sum of these three lines as

$$\frac{\left(-4c_{i,k}n_{i,k}^\star T_k^2 + \frac{T_k^2 \sigma^2}{n_{i,k}^\star}\right)\left(-n_{i,k}^\star T_k + 4n_{i,k}^\star \alpha_{i,k}^2 + 4T_k \alpha_{i,k}^2\right)}{\left(n_{i,k}^\star + T_k\right)\left(-n_{i,k}^\star T_k + 4n_{i,k}^\star \alpha_{i,k}^2 + 4T_k \alpha_{i,k}^2\right)\sigma^2}$$

$$+ \frac{\left(1 + \frac{T_k}{n_{i,k}^\star}\right)\left(16c_{i,k}(n_{i,k}^\star)^2 T_k \alpha_{i,k}^2 - 4T_k^2 \alpha_{i,k}^2 \sigma^2\right)}{\left(n_{i,k}^\star + T_k\right)\left(-n_{i,k}^\star T_k + 4n_{i,k}^\star \alpha_{i,k}^2 + 4T_k \alpha_{i,k}^2\right)\sigma^2}$$

$$+ \frac{-4c_{i,k}(n_{i,k}^\star)^2 T_k^3 + T_k^3 \sigma^2}{\left(n_{i,k}^\star + T_k\right)\left(-n_{i,k}^\star T_k + 4n_{i,k}^\star \alpha_{i,k}^2 + 4T_k \alpha_{i,k}^2\right)\sigma^2}$$

$$= \left(\left(n_{i,k}^\star + T_k\right)\left(-n_{i,k}^\star T_k + 4n_{i,k}^\star \alpha_{i,k}^2 + 4T_k \alpha_{i,k}^2\right)\sigma^2\right)^{-1}$$

$$\times \left(\left(\left(-4c_{i,k}n_{i,k}^\star T_k^2 + \frac{T_k^2 \sigma^2}{n_{i,k}^\star}\right)\left(-n_{i,k}^\star T_k + 4n_{i,k}^\star \alpha_{i,k}^2 + 4T_k \alpha_{i,k}^2\right) \right. \right. \tag{45}$$

$$\left. \left. + \left(1 + \frac{T_k}{n_{i,k}^\star}\right)\left(16c_{i,k}(n_{i,k}^\star)^2 T_k \alpha_{i,k}^2 - 4T_k^2 \alpha_{i,k}^2 \sigma^2\right) - 4c_{i,k}(n_{i,k}^\star)^2 T_k^3 + T_k^3 \sigma^2\right)\right) \tag{46}$$

We will conclude by showing that lines (45) and (46) sum to 0 which will complete the proof.

$$\left(-4c_{i,k}n_{i,k}^\star T_k^2 + \frac{T_k^2 \sigma^2}{n_{i,k}^\star}\right)\left(-n_{i,k}^\star T_k + 4n_{i,k}^\star \alpha_{i,k}^2 + 4T_k \alpha_{i,k}^2\right)$$

$$+ \left(1 + \frac{T_k}{n_{i,k}^\star}\right)\left(16c_{i,k}(n_{i,k}^\star)^2 T_k \alpha_{i,k}^2 - 4T_k^2 \alpha_{i,k}^2 \sigma^2\right) - 4c_{i,k}(n_{i,k}^\star)^2 T_k^3 + T_k^3 \sigma^2$$

$$= -c_{i,k}(n_{i,k}^\star)^2 T_k^3 - 16c_{i,k}(n_{i,k}^\star)^2 T_k^2 \alpha_{i,k}^2 - 16c_{i,k}n_{i,k}^\star T_k^3 \alpha_{i,k}^2$$

$$- T_k^3 \sigma^2 + 4T_k^2 \sigma^2 \alpha_{i,k}^2 + \frac{4T_k^3 \sigma^2 \alpha_{i,k}^2}{n_{i,k}^\star}$$

$$+ 16c_{i,k}(n_{i,k}^\star)^2 T_k^2 \alpha_{i,k}^2 - 4T_k^2 \alpha_{i,k}^2 \sigma^2$$

$$+ 16c_{i,k}n_{i,k}^\star T_k^3 \alpha_{i,k}^2 - \frac{4T_k^3 \alpha_{i,k}^2 \sigma^2}{n_{i,k}^\star}$$

$$- 4c_{i,k}(n_{i,k}^\star)^2 T_k^3 + T_k^3 \sigma^2$$

$$= 0$$

$$\square$$

**Lemma 20.** *Let* $\alpha_{i,k} = \sqrt{n_{i,k}^\star}$. *Then*

$$\frac{T_k(2c_{i,k}(n_{i,k}^\star)^2 + 6c_{i,k}n_{i,k}^\star T_k + \sigma^2)}{(n_{i,k}^\star + T_k)(2c_{i,k}n_{i,k}^\star T_k + \sigma^2)} + \frac{4c_{i,k}(n_{i,k}^\star)^2(T_k)^3 - 2(n_{i,k}^\star)^2 T_k \sigma^2 - n_{i,k}^\star(T_k)^2 \sigma^2}{(n_{i,k}^\star + T_k)(-n_{i,k}^\star T_k + 4n_{i,k}^\star \alpha_{i,k}^2 + 4T_k \alpha_{i,k}^2)(2c_{i,k}n_{i,k}^\star T_k + \sigma^2)}$$

$$= \frac{2T_k\left(c_{i,k}n_{i,k}^\star\left(4n_{i,k}^\star + 11T_k\right) + \sigma^2\right)}{\left(4n_{i,k}^\star + 3T_k\right)\left(2c_{i,k}n_{i,k}^\star T_k + \sigma^2\right)} .$$

*Proof.* Via a sequence of algebraic manipulations, we have

$$
\frac{T_k(2c_{i,k}(n_{i,k}^\star)^2 + 6c_{i,k}n_{i,k}^\star T_k + \sigma^2)}{(n_{i,k}^\star + T_k)\left(2c_{i,k}n_{i,k}^\star T_k + \sigma^2\right)} + \frac{4c_{i,k}(n_{i,k}^\star)^2(T_k)^3 - 2(n_{i,k}^\star)^2 T_k \sigma^2 - n_{i,k}^\star(T_k)^2\sigma^2}{(n_{i,k}^\star + T_k)(-n_{i,k}^\star T_k + 4n_{i,k}^\star \alpha_{i,k}^2 + 4T_k \alpha_{i,k}^2)\left(2c_{i,k}n_{i,k}^\star T_k + \sigma^2\right)}
$$

$$
= \frac{\frac{(4n_{i,k}^\star + 3T_k)}{(n_{i,k}^\star + T_k)}T_k(2c_{i,k}(n_{i,k}^\star)^2 + 6c_{i,k}n_{i,k}^\star T_k + \sigma^2)}{(4n_{i,k}^\star + 3T_k)\left(2c_{i,k}n_{i,k}^\star T_k + \sigma^2\right)} + \frac{4c_{i,k}(n_{i,k}^\star)^2(T_k)^3 - 2(n_{i,k}^\star)^2 T_k \sigma^2 - n_{i,k}^\star(T_k)^2\sigma^2}{(n_{i,k}^\star + T_k)n_{i,k}^\star(4n_{i,k}^\star + 3T_k)\left(2c_{i,k}n_{i,k}^\star T_k + \sigma^2\right)}
$$

$$
= \frac{\frac{(4n_{i,k}^\star + 3T_k)}{(n_{i,k}^\star + T_k)}T_k(2c_{i,k}(n_{i,k}^\star)^2 + 6c_{i,k}n_{i,k}^\star T_k + \sigma^2)}{(4n_{i,k}^\star + 3T_k)\left(2c_{i,k}n_{i,k}^\star T_k + \sigma^2\right)} + \frac{\frac{1}{n_{i,k}^\star + T_k}\left(4c_{i,k}(n_{i,k}^\star)^2(T_k)^3 - 2(n_{i,k}^\star)^2 T_k \sigma^2 - n_{i,k}^\star(T_k)^2\sigma^2\right)}{(4n_{i,k}^\star + 3T_k)\left(2c_{i,k}n_{i,k}^\star T_k + \sigma^2\right)}
$$

$$
= \frac{\frac{1}{(n_{i,k}^\star + T_k)}}{(4n_{i,k}^\star + 3T_k)\left(2c_{i,k}n_{i,k}^\star T_k + \sigma^2\right)}\bigg(\left(4n_{i,k}^\star + 3T_k\right)T\left(2c_{i,k}(n_{i,k}^\star)^2 + 6c_{i,k}n_{i,k}^\star T_k + \sigma^2\right)
$$

$$
+ 4c_{i,k}n_i^\star T_k^3 - 2n_{i,k}^\star T_k \sigma^2 - T_k^2 \sigma^2\bigg)
$$

$$
= \frac{\frac{1}{(n_{i,k}^\star + T_k)}}{(4n_{i,k}^\star + 3T_k)\left(2c_{i,k}n_{i,k}^\star T_k + \sigma^2\right)}\left(8c_{i,k}(n_{i,k}^\star)^3 T_k + 30c_{i,k}(n_{i,k}^\star)^2 T_k^2 + 2n_{i,k}^\star T_k \sigma^2 + 22c_{i,k}n_{i,k}^\star T_k^3 + 2T_k^2 \sigma^2\right)
$$

$$
= \frac{\frac{1}{(n_{i,k}^\star + T_k)}}{(4n_{i,k}^\star + 3T_k)\left(2c_{i,k}n_{i,k}^\star T_k + \sigma^2\right)}\left(2T_k\left(n_{i,k}^\star + T_k\right)\left(c_{i,k}n_{i,k}^\star\left(4n_{i,k}^\star + 11T_k\right) + \sigma^2\right)\right)
$$

$$
= \frac{2T_k\left(c_{i,k}n_{i,k}^\star\left(4n_{i,k}^\star + 11T_k\right) + \sigma^2\right)}{(4n_{i,k}^\star + 3T_k)\left(2c_{i,k}n_{i,k}^\star T_k + \sigma^2\right)}
$$

$\square$

**Lemma 21.** *Let* $\alpha_{i,k} = \sqrt{n_{i,k}^\star}$*. Then*

$$
\frac{T_k(2c_{i,k}(n_{i,k}^\star)^2 + 6c_{i,k}n_{i,k}^\star T_k + \sigma^2)}{(n_{i,k}^\star + T_k)\sigma^2} + \frac{4c_{i,k}(n_{i,k}^\star)^2(T_k)^3 - 2(n_{i,k}^\star)^2 T_k \sigma^2 - n_{i,k}^\star(T_k)^2\sigma^2}{(n_{i,k}^\star + T_k)(-n_{i,k}^\star T_k + 4n_{i,k}^\star \alpha_{i,k}^2 + 4T_k \alpha_{i,k}^2)\sigma^2}
$$

$$
= \frac{2T_k\left(c_{i,k}n_{i,k}^\star\left(4n_{i,k}^\star + 11T_k\right) + \sigma^2\right)}{\left(4n_{i,k}^\star + 3T_k\right)\sigma^2}\ .
$$

*Proof.* Via a sequence of algebraic manipulations, we have

$$
\frac{T_k(2c_{i,k}(n_{i,k}^\star)^2 + 6c_{i,k}n_{i,k}^\star T_k + \sigma^2)}{(n_{i,k}^\star + T_k)\sigma^2} + \frac{4c_{i,k}(n_{i,k}^\star)^2(T_k)^3 - 2(n_{i,k}^\star)^2 T_k \sigma^2 - n_{i,k}^\star(T_k)^2\sigma^2}{(n_{i,k}^\star + T_k)(-n_{i,k}^\star T_k + 4n_{i,k}^\star \alpha_{i,k}^2 + 4T_k \alpha_{i,k}^2)\sigma^2}
$$

$$
= \frac{\frac{(4n_{i,k}^\star + 3T_k)}{(n_{i,k}^\star + T_k)}T_k(2c_{i,k}(n_{i,k}^\star)^2 + 6c_{i,k}n_{i,k}^\star T_k + \sigma^2)}{(4n_{i,k}^\star + 3T_k)\sigma^2} + \frac{4c_{i,k}(n_{i,k}^\star)^2(T_k)^3 - 2(n_{i,k}^\star)^2 T_k \sigma^2 - n_{i,k}^\star(T_k)^2\sigma^2}{(n_{i,k}^\star + T_k)n_{i,k}^\star(4n_{i,k}^\star + 3T_k)\sigma^2}
$$

$$
= \frac{\frac{(4n_{i,k}^\star + 3T_k)}{(n_{i,k}^\star + T_k)}T_k(2c_{i,k}(n_{i,k}^\star)^2 + 6c_{i,k}n_{i,k}^\star T_k + \sigma^2)}{(4n_{i,k}^\star + 3T_k)\sigma^2} + \frac{\frac{1}{n_{i,k}^\star + T_k}\left(4c_{i,k}(n_{i,k}^\star)^2(T_k)^3 - 2(n_{i,k}^\star)^2 T_k \sigma^2 - n_{i,k}^\star(T_k)^2\sigma^2\right)}{(4n_{i,k}^\star + 3T_k)\sigma^2}
$$

$$
= \frac{\frac{1}{(n_{i,k}^\star + T_k)}}{(4n_{i,k}^\star + 3T_k)\sigma^2}\bigg(\left(4n_{i,k}^\star + 3T_k\right)T\left(2c_{i,k}(n_{i,k}^\star)^2 + 6c_{i,k}n_{i,k}^\star T_k + \sigma^2\right) + 4c_{i,k}n_i^\star T_k^3 - 2n_{i,k}^\star T_k \sigma^2 - T_k^2 \sigma^2\bigg)
$$

$$= \frac{\frac{1}{(n_{i,k}^\star + T_k)}}{(4n_{i,k}^\star + 3T_k)\sigma^2} \left( 8c_{i,k}(n_{i,k}^\star)^3 T_k + 30c_{i,k}(n_{i,k}^\star)^2 T_k^2 + 2n_{i,k}^\star T_k \sigma^2 + 22c_{i,k}n_{i,k}^\star T_k^3 + 2T_k^2 \sigma^2 \right)$$

$$= \frac{\frac{1}{(n_{i,k}^\star + T_k)}}{(4n_{i,k}^\star + 3T_k)\sigma^2} \left( 2T_k \left( n_{i,k}^\star + T_k \right) \left( c_{i,k} n_{i,k}^\star \left( 4n_{i,k}^\star + 11T_k \right) + \sigma^2 \right) \right)$$

$$= \frac{2T_k \left( c_{i,k} n_{i,k}^\star \left( 4n_{i,k}^\star + 11T_k \right) + \sigma^2 \right)}{(4n_{i,k}^\star + 3T_k)\sigma^2} .$$

$\square$

# H. Results based on other work

This section presents the proof of lemmas related to incentive-compatibility. As we use a similar corruption strategy to Chen et al. (2023), the high-level proof ideas are similar to theirs. However, adapting specific techniques to unequal costs requires nontrivial algebraic adaptations.

## H.1. Proof of Lemma 3

In Algorithm 2 we have specified the information set space for $M_{\text{CBL}}$ to be $\mathbb{R}^d$, which corresponds to $d$ estimates of the $d$ distributions. We could have defined an equivalent mechanism where the information set space is $\mathcal{I} = \mathcal{D} \times \mathcal{D} \times \mathbb{R}^d$ and $I_i = (I_{i,1}, \ldots, I_{i,d})$ where $I_{i,k} = (Z, Z', \eta^2)$. In this setup $Z$ is clean data, $Z'$ is corrupted data, and $\eta^2$ is the amount of noise added to obtain $Z'$. For the sake of convenience, we define the following sets to characterize the types of information sets $i$ receives for each distribution. We also specify the information sets provided under this alternative information set space.

$$
\begin{aligned}
A &:= \{k : i \in V_k\} & &\text{if } k \in A \text{ then } I_{i,k} = (\varnothing, \varnothing, 0) \\
B &:= \{k : n_{i,k}^\star = 0 \text{ and } i \notin V_k\} & &\text{if } k \in B \text{ then } I_{i,k} = (Y_{-i,k}, \varnothing, 0) \\
C &:= \{k : n_{i,k}^\star > 0 \text{ and } i \notin V_k\} & &\text{if } k \in C \text{ then } I_{i,k} = \left( Z_{i,k}, Z'_{i,k}, \eta_{i,k}^2 \right)
\end{aligned}
$$

Under this change, we would now recommend agent $i$ to use the estimator

$$
h_{i,k}(X_i, Y_i, I_i) = \frac{\frac{1}{\sigma^2 |Y_{i,k} \cup Z_{i,k}|} \sum_{y \in Y_{i,k} \cup Z_{i,k}} y + \frac{1}{(\sigma^2 + \eta_{i,k}^2)|Z'_{i,k}|} \sum_{z \in Z'_{i,k}} z}{\frac{1}{\sigma^2} |Y_{i,k} \cup Z_{i,k}| + \frac{1}{\sigma^2 + \eta_{i,k}^2} |Z'_{i,k}|}
$$

instead of directly accepting estimates directly from the mechanism. In fact, Chen et al. (2023) use the single variable version of this exact information set space. Notice that in this setup agents receive more information since, in each case, the estimate recommended by $M_{\text{CBL}}$ is function of $(Z, Z', \eta^2)$. In the proof of this lemma, we will show that if agent $i$ has access to $I_{i,k} = (Z, Z', \eta^2)$ then it is optimal to submit data truthfully and compute the same estimates that are returned by $M_{\text{CBL}}$. Therefore, when $\mathcal{I} = \mathbb{R}^d$, and the agent only receives recommended estimates in place of $(Z, Z', \eta^2)$, the agent will achieve the same penalty by submitting truthfully and accepting the estimates of the mechanism, even when they now receive less information. This will imply that submitting truthfully and accepting estimates from the mechanism is the optimal strategy in the information set space $\mathcal{I} = \mathbb{R}^d$. We now present an argument of this claim, adapting the proof of Lemma 5 from Chen et al. (2023).

- **Step 1.** First, we construct a sequence of multivariable prior distributions $\{\Lambda_\ell\}_{\ell \geq 1}$ for $\mu \in \mathbb{R}^d$ and calculate the sequence of Bayesian risks under the prior distributions

$$
R_\ell := \inf_{s'_i \in \Delta(\mathcal{S})'} \mathbb{E}_{\mu \sim \Lambda_\ell} \left[ \mathbb{E}_{(s'_i, s^\star_{-i})} \left[ \mathbb{E}_{\mu, M_{\text{CBL}}} \left[ \| h_i(X_i, Y_i, I_i) - \mu \|_2^2 \right] \right] \right], \quad \ell \geq 1
$$

Here $\Delta(\mathcal{S})'$ is the set of strategies where $i$ collects the same amount of data as in $s_i$, i.e. $\Delta(\mathcal{S})' := \{(n'_i f'_i, h'_i) = s'_i \in \Delta(\mathcal{S}) : n'_i = n_i \text{ a.s.}\}$.

- **Step 2.** Next, we define $\widetilde{s}_i$ to be the randomized strategy given by $(n_i, f_i^\star, h_i^\star)$. We then show that

$$\lim_{\ell \to \infty} R_\ell = \sup_{\mu} \left[ \mathbb{E}_{(\widetilde{s}_i, s_{-i}^\star)} \left[ \mathbb{E}_{\mu, M_{\text{CBL}}} \left[ \| h_i\left(X_i, Y_i, I_i\right) - \mu \|_2^2 \right] \right] \right]$$

- **Step 3.** Finally, using that the Bayesian risk is a lower bound on the maximum risk, we conclude that amongst the strategies in $\Delta(\mathcal{S})'$, $\widetilde{s}_i$ is minimax optimal. Thus, an agent will suffer no worse of an error when using $f_i = f_i^\star$ and $h_i = h_i^\star$, regardless of the distribution over $n_i$.

**Step 1. (Bounding the Bayes' risk under the sequence of priors)** We will use a sequence of multivariable normal priors $\Lambda_\ell := \mathcal{N}\left(0, \ell^2 I_d\right)$ for $\ell \geq 1$. Notice that for an agent $i$, the mechanism returns an uncorrupted estimate for distribution $k$ if $i \in V_k$ or $n_{i,k}^\star = 0$ and a corrupted estimate otherwise. More specifically, if $i \in V_k$ an uncorrupted estimate using only $Y_{i,k}$ is returned whereas if $n_{i,k}^\star = 0$ an uncorrupted estimate using $Y_{-i,k}$ is returned.

Fix some $f_i \in \mathcal{F}$. We now analyze the optimal choice for $h_{i,k}$ depending on whether $k$ is in $A$, $B$, or $C$.

Suppose that $k \in A$. Recall that $x \mid \mu_k \sim \mathcal{N}(\mu_k, \sigma^2)$, $\forall x \in X_{i,k}$. Since $k \in A$, $I_{i,k}$ is just a function of $X_{i,k}$ as no other data from the other agents is given. Therefore, the posterior distribution for $\mu_k$ conditioned on $X_{i,k}, Y_{i,k}, I_{i,k}$ can be written as $\mu_k \mid X_{i,k}, Y_{i,k}, I_{i,k} = \mu_k \mid X_{i,k}$. Furthermore, it is well known (See Example 1.14 in Chapter 5 of (Lehmann & Casella, 2006)) that $\mu_k \mid X_{i,k} \sim \mathcal{N}\left(\mu_{A,\ell,k}, \sigma_{A,\ell,k}^2\right)$ where

$$\mu_{A,\ell,k} := \frac{\frac{|X_{i,k}|}{\sigma^2}}{\frac{|X_{i,k}|}{\sigma^2} + \frac{1}{\ell^2}} \mu_k + \frac{\frac{1}{\ell^2}}{\frac{|X_{i,k}|}{\sigma^2} + \frac{1}{\ell^2}} h^{\text{SM}}\left(X_{i,k}\right) \qquad \sigma_{A,\ell,k}^2 := \frac{1}{\frac{|X_{i,k}|}{\sigma^2} + \frac{1}{\ell^2}}$$

Therefore, $\mu_k \mid X_{i,k}, Y_{i,k}, I_{i,k} \sim \mathcal{N}\left(\mu_{A,\ell,k}, \sigma_{A,\ell,k}^2\right)$.

If $k \in B$ then agent $i$ receives $Y_{-i,k}$ regardless of what they submit. Again, we know from (Lehmann & Casella, 2006) that $\mu_k \mid X_{i,k}, Y_{i,k}, I_{i,k} \sim \mathcal{N}\left(\mu_{B,\ell,k}, \sigma_{B,\ell,k}^2\right)$ where

$$\mu_{B,\ell,k} := \frac{\frac{|X_{i,k} \cup Y_{-i,k}|}{\sigma^2}}{\frac{|X_{i,k} \cup Y_{-i,k}|}{\sigma^2} + \frac{1}{\ell^2}} \mu_k + \frac{\frac{1}{\ell^2}}{\frac{|X_{i,k} \cup Y_{-i,k}|}{\sigma^2} + \frac{1}{\ell^2}} h^{\text{SM}}\left(X_{i,k} \cup Y_{-i,k}\right) \qquad \sigma_{B,\ell,k}^2 := \frac{1}{\frac{|X_{i,k}|}{\sigma^2} + \frac{1}{\ell^2}}$$

If $k \in C$ we have that

$$x \mid \mu_k \sim \mathcal{N}\left(\mu_k, \sigma^2\right) \ \forall x \in X_{i,k} \cup Z_{i,k}$$
$$x \mid \mu_k, \eta_{i,k}^2 \sim \mathcal{N}\left(\mu_k, \sigma^2 + \eta_{i,k}^2\right) \ \forall x \in Z_{i,k}'$$

Recall that $\eta_{i,k}^2$ is a function of $Y_{i,k}$ and $Z_{i,k}$. Assume for now that $X_{i,-k}$ is fixed. Under this assumption, and having fixed $f_i$, both $Y_{i,k}$ and $\eta_{i,k}^2$ are deterministic functions of $X_{i,k}$ and $Z_{i,k}$. Therefore, the posterior distribution for $\mu_k$ conditioned on $(X_{i,k}, Y_{i,k}, I_{i,k})$ can be calculated as follows:

$$p\left(\mu_k | X_{i,k}, Y_{i,k}, I_{i,k}\right) = p\left(\mu_k | X_{i,k}, Y_{i,k}, Z_{i,k}, Z_{i,k}', \eta_{i,k}^2\right) = p\left(\mu_k | X_{i,k}, Z_{i,k}, Z_{i,k}'\right)$$
$$\propto p\left(\mu_k, X_{i,k}, Z_{i,k}, Z_{i,k}'\right) = p\left(Z_{i,k}' | X_{i,k}, Z_{i,k}, \mu_k\right) p\left(X_{i,k}, Z_{i,k} | \mu_k\right) p(\mu_k)$$
$$= p\left(Z_{i,k}' | X_{i,k}, Z_{i,k}, \mu_k\right) p\left(X_{i,k} | \mu_k\right) p\left(Z_{i,k} | \mu_k\right) p(\mu_k)$$
$$\propto \exp\left(-\frac{1}{2(\sigma^2 + \eta_{i,k}^2)} \sum_{x \in Z_{i,k}'} (x - \mu_k)^2\right) \exp\left(-\frac{1}{2\sigma^2} \sum_{x \in X_{i,k} \cup Z_{i,k}} (x - \mu_k)^2\right) \exp\left(-\frac{\mu_k^2}{2\ell^2}\right)$$
$$\propto \exp\left(-\frac{1}{2}\left(\frac{|Z_{i,k}'|}{\sigma^2 + \eta_{i,k}^2} + \frac{|X_{i,k}| + |Z_{i,k}|}{\sigma^2} + \frac{1}{\ell^2}\right) \mu_k^2\right) \exp\left(\frac{1}{2} 2 \left(\frac{\sum_{x \in Z_{i,k}'} x}{\sigma^2 + \eta_{i,k}^2} + \frac{\sum_{x \in X_{i,k} \cup Z_{i,k}} x}{\sigma^2}\right) \mu_k\right)$$

$$= \exp\left(-\frac{1}{2}\left(\frac{1}{\sigma_{C,\ell,k}^2}\mu_k^2 - 2\frac{\mu_{C,\ell,k}}{\sigma_{C,\ell,k}^2}\mu_k\right)\right) \propto \exp\left(-\frac{1}{2\sigma_{C,\ell,k}^2}(\mu_k - \mu_{C,\ell,k})^2\right)$$

where

$$\mu_{C,\ell,k} := \frac{\frac{\sum_{x \in Z'_{i,k}} x}{\sigma^2 + \eta_{i,k}^2} + \frac{\sum_{x \in X_{i,k} \cup Z_{i,k}} x}{\sigma^2}}{\frac{|Z'_{i,k}|}{\sigma^2 + \eta_{i,k}^2} + \frac{|X_{i,k}| + |Z_{i,k}|}{\sigma^2} + \frac{1}{\ell^2}} \quad \text{and} \quad \sigma_{C,\ell,k}^2 := \frac{1}{\frac{|Z'_{i,k}|}{\sigma^2 + \eta_{i,k}^2} + \frac{|X_{i,k}| + |Z_{i,k}|}{\sigma^2} + \frac{1}{\ell^2}}$$

We can therefore conclude that (despite the non i.i.d nature of the data), the posterior distribution of $\mu_k$ conditioned on $X_{i,k}, Y_{i,k}, I_{i,k}$, assuming $X_{i,-k}$ is fixed, is Gaussian with mean and variance as given above, i.e.

$$\mu_k \mid X_{i,k}, Y_{i,k}, I_{i,k} \sim \mathcal{N}\left(\mu_{C,\ell,k}, \sigma_{C,\ell,k}^2\right)$$

Next, following standard steps (See Corollary 1.2 in Chapter 4 of (Lehmann & Casella, 2006)), we know that $\mathbb{E}_{\mu_k}\left[(h_{i,k}(X_i, Y_i, I_i) - \mu_k)^2 \mid X_{i,k}, Y_{i,k}, I_{i,k}\right]$ is minimized when

$$h_{i,k}(X_i, Y_i, I_i) = \mathbb{E}_{\mu_k}\left[\mu_k \mid X_{i,k}, Y_{i,k}, I_{i,k}\right]$$

When $X_{i,-k}$ is assumed to be fixed, $\mathbb{E}_{\mu_k}\left[\mu_k \mid X_{i,k}, Y_{i,k}, I_{i,k}\right] = \mu_{C,\ell,k}$. This shows that for any $f_i \in \mathcal{F}$ and fixed $X_{i,-k}$, the optimal $h_{i,k}$ is simply the posterior mean of $\mu_k$ under the prior $\Lambda_\ell$ conditioned on $(X_{i,k}, Y_{i,k}, I_{i,k})$.

We have now derived our optimal choices for $h_{i,k}$ (either $\mu_{A,\ell,k}, \mu_{B,\ell,k}$, or $\mu_{C,\ell,k}$) depending on whether $k \in A$, $k \in B$, or $k \in C$. Going forward, we will also use $f_i$ and $h_i$ to denote probability distributions over $\mathcal{F}$ and $\mathcal{H}$ respectively. For some fixed $f_i \in \Delta(\mathcal{F})$, we can rewrite the minimum average risk over $h_i \in \Delta(\mathcal{H})$ by switching the order of expectation:

$$\inf_{h_i \in \Delta(\mathcal{H})} \mathbb{E}_{\mu \sim \Lambda_\ell}\left[\mathbb{E}_{((n_i, f_i, h_i), s^\star_{-i})}\left[\mathbb{E}_{\mu, M_{\text{CBL}}}\left[\|h_i(X_i, Y_i, I_i) - \mu\|_2^2\right]\right]\right]$$

$$= \inf_{h_i \in \Delta(\mathcal{H})} \mathbb{E}_{\mu \sim \Lambda_\ell}\left[\mathbb{E}_{(n_i, f_i, h_i)}\left[\mathbb{E}_{\mu, M_{\text{CBL}}}\left[\sum_{k=1}^d (h_{i,k}(X_i, Y_i, I_i) - \mu_k)^2\right]\right]\right]$$

$$= \inf_{h_i \in \Delta(\mathcal{H})} \sum_{k=1}^d \mathbb{E}_{(n_i, f_i, h_i)}\left[\mathbb{E}_{X_i, Z_i, Z'_i}\left[\mathbb{E}_\mu\left[(h_{i,k}(X_i, Y_i, I_i) - \mu_k)^2 \mid X_i, Z_i, Z'_i\right]\right]\right]$$

$$= \inf_{h_i \in \Delta(\mathcal{H})} \sum_{k=1}^d \mathbb{E}_{(n_i, f_i, h_i)}\left[\mathbb{E}_{\substack{X_{i,-k}, Z_{i,-k}, \\ Z'_{i,-k}, \mu_{-k}}}\left[\mathbb{E}_{X_{i,k}, Z_{i,k}, Z'_{i,k}}\right.\right. \tag{47}$$

$$\left.\left.\mathbb{E}_{\mu_k}\left[(h_{i,k}(X_i, Y_i, I_i) - \mu_k)^2 \mid X_{i,k}, Z_{i,k}, Z'_{i,k}\right] \mid X_{i,-k}, Z_{i,-k}, Z'_{i,-k}, \mu_{-k}\right]\right] \tag{48}$$

Substituting in the appropriate posterior depending on whether $k$ is in $A$, $B$, or $C$ tell us that the equation in lines (47) and (48) is bounded below by

$$\sum_{k \in A} \mathbb{E}_{(n_i, f_i)}\left[\mathbb{E}_{\substack{X_{i,-k}, Z_{i,-k}, \\ Z'_{i,-k}, \mu_{-k}}}\left[\mathbb{E}_{X_i, Z_i, Z'_i}\left[\mathbb{E}_{\mu_k}\left[(\mu_{A,\ell,k} - \mu_k)^2 \mid X_{i,k}, Z_{i,k}, Z'_{i,k}\right] \mid X_{i,-k}, Z_{i,-k}, Z'_{i,-k}, \mu_{-k}\right]\right]\right]$$

$$+ \sum_{k \in B} \mathbb{E}_{(n_i, f_i)}\left[\mathbb{E}_{\substack{X_{i,-k}, Z_{i,-k}, \\ Z'_{i,-k}, \mu_{-k}}}\left[\mathbb{E}_{X_{i,k}, Z_{i,k}, Z'_{i,k}}\left[\mathbb{E}_{\mu_k}\left[(\mu_{B,\ell,k} - \mu_k)^2 \mid X_{i,k}, Z_{i,k}, Z'_{i,k}\right] \mid X_{i,-k}, Z_{i,-k}, Z'_{i,-k}, \mu_{-k}\right]\right]\right]$$

$$+ \sum_{k \in C} \mathbb{E}_{(n_i, f_i)}\left[\mathbb{E}_{\substack{X_{i,-k}, Z_{i,-k}, \\ Z'_{i,-k}, \mu_{-k}}}\left[\mathbb{E}_{X_{i,k}, Z_{i,k}, Z'_{i,k}}\left[\mathbb{E}_{\mu_k}\left[(\mu_{C,\ell,k} - \mu_k)^2 \mid X_{i,k}, Z_{i,k}, Z'_{i,k}\right] \mid X_{i,-k}, Z_{i,-k}, Z'_{i,-k}, \mu_{-k}\right]\right]\right]$$

$$= \sum_{k \in A} \mathop{\mathbb{E}}_{(n_i, f_i)} \left[ \mathop{\mathbb{E}}_{\substack{X_{i,-k}, Z_{i,-k}, \\ Z'_{i,-k}, \mu_{-k}}} \left[ \mathop{\mathbb{E}}_{X_{i,k}, Z_{i,k}, Z'_{i,k}} \left[ \frac{1}{\frac{|X_{i,k}|}{\sigma^2} + \frac{1}{\ell^2}} \right] \right] \right] + \sum_{k \in B} \mathop{\mathbb{E}}_{(n_i, f_i)} \left[ \mathop{\mathbb{E}}_{\substack{X_{i,-k}, Z_{i,-k}, \\ Z'_{i,-k}, \mu_{-k}}} \left[ \mathop{\mathbb{E}}_{X_{i,k}, Z_{i,k}, Z'_{i,k}} \left[ \frac{1}{\frac{|X_{i,k} \cup Y_{-i,k}|}{\sigma^2} + \frac{1}{\ell^2}} \right] \right] \right] \tag{49}$$

$$+ \sum_{k \in C} \mathop{\mathbb{E}}_{(n_i, f_i)} \left[ \mathop{\mathbb{E}}_{\substack{X_{i,-k}, Z_{i,-k}, \\ Z'_{i,-k}, \mu_{-k}}} \left[ \mathop{\mathbb{E}}_{X_{i,k}, Z_{i,k}, Z'_{i,k}} \left[ \frac{1}{\frac{|Z'_{i,k}|}{\sigma^2 + \eta_{i,k}^2} + \frac{|X_{i,k}| + |Z_{i,k}|}{\sigma^2} + \frac{1}{\ell^2}} \right] \right] \right] \tag{50}$$

Now notice that if $k \in A$ or $k \in B$ then $I_{i,k}$ does not depend on $Y_{i,k}$ so lines (49) and (50) become

$$\sum_{k \in A} \mathop{\mathbb{E}}_{n_{i,k}} \left[ \frac{1}{\frac{|X_{i,k}|}{\sigma^2} + \frac{1}{\ell^2}} \right] + \sum_{k \in B} \mathop{\mathbb{E}}_{n_{i,k}} \left[ \frac{1}{\frac{|X_{i,k} \cup Y_{-i,k}|}{\sigma^2} + \frac{1}{\ell^2}} \right] \tag{51}$$

$$+ \sum_{k \in C} \mathop{\mathbb{E}}_{(n_i, f_i)} \left[ \mathop{\mathbb{E}}_{\substack{X_{i,-k}, Z_{i,-k}, \\ Z'_{i,-k}, \mu_{-k}}} \left[ \mathop{\mathbb{E}}_{X_{i,k}, Z_{i,k}} \left[ \frac{1}{\frac{|Z'_{i,k}|}{\sigma^2 + \eta_{i,k}^2} + \frac{|X_{i,k}| + |Z_{i,k}|}{\sigma^2} + \frac{1}{\ell^2}} \right] \right] \right] \tag{52}$$

Since $\eta_{i,k}^2$ depends on $\mu_{-k}, X_{i,-k}, X_{i,k}, Z_{i,k}, |X_i|, |Z_{i,k}|$, and $\left| Z'_{i,k} \right|$ but not $Z_{i,-k}$ nor $Z'_{i,-k}$, we can write (51) and (52) as

$$\sum_{k \in A} \mathop{\mathbb{E}}_{n_{i,k}} \left[ \frac{1}{\frac{|X_{i,k}|}{\sigma^2} + \frac{1}{\ell^2}} \right] + \sum_{k \in B} \mathop{\mathbb{E}}_{n_{i,k}} \left[ \frac{1}{\frac{|X_{i,k} \cup Y_{-i,k}|}{\sigma^2} + \frac{1}{\ell^2}} \right] \tag{53}$$

$$+ \sum_{k \in C} \mathop{\mathbb{E}}_{(n_i, f_i)} \left[ \mathop{\mathbb{E}}_{\mu_{-k}, X_{i,-k}} \left[ \mathop{\mathbb{E}}_{X_{i,k}, Z_{i,k}} \left[ \frac{1}{\frac{|Z'_{i,k}|}{\sigma^2 + \eta_{i,k}^2} + \frac{|X_{i,k}| + |Z_{i,k}|}{\sigma^2} + \frac{1}{\ell^2}} \right] \right] \right] \tag{54}$$

The second to last step follows from the fact that if $k \in A$ or $k \in B$ then $I_{i,k}$ does not depend on $Y_{i,k}$. Notice that the terms in (53) do not depend on $f_i$ whereas the terms in (54) do. We will now show that (54) (and thus the entire equation) is minimized for the following choice of $f_i$ which shrinks each point in $X_{i,k}$ by an amount that depends on the prior $\Lambda_\ell$:

$$f_{i,k}(X_i) = \left\{ \frac{\frac{|X_{i,k}|}{\sigma^2}}{\frac{|X_{i,k}|}{\sigma^2} + \frac{1}{\ell^2}} x, \quad \forall x \in X_{i,k} \right\} \tag{55}$$

To prove this, we first define the following quantities:

$$\widehat{\mu}(X_{i,k}) := \frac{1}{|X_{i,k}|} \sum_{x \in X_{i,k}} x \qquad \widehat{\mu}(Y_{i,k}) := \frac{1}{|Y_{i,k}|} \sum_{y \in Y_{i,k}} y \qquad \widehat{\mu}(Z_{i,k}) := \frac{1}{|Z_{i,k}|} \sum_{z \in Z_{i,k}} z$$

We will also find it useful to express $\eta_{i,k}^2$ as follows (here $\alpha_{i,k}$ is as defined in (11)):

$$\eta_{i,k}^2 = \alpha_{i,k}^2 \left( \widehat{\mu}(Y_{i,k}) - \widehat{\mu}(Z_{i,k}) \right)^2$$

The following calculations show that, conditioned on $X_{i,k}$, $\widehat{\mu}(Z_{i,k}) - \mu_k$ and $\mu_k - \frac{|X_{i,k}|/\sigma^2}{|X_{i,k}|/\sigma^2 + 1/\ell^2} \widehat{\mu}(X_{i,k})$ are independent Gaussian random variables:

$$p\left( \widehat{\mu}(Z_{i,k}) - \mu_k, \mu_k | X_{i,k} \right) \propto p\left( \widehat{\mu}(Z_{i,k}) - \mu_k, \mu_k, X_{i,k} \right)$$
$$= p\left( \widehat{\mu}(Z_{i,k}) - \mu_k, X_{i,k} | \mu_k \right) p(\mu_k) = p\left( \widehat{\mu}(Z_{i,k}) - \mu_k | \mu_k \right) p\left( X_{i,k} | \mu_k \right) p(\mu_k)$$

$$\propto \exp\left( -\frac{1}{2} \frac{|Z_{i,k}|}{\sigma^2} \left( \widehat{\mu}(Z_{i,k}) - \mu_k \right)^2 \right) \exp\left( -\frac{1}{2\sigma^2} \sum_{x \in X_{i,k}} (x - \mu_k)^2 \right) \exp\left( -\frac{1}{2\ell^2} \mu_k^2 \right)$$

$$\propto \underbrace{\exp\left(-\frac{1}{2}\frac{|Z_{i,k}|}{\sigma^2}\left(\widehat{\mu}(Z_{i,k})-\mu_k\right)^2\right)}_{\propto p(\widehat{\mu}(Z_{i,k})-\mu_k|X_{i,k})}\underbrace{\exp\left(-\frac{1}{2}\left(\frac{|X_{i,k}|}{\sigma^2}+\frac{1}{\ell^2}\right)\left(\mu_k-\frac{|X_{i,k}|/\sigma^2}{|X_{i,k}|/\sigma^2+1/\ell^2}\widehat{\mu}(X_{i,k})\right)^2\right)}_{\propto p\left(\mu_k-\frac{|X_{i,k}|/\sigma^2}{|X_{i,k}|/\sigma^2+1/\ell^2}\widehat{\mu}(X_{i,k})|X_{i,k}\right)}$$

Thus, conditioning on $X_{i,k}$, we can write

$$\begin{pmatrix}\widehat{\mu}(Z_{i,k})-\mu_k\\\mu_k-\frac{|X_{i,k}|/\sigma^2}{|X_{i,k}|/\sigma^2+1/\ell^2}\widehat{\mu}(X_{i,k})\end{pmatrix}\sim\mathcal{N}\left(\begin{pmatrix}0\\0\end{pmatrix},\begin{pmatrix}\frac{\sigma^2}{|Z_{i,k}|}&0\\0&\frac{1}{|X_{i,k}|/\sigma^2+1/\ell^2}\end{pmatrix}\right)$$

which leads us to

$$\widehat{\mu}(Z_{i,k})-\frac{|X_{i,k}|/\sigma^2}{|X_{i,k}|/\sigma^2+1/\ell^2}\widehat{\mu}(X_{i,k})\Bigg|X_{i,k}\sim\mathcal{N}\left(0,\underbrace{\frac{\sigma^2}{|Z_{i,k}|}+\frac{1}{|X_{i,k}|/\sigma^2+1/\ell^2}}_{=:\widetilde{\sigma}_{\ell,k}^2}\right)$$

Next, we rewrite the squared difference in $\eta_{i,k}^2$ as follows:

$$\frac{\eta_{i,k}^2}{\alpha_{i,k}^2}=\left(\widehat{\mu}(Y_{i,k})-\widehat{\mu}(Z_{i,k})\right)^2$$

$$=\left(\underbrace{\widehat{\mu}(Z_{i,k})-\frac{|X_{i,k}|/\sigma^2}{|X_{i,k}|/\sigma^2+1/\ell^2}\widehat{\mu}(X_{i,k})}_{=\widetilde{\sigma}_{\ell,k}e}+\underbrace{\left(\frac{|X_{i,k}|/\sigma^2}{|X_{i,k}|/\sigma^2+1/\ell^2}\widehat{\mu}(X_{i,k})-\widehat{\mu}(Y_{i,k})\right)}_{=:\phi(X_{i,k},f_i)}\right)^2 \tag{56}$$

Here we observe that the first part of the RHS above is equal to $\widetilde{\sigma}_{\ell,k}$ where $e$ is a noise $e|X_{i,k}\sim\mathcal{N}(0,1)$ where $\widetilde{\sigma}_{\ell,k}$ is defined in (56). For brevity, we denote the second part of the RHS as $\phi(X_{i,k},h_{i,k})$ which intuitively characterizes the difference between $X_{i,k}$ and $Y_{i,k}$. Importantly, $\phi(X_{i,k},h_{i,k})=0$ when $f_i$ is chosen to be (55). Using $e$ and $\phi$, we can rewrite each term in (54) using conditional expectation.

$$\underset{(n_i,f_i)}{\mathbb{E}}\left[\underset{\mu_{-k},X_{i,-k}}{\mathbb{E}}\left[\underset{X_i,Z_i}{\mathbb{E}}\left[\frac{1}{\frac{|Z'_{i,k}|}{\sigma^2+\eta_{i,k}^2}+\frac{|X_{i,k}|+|Z_{i,k}|}{\sigma^2}+\frac{1}{\ell^2}}\right]\right]\right]$$

$$=\underset{(n_i,f_i)}{\mathbb{E}}\left[\underset{\mu_{-k},X_{i,-k}}{\mathbb{E}}\left[\underset{X_i}{\mathbb{E}}\left[\underset{Z_{i,k}|X_{i,k}}{\mathbb{E}}\left[\frac{1}{\frac{|Z'_{i,k}|}{\sigma^2+\eta_{i,k}^2}+\frac{|X_{i,k}|+|Z_{i,k}|}{\sigma^2}+\frac{1}{\ell^2}}\right]\right]\right]\right]$$

$$=\underset{(n_i,f_i)}{\mathbb{E}}\left[\underset{\mu_{-k},X_{i,-k}}{\mathbb{E}}\left[\underset{X_i}{\mathbb{E}}\left[\underset{e|X_{i,k}}{\mathbb{E}}\left[\frac{1}{\frac{|Z'_{i,k}|}{\sigma^2+\alpha_{i,k}^2(\widetilde{\sigma}_{\ell,k}e+\phi(X_{i,k},f_{i,k}))^2}+\frac{|X_{i,k}|+|Z_{i,k}|}{\sigma^2}+\frac{1}{\ell^2}}\right]\right]\right]\right]$$

$$=\underset{(n_i,f_i)}{\mathbb{E}}\left[\underset{\mu_{-k},X_{i,-k}}{\mathbb{E}}\left[\underset{X_i}{\mathbb{E}}\left[\int_{-\infty}^{\infty}\underbrace{\frac{1}{\frac{|Z'_{i,k}|}{\sigma^2+\alpha_{i,k}^2\widetilde{\sigma}_{\ell,k}^2\left(e+\frac{\phi(X_{i,k},f_{i,k})}{\widetilde{\sigma}_{\ell,k}}\right)^2}+\frac{|X_{i,k}|+|Z_{i,k}|}{\sigma^2}+\frac{1}{\ell^2}}}_{=:F_{1,k}(e+\phi(X_{i,k},f_{i,k})/\widetilde{\sigma}_{\ell,k})}\underbrace{\frac{\exp\left(\frac{-e^2}{2}\right)de}{\sqrt{2\pi}}}_{=:F_2(e)}\right]\right]\right] \tag{57}$$

where we use the fact that $e|X_{i,k}\sim\mathcal{N}(0,1)$ in the last step. To proceed, we will consider the inner expectation in the RHS above. For any fixed $X_{i,k}$, $F_{1,k}(\cdot)$ (as marked on the RHS) is an even function that monotonically increases on $[0,\infty)$

bounded by $\frac{\sigma}{|X_{i,k}| + |Z_{i,k}|}$ and $F_2(\cdot)$ (as marked on the RHS) is an even function that monotonically decreases on $[0, \infty)$. That means, for any $a \in \mathbb{R}$,

$$\int_{-\infty}^{\infty} F_{1,k}(e - a)F_2(e)de \leq \int_{-\infty}^{\infty} \frac{\sigma}{|X_{i,k}| + |Z_{i,k}|}F_2(e)de = \frac{\sigma}{|X_{i,k}| + |Z_{i,k}|} < \infty$$

By Lemma 23, we have

$$\int_{-\infty}^{\infty} F_{1,k}(e + \phi(X_{i,k}, f_{i,k})/\widetilde{\sigma}_{\ell,k})F_2(e)de \geq \int_{-\infty}^{\infty} F_{1,k}(e)F_2(e)de, \tag{58}$$

the equality is achieved when $\phi(X_{i,k}, f_{i,k})/\widetilde{\sigma}_{\ell_k} = 0$. In particular, the equality holds when $f_{i,k}$ is chosen as specified in (55).

Now, to complete Step 1, we combine (53)/(54), (57), and (58) to obtain

$$\inf_{h_i \in \Delta(\mathcal{H})} \mathbb{E}_{\mu \sim \Lambda_\ell} \left[ \mathbb{E}_{((n_i, f_i, h_i), s_{-i}^\star)} \left[ \mathbb{E}_{\mu, M_{\mathrm{CBL}}} [\|h_i(X_i, Y_i, I_i) - \mu\|_2^2] \right] \right]$$

$$= \sum_{k \in A} \mathbb{E}_{n_{i,k}} \left[ \frac{1}{\frac{|X_{i,k}|}{\sigma^2} + \frac{1}{\ell^2}} \right] + \sum_{k \in B} \mathbb{E}_{n_{i,k}} \left[ \frac{1}{\frac{|X_{i,k} \cup Y_{-i,k}|}{\sigma^2} + \frac{1}{\ell^2}} \right]$$

$$+ \sum_{k \in C} \mathbb{E}_{(n_i, f_i)} \left[ \mathbb{E}_{\mu_{-k}, X_{i,-k}} \left[ \mathbb{E}_{X_{i,k}} \left[ \int_{-\infty}^{\infty} F_{1,k} \left( e + \frac{\phi(X_{i,k}, f_{i,k})}{\widetilde{\sigma}_{\ell,k}} \right) F_2(e)de \right] \right] \right]$$

$$\geq \sum_{k \in A} \mathbb{E}_{n_{i,k}} \left[ \frac{1}{\frac{|X_{i,k}|}{\sigma^2} + \frac{1}{\ell^2}} \right] + \sum_{k \in B} \mathbb{E}_{n_{i,k}} \left[ \frac{1}{\frac{|X_{i,k} \cup Y_{-i,k}|}{\sigma^2} + \frac{1}{\ell^2}} \right] + \sum_{k \in C} \mathbb{E}_{n_{i,k}} \left[ \int_{-\infty}^{\infty} F_{1,k}(e) F_2(e)de \right] \tag{59}$$

Here the inner most expectations in the last summation are dropped as $F_{1,k}$ only depends on $|X_{i,k}|, |Z_{i,k}|$, and $|Z_{i,k}'|$ but not their instantiations. Using (59), we can rewrite the Bayes risk under any prior $\Lambda_\ell$ as:

$$R_\ell := \inf_{s_i'} \inf_{s_i' \in \Delta(\mathcal{S})'} \mathbb{E}_{\mu \sim \Lambda_\ell} \left[ \mathbb{E}_{(s_i', s_{-i}^\star)} \left[ \mathbb{E}_{\mu, M_{\mathrm{CBL}}} [\|h_i(X_i, Y_i, I_i) - \mu\|_2^2] \right] \right]$$

$$= \sum_{k \in A} \mathbb{E}_{n_{i,k}} \left[ \frac{1}{\frac{|X_{i,k}|}{\sigma^2} + \frac{1}{\ell^2}} \right] + \sum_{k \in B} \mathbb{E}_{n_{i,k}} \left[ \frac{1}{\frac{|X_{i,k} \cup Y_{-i,k}|}{\sigma^2} + \frac{1}{\ell^2}} \right] + \sum_{k \in C} \mathbb{E}_{n_{i,k}} \left[ \int_{-\infty}^{\infty} F_{1,k}(e) F_2(e)de \right]$$

$$= \sum_{k \in A} \mathbb{E}_{n_{i,k}} \left[ \frac{1}{\frac{|X_{i,k}|}{\sigma^2} + \frac{1}{\ell^2}} \right] + \sum_{k \in B} \mathbb{E}_{n_{i,k}} \left[ \frac{1}{\frac{|X_{i,k} \cup Y_{-i,k}|}{\sigma^2} + \frac{1}{\ell^2}} \right]$$

$$+ \sum_{k \in C} \mathbb{E}_{n_{i,k}} \left[ \mathbb{E}_{e \sim \mathcal{N}(0,1)} \left[ \frac{1}{\frac{|Z_{i,k}'|}{\sigma^2 + \alpha_{i,k}^2 \widetilde{\sigma}^2 e^2} + \frac{|X_{i,k}| + |Z_{i,k}|}{\sigma^2} + \frac{1}{\ell^2}} \right] \right]$$

Because all the terms inside the expectations are bounded and $\lim_{\ell \to \infty} \widetilde{\sigma}_{\ell,k}^2 = \frac{\sigma^2}{|X_{i,k}|} + \frac{\sigma^2}{|Z_{i,k}|}$, we can use dominated convergence to show that

$$R_\infty := \lim_{\ell \to \infty} R_\ell = \sum_{k \in A} \mathbb{E}_{n_{i,k}} \left[ \frac{\sigma^2}{|X_{i,k}|} \right] + \sum_{k \in B} \mathbb{E}_{n_{i,k}} \left[ \frac{\sigma^2}{|X_{i,k} \cup Y_{-i,k}|} \right]$$

$$+ \sum_{k \in C} \mathbb{E}_{n_{i,k}} \left[ \mathbb{E}_{e \sim \mathcal{N}(0,1)} \left[ \frac{1}{\frac{|Z_{i,k}'|}{\sigma^2 + \alpha_{i,k}^2 \left( \frac{\sigma^2}{|X_{i,k}|} + \frac{\sigma^2}{|Z_{i,k}|} \right) e^2} + \frac{|X_{i,k}| + |Z_{i,k}|}{\sigma^2}} \right] \right] \tag{60}$$

**Step 2. (Maximum risk of $\widetilde{s}_i$)** Define $\widetilde{s}_i$ to be the randomized strategy given by $(n_i, f_i^\star, h_i^\star)$. Recall that $n_i$ is the randomized distribution over how much data is collected. Thus, $\widetilde{s}_i$ is the strategy always submits data truthfully and uses the suggested estimator while following the same data collection strategy as $s_i$.

We will now compute the maximum risk of $\widetilde{s}_i$ and show that it is equal to the RHS of (60). First note that we can write,

$$\begin{pmatrix} \widehat{\mu}(X_{i,k}) - \mu_k \\ \widehat{\mu}(Z_{i,k}) - \mu_k \end{pmatrix} \sim \mathcal{N}\left( \begin{pmatrix} 0 \\ 0 \end{pmatrix}, \begin{pmatrix} \frac{\sigma^2}{|X_{i,k}|} & 0 \\ 0 & \frac{\sigma^2}{|Z_{i,k}|} \end{pmatrix} \right)$$

By a linear transformation of this Gaussian vector, we obtain

$$\begin{pmatrix} \frac{|X_{i,k}|}{\sigma^2} \left( \widehat{\mu}(X_{i,k}) - \mu_k \right) + \frac{|Z_{i,k}|}{\sigma^2} \left( \widehat{\mu}(Z_{i,k}) - \mu_k \right) \\ \widehat{\mu}(X_{i,k}) - \widehat{\mu}(Z_{i,k}) \end{pmatrix} = \begin{pmatrix} \frac{|X_{i,k}|}{\sigma^2} & \frac{|Z_{i,k}|}{\sigma^2} \\ 1 & -1 \end{pmatrix} \begin{pmatrix} \widehat{\mu}(X_{i,k}) - \mu_k \\ \widehat{\mu}(Z_{i,k}) - \mu_k \end{pmatrix}$$

$$\sim \mathcal{N}\left( \begin{pmatrix} 0 \\ 0 \end{pmatrix}, \begin{pmatrix} \frac{|X_{i,k}| + |Z_{i,k}|}{\sigma^2} & 0 \\ 0 & \frac{\sigma^2}{|X_{i,k}|} + \frac{\sigma^2}{|Z_{i,k}|} \end{pmatrix} \right)$$

which means $\frac{|X_{i,k}|}{\sigma^2} \left( \widehat{\mu}(X_{i,k}) - \mu_k \right) + \frac{|Z_{i,k}|}{\sigma^2} \left( \widehat{\mu}(Z_{i,k}) - \mu_k \right)$ and $\frac{\eta_{i,k}}{\alpha_{i,k}} = \widehat{\mu}(X_{i,k}) - \widehat{\mu}(Z_{i,k})$ are independent Gaussian random variables. Therefore, the maximum risk of $\widetilde{s}_i$ is:

$$\sup_{\mu \in \mathbb{R}^d} \mathbb{E}_{(\widetilde{s}_i, s_{-i}^\star)} \left[ \mathbb{E}_{\mu, M_{\text{CBL}}} \left[ \| h_i (X_i, Y_i, I_i) - \mu \|_2^2 \right] \right]$$

$$= \sup_{\mu \in \mathbb{R}^d} \mathbb{E}_{n_i} \left[ \mathbb{E}_{\mu, M_{\text{CBL}}} \left[ \sum_{k \in A} (h_{i,k} (X_i, Y_i, I_i) - \mu_k)^2 + \sum_{k \in B} (h_{i,k} (X_i, Y_i, I_i) - \mu_k)^2 + \sum_{k \in C} (h_{i,k} (X_i, Y_i, I_i) - \mu_k)^2 \right] \right]$$

$$= \sup_{\mu \in \mathbb{R}^d} \mathbb{E}_{n_i} \left[ \mathbb{E}_{\mu, M_{\text{CBL}}} \left[ \sum_{k \in A} (\widehat{\mu}(X_{i,k}) - \mu_k)^2 + \sum_{k \in B} (\widehat{\mu}(X_{i,k} \cup Y_{-i,k}) - \mu_k)^2 + \sum_{k \in C} (h_{i,k} (X_i, Y_i, I_i) - \mu_k)^2 \right] \right]$$

$$= \sup_{\mu \in \mathbb{R}^d} \mathbb{E}_{n_i} \left[ \mathbb{E}_{\mu, M_{\text{CBL}}} \left[ \sum_{k \in A} \frac{\sigma^2}{|X_{i,k}|} + \sum_{k \in B} \frac{\sigma^2}{|X_{i,k} \cup Y_{-i,k}|} + \sum_{k \in C} (h_{i,k} (X_i, Y_i, I_i) - \mu_k)^2 \right] \right]$$

$$= \sum_{k \in A} \mathbb{E}_{n_{i,k}} \left[ \frac{\sigma^2}{|X_{i,k}|} \right] + \sum_{k \in B} \mathbb{E}_{n_{i,k}} \left[ \frac{\sigma^2}{|X_{i,k} \cup Y_{-i,k}|} \right] + \sup_{\mu \in \mathbb{R}^d} \mathbb{E}_{n_i} \mathbb{E}_{\mu, M_{\text{CBL}}} \left[ \sum_{k \in C} (h_{i,k} (X_i, Y_i, I_i) - \mu_k)^2 \right] \tag{61}$$

Now notice that we can rewrite the last summation, given in (61), as:

$$\sup_{\mu \in \mathbb{R}^d} \mathbb{E}_{n_i} \mathbb{E}_{\mu, M_{\text{CBL}}} \left[ \sum_{k \in C} (h_{i,k} (X_i, Y_i, I_i) - \mu_k)^2 \right]$$

$$= \sup_{\mu \in \mathbb{R}^d} \sum_{k \in C} \mathbb{E}_{n_i} \left[ \mathbb{E}_{\mu, M_{\text{CBL}}} \left[ (h_{i,k} (X_i, Y_i, I_i) - \mu_k)^2 \right] \right]$$

$$= \sup_{\mu \in \mathbb{R}^d} \sum_{k \in C} \mathbb{E}_{n_i} \left[ \mathbb{E}_{\eta_{i,k}} \left[ \mathbb{E}\left[ \left( \frac{\frac{\widehat{\mu}(Z'_{i,k})|Z'_{i,k}|}{\sigma^2 + \eta_{i,k}^2} + \frac{\widehat{\mu}(Z_{i,k})|Z_{i,k}|}{\sigma^2} + \frac{\widehat{\mu}(X_{i,k})|X_{i,k}|}{\sigma^2}}{\frac{|Z'_{i,k}|}{\sigma^2 + \eta_{i,k}^2} + \frac{|Z_{i,k} \cup X_{i,k}|}{\sigma^2}} - \mu_k \right)^2 \middle| \eta_{i,k} \right] \right] \right]$$

$$= \sup_{\mu \in \mathbb{R}^d} \sum_{k \in C} \mathbb{E}_{n_{i,k}} \left[ \mathbb{E}_{\eta_{i,k}} \left[ \mathbb{E}\left[ \left( \frac{\frac{(\widehat{\mu}(Z'_{i,k}) - \mu_k)|Z'_{i,k}|}{\sigma^2 + \eta_{i,k}^2} + \frac{(\widehat{\mu}(Z_{i,k}) - \mu_k)|Z_{i,k}|}{\sigma^2} + \frac{(\widehat{\mu}(X_{i,k}) - \mu_k)|X_{i,k}|}{\sigma^2}}{\frac{|Z'_{i,k}|}{\sigma^2 + \eta_{i,k}^2} + \frac{|Z_{i,k} \cup X_{i,k}|}{\sigma^2}} \right)^2 \middle| \eta_{i,k} \right] \right] \right]$$

$$= \sup_{\mu \in \mathbb{R}^d} \sum_{k \in C} \mathbb{E}_{n_{i,k}} \left[ \mathbb{E}_{\eta_{i,k}} \left[ \frac{\mathbb{E}\left[ \left( \frac{(\widehat{\mu}(Z'_{i,k}) - \mu_k)|Z'_{i,k}|}{\sigma^2 + \eta_{i,k}^2} + \frac{(\widehat{\mu}(Z_{i,k}) - \mu_k)|Z_{i,k}|}{\sigma^2} + \frac{(\widehat{\mu}(X_{i,k}) - \mu_k)|X_{i,k}|}{\sigma^2} \right)^2 \middle| \eta_{i,k} \right]}{\left( \frac{|Z'_{i,k}|}{\sigma^2 + \eta_{i,k}^2} + \frac{|Z_{i,k} \cup X_{i,k}|}{\sigma^2} \right)^2} \right] \right]$$

$$= \sup_{\mu \in \mathbb{R}^d} \sum_{k \in C} \mathbb{E}_{n_{i,k}} \left[ \mathbb{E}_{\eta_{i,k}} \left[ \frac{1}{\left( \frac{|Z'_{i,k}|}{\sigma^2 + \eta_{i,k}^2} + \frac{|Z_{i,k} \cup X_{i,k}|}{\sigma^2} \right)^2} \left( \frac{|Z'_{i,k}| \left( \sigma^2 + \eta_{i,k}^2 \right)}{\left( \sigma^2 + \eta_{i,k}^2 \right)^2} + \frac{|X_{i,k}| + |Z_{i,k}|}{\sigma^2} \right) \right] \right]$$

$$= \sum_{k \in C} \mathbb{E}_{n_{i,k}} \left[ \mathbb{E}_{\eta_{i,k}} \left[ \frac{1}{\frac{|Z'_{i,k}|}{\sigma^2 + \eta_{i,k}^2} + \frac{|Z_{i,k} \cup X_{i,k}|}{\sigma^2}} \right] \right]$$

$$= \sum_{k \in C} \mathbb{E}_{n_{i,k}} \left[ \mathbb{E} \left[ \frac{1}{\frac{|Z'_{i,k}|}{\sigma^2 + \alpha_{i,k}^2 (\widehat{\mu}(X_{i,k}) - \widehat{\mu}(Z_{i,k}))^2} + \frac{|Z_{i,k} \cup X_{i,k}|}{\sigma^2}} \right] \right]$$

$$= \sum_{k \in C} \mathbb{E}_{n_{i,k}} \left[ \mathbb{E}_{e \sim \mathcal{N}(0,1)} \left[ \frac{1}{\frac{|Z'_{i,k}|}{\sigma^2 + \alpha_{i,k}^2 \left( \frac{\sigma^2}{|Z_{i,k}|} + \frac{\sigma^2}{|X_{i,k}|} \right)} + \frac{|Z_{i,k} \cup X_{i,k}|}{\sigma^2}} \right] \right]$$

Here the last line follows from the fact that $\widehat{\mu}(X_{i,k}) - \widehat{\mu}(Z_{i,k}) \sim \mathcal{N}\left( 0, \frac{\sigma^2}{|X_{i,k}|} + \frac{\sigma^2}{|Z_{i,k}|} \right)$. Substituting this expression into line (61) we get

$$\sup_{\mu \in \mathbb{R}^d} \mathbb{E}_{(\widetilde{s}_i, s_{-i}^\star)} \left[ \mathbb{E}_{\mu, M_{\text{CBL}}} \left[ \|h_i(X_i, Y_i, I_i) - \mu\|_2^2 \right] \right] = \sum_{k \in A} \mathbb{E}_{n_{i,k}} \left[ \frac{\sigma^2}{|X_{i,k}|} \right] + \sum_{k \in B} \mathbb{E}_{n_{i,k}} \left[ \frac{\sigma^2}{|X_{i,k} \cup Y_{-i,k}|} \right] \tag{62}$$

$$+ \sum_{k \in C} \mathbb{E}_{n_{i,k}} \left[ \mathbb{E}_{e \sim \mathcal{N}(0,1)} \left[ \frac{1}{\frac{|Z'_{i,k}|}{\sigma^2 + \alpha_{i,k}^2 \left( \frac{\sigma^2}{|Z_{i,k}|} + \frac{\sigma^2}{|X_{i,k}|} \right)} + \frac{|Z_{i,k} \cup X_{i,k}|}{\sigma^2}} \right] \right] \tag{63}$$

$$= R_\infty$$

Here, we have observed that the final expression in the above equation is exactly the same as the Bayes risk in the limit in (60) from Step 1.

**Step 3. (Minimax optimality of $\widetilde{s}_i$)**   As the maximum is larger than the average, we can write, for any prior $\Lambda_\ell$ and any $s_i' \in \Delta(\mathcal{S})$

$$\sup_{\mu \in \mathbb{R}^d} \mathbb{E}_{(s_i', s_{-i}^\star)} \left[ \mathbb{E}_{\mu, M_{\text{CBL}}} \left[ \|h_i(X_i, Y_i, I_i) - \mu\|_2^2 \right] \right] \geq \mathbb{E}_{\mu \sim \Lambda_\ell} \left[ \mathbb{E}_{(s_i', s_{-i}^\star)} \left[ \mathbb{E}_{\mu, M_{\text{CBL}}} \left[ \|h_i(X_i, Y_i, I_i) - \mu\|_2^2 \right] \right] \right] \geq R_\ell$$

As this is true for all $\ell$, taking the limit, we have that $\forall s_i' \in \Delta(\mathcal{S})'$,

$$\sup_{\mu \in \mathbb{R}^d} \mathbb{E}_{(s_i', s_{-i}^\star)} \left[ \mathbb{E}_{\mu, M_{\text{CBL}}} \left[ \|h_i(X_i, Y_i, I_i) - \mu\|_2^2 \right] \right] \geq R_\infty = \sup_{\mu \in \mathbb{R}^d} \mathbb{E}_{(\widetilde{s}_i, s_{-i}^\star)} \left[ \mathbb{E}_{\mu, M_{\text{CBL}}} \left[ \|h_i(X_i, Y_i, I_i) - \mu\|_2^2 \right] \right]$$

that is, $\widetilde{s}_i = (n_i, f_i^\star, h_i^\star)$ has a smaller maximum risk than any other $s_i' \in \Delta(\mathcal{S})$.

### H.2. Proof of Lemma 4

Since we are assuming that $f_i = f_i^\star, h_i = h_i^\star$, we have that

$$p_i := p_i \left( M_{\text{CBL}}, ((n_i, f_i^\star, h_i^\star), s_{-i}^\star) \right)$$

$$= \sup_{\mu \in \mathbb{R}^d} \mathop{\mathbb{E}}_{(s_i, s_{-i}^\star)} \left[ \mathop{\mathbb{E}}_{\mu, M_{\mathrm{CBL}}} \left[ \|h_i(X_i, Y_i, I_i) - \mu\|_2^2 \right] + \sum_{k=1}^{d} c_{i,k} n_{i,k} \right]$$

$$= \mathop{\mathbb{E}}_{(s_i, s_{-i}^\star)} \left[ \sum_{k=1}^{d} c_{i,k} n_{i,k} \right] + \sup_{\mu \in \mathbb{R}^d} \mathop{\mathbb{E}}_{(s_i, s_{-i}^\star)} \left[ \mathop{\mathbb{E}}_{\mu, M_{\mathrm{CBL}}} \left[ \|h_i(X_i, Y_i, I_i) - \mu\|_2^2 \right] \right]$$

Since $f_i = f_i^\star, h_i = h_i^\star$ we can use the expression we calculated for the maximum risk in lines (62) and (63) to rewrite the penalty as

$$p_i = \mathop{\mathbb{E}}_{n_{i,k}} \left[ \sum_{k=1}^{d} c_{i,k} n_{i,k} \right] + \sum_{k \in A} \mathop{\mathbb{E}}_{n_{i,k}} \left[ \frac{\sigma^2}{|X_{i,k}|} \right] + \sum_{k \in B} \mathop{\mathbb{E}}_{n_{i,k}} \left[ \frac{\sigma^2}{|X_{i,k} \cup Y_{-i,k}|} \right]$$

$$+ \sum_{k \in C} \mathop{\mathbb{E}}_{n_{i,k}} \left[ \mathop{\mathbb{E}}_{e \sim \mathcal{N}(0,1)} \left[ \frac{1}{\frac{|Z'_{i,k}|}{\sigma^2 + \alpha_{i,k}^2 \left( \frac{\sigma^2}{|Z_{i,k}|} + \frac{\sigma^2}{|X_{i,k}|} \right)} + \frac{|Z_{i,k} \cup X_{i,k}|}{\sigma^2}} \right] \right]$$

$$= \sum_{k \in A} \mathop{\mathbb{E}}_{n_{i,k}} \left[ \frac{\sigma^2}{|X_{i,k}|} + c_{i,k} n_{i,k} \right] + \sum_{k \in B} \mathop{\mathbb{E}}_{n_{i,k}} \left[ \frac{\sigma^2}{|X_{i,k} \cup Y_{-i,k}|} + c_{i,k} n_{i,k} \right]$$

$$+ \sum_{k \in C} \mathop{\mathbb{E}}_{n_{i,k}} \left[ \mathop{\mathbb{E}}_{e \sim \mathcal{N}(0,1)} \left[ \frac{1}{\frac{|Z'_{i,k}|}{\sigma^2 + \alpha_{i,k}^2 \left( \frac{\sigma^2}{|Z_{i,k}|} + \frac{\sigma^2}{|X_{i,k}|} \right)} + \frac{|Z_{i,k} \cup X_{i,k}|}{\sigma^2}} \right] + c_{i,k} n_{i,k} \right]$$

$$\geq \sum_{k \in A} \min_{n_{i,k}} \left( \frac{\sigma^2}{|X_{i,k}|} + c_{i,k} n_{i,k} \right) + \sum_{k \in B} \min_{n_{i,k}} \left( \frac{\sigma^2}{|X_{i,k} \cup Y_{-i,k}|} + c_{i,k} n_{i,k} \right)$$

$$+ \sum_{k \in C} \min_{n_{i,k}} \left( \mathop{\mathbb{E}}_{e \sim \mathcal{N}(0,1)} \left[ \frac{1}{\frac{|Z'_{i,k}|}{\sigma^2 + \alpha_{i,k}^2 \left( \frac{\sigma^2}{|Z_{i,k}|} + \frac{\sigma^2}{|X_{i,k}|} \right)} + \frac{|Z_{i,k} \cup X_{i,k}|}{\sigma^2}} \right] + c_{i,k} n_{i,k} \right)$$

To minimize this entire quantity, we minimize each of the terms, making use of the fact that these terms are all separable and each only depend on one of $n_{i,1}, \ldots, n_{i,d}$. We do this by cases, depending on whether $k$ is in $A$, $B$, or $C$.

If $k \in A$, then we know that by definition

$$\operatorname*{argmin}_{n_{i,k}} \frac{\sigma^2}{|X_{i,k}|} + c_{i,k} n_{i,k} = \operatorname*{argmin}_{n_{i,k}} = \frac{\sigma^2}{n_{i,k}} + c_{i,k} n_{i,k} = n_{i,k}^{\mathrm{IND}}$$

Furthermore, since $k \in A$, $n_{i,k}^\star = n_{i,k}^{\mathrm{IND}}$ by the definition given in (9).

If $k \in B$, then $n_{i,k}^\star = 0$. We know that $Y_{-i,k} \geq T_k$ by the definition of Algorithm 1. Since $i \notin V_k$, Lemma 9 tells us that $T_k > 2n_{i,k}^{\mathrm{IND}} > n_{i,k}^{\mathrm{IND}}$ and so $Y_{-i,k} > n_{i,k}^{\mathrm{IND}}$. Therefore, the cost for agent $i$ to collect more data is not offset by the marginal decrease in estimation error due to this extra data. This means that

$$\operatorname*{argmin}_{n_{i,k}} \frac{\sigma^2}{|X_{i,k} \cup Y_{-i,k}|} + c_{i,k} n_{i,k} = \operatorname*{argmin}_{n_{i,k}} = \frac{\sigma^2}{n_{i,k} + |Y_{-i,k}|} + c_{i,k} n_{i,k} = 0$$

Furthermore, since $k \in B$, $n_{i,k}^\star = 0$ by the definition given in (9).

If $k \in C$, then more work is required to show that

$$\underset{n_{i,k}}{\operatorname{argmin}} \left( \underset{e \sim \mathcal{N}(0,1)}{\mathbb{E}} \left[ \frac{1}{\frac{|Z'_{i,k}|}{\sigma^2 + \alpha_{i,k}^2 \left( \frac{\sigma^2}{|Z_{i,k}|} + \frac{\sigma^2}{|X_{i,k}|} \right)} + \frac{|Z_{i,k} \cup X_{i,k}|}{\sigma^2}} \right] + c_{i,k} n_{i,k} \right) = n_{i,k}^{\star}$$

Let $D_{i,k} = T_k - 2n_{i,k}^{\star}$. Now notice that

$$\min_{n_{i,k}} \left( \underset{x \sim \mathcal{N}(0,1)}{\mathbb{E}} \left[ \frac{1}{\frac{|Z'_{i,k}|}{\sigma^2 + \alpha_{i,k}^2 \left( \frac{\sigma^2}{|X_{i,k}|} + \frac{\sigma^2}{|Z_{i,k}|} \right) x^2} + \frac{|X_{i,k}| + |Z_{i,k}|}{\sigma^2}} \right] + c_{i,k} n_{i,k} \right)$$

$$= \min_{n_{i,k}} \left( \underset{x \sim \mathcal{N}(0,1)}{\mathbb{E}} \left[ \underbrace{\frac{1}{\frac{D_{i,k}}{\sigma^2 + \alpha_{i,k}^2 \left( \frac{\sigma^2}{n_{i,k}} + \frac{\sigma^2}{n_{i,k}^{\star}} \right) x^2} + \frac{n_{i,k} + n_{i,k}^{\star}}{\sigma^2}}}_{=: l(n_{i,k}, x; \alpha_{i,k})} \right] + c_{i,k} n_{i,k} \right) \tag{64}$$

We will now show that $l$ is convex in $n_{i,k}$.

$$\frac{\partial}{\partial n_{i,k}} l(n_{i,k}, x; \alpha_{i,k}) - \sigma^2 \frac{1 + \frac{D_{i,k}}{\left( 1 + \alpha_{i,k}^2 \left( \frac{1}{n_{i,k}} + \frac{1}{n_{i,k}^{\star}} \right) x^2 \right)^2} \frac{\alpha_{i,k}^2 x^2}{n_{i,k}^2}}{\left( \frac{D_{i,k}}{1 + \alpha_{i,k}^2 \left( \frac{1}{n_{i,k}} + \frac{1}{n_{i,k}^{\star}} \right) x^2} + n_{i,k} + n_{i,k}^{\star} \right)^2} - \sigma^2 \frac{1 + \frac{D_{i,k} \alpha_{i,k}^2 x^2}{\left( n_{i,k} + \alpha_{i,k}^2 \left( 1 + \frac{n_{i,k}}{n_{i,k}^{\star}} \right) x^2 \right)^2}}{\left( \frac{D_{i,k}}{1 + \alpha_{i,k}^2 \left( \frac{1}{n_{i,k}} + \frac{1}{n_{i,k}^{\star}} \right) x^2} + n_{i,k} + n_{i,k}^{\star} \right)^2}$$

We have that $\frac{\partial}{\partial n_{i,k}} l(n_{i,k}, x; \alpha_{i,k})$ is an increasing function and thus $l(n_{i,k}, x; \alpha_{i,k})$ is convex. Because taking expectation preserves convexity (see Lemma 22), the penalty term as a whole is convex. By Lemma 15 we have that

$$\frac{\partial p_i}{\partial n_{i,k}}(n_{i,k}^{\star}) = -\frac{\sigma^2 D_{i,k}}{64 \alpha_{i,k}^3 \sqrt{T_k} n_{i,k}^{\star}} \left( \frac{4\alpha_{i,k}}{\sqrt{T_k}} \left( \frac{4\alpha_{i,k}^2 T_k}{D_{i,k} n_{i,k}^{\star}} - 1 - c_{i,k} \frac{16\alpha_{i,k}^2 T_k n_{i,k}^{\star}}{\sigma^2 D_{i,k}} \right) \right.$$
$$\left. - \exp\left( \frac{T_k}{8\alpha_{i,k}^2} \right) \left( \frac{4\alpha_{i,k}^2}{T_k} \left( \frac{T_k}{n_{i,k}^{\star}} + 1 \right) - 1 \right) \sqrt{2\pi} \operatorname{Erfc}\left( \sqrt{\frac{T_k}{8\alpha_{i,k}^2}} \right) \right).$$

Now notice that the right hand term has a factor of $G_{i,k}(\alpha_{i,k}) = 0$. This means that (64) is minimized when $n_{i,k} = n_{i,k}^{\star}$. Thus, by combining our analyses for when $k \in A, B$ or $C$, we conclude that $p_i$ is minimized when $n_{i,k} = n_{i,k}^{\star}$.

## I. Known results

In this section, we will state known results that are either used directly or with minimal modifications.

**Lemma 22.** *Let $Y$ be a random variable and $f : \mathbb{R}^2 \to \mathbb{R}$ be a function for which $f(x, Y)$ is convex in $x$ and $\forall x \in \mathbb{R}$, $\mathbb{E}[|f(x, Y)|] < \infty$. Then $\mathbb{E}[f(x, Y)]$ is also convex in $x$.*

*Proof.* For all $x_1, x_2 \in \mathbb{R}$ and $t \in [0, 1]$ we have

$$f(x_1 t + x_2(1 - t), Y) \le f(x_1, Y)t + f(x_2, Y)(1 - t)$$
$$\Rightarrow \mathbb{E}[f(x_1 t + x_2(1 - t), Y)] \le \mathbb{E}[f(x_1, Y)]t + \mathbb{E}[f(x_2, Y)](1 - t).$$

$\square$

**Lemma 23** (Chen et al. (2023)) (A corollary of Hardy-Littlewood). *Let $f$, $g$ be nonnegative even functions such that,*

- *$f$ is monotonically increasing on $[0, \infty)$.*

- *$g$ is monotonically decreasing on $[0, \infty)$, and has a finite integral $\int_{\mathbb{R}} g(x)dx < \infty$.*

- *$\forall a$, $\int_{\mathbb{R}} f(x-a)g(x)dx < \infty$.*

*Then for all $a$,*

$$\int_{\mathbb{R}} f(x)g(x)dx \le \int_{\mathbb{R}} f(x-a)g(x)dx$$

**Lemma 24** (Chen et al. (2023)). *For all $t \ge 0$ and some $L > 0$, define*

$$I(t) := \int_{-\infty}^{\infty} \frac{1}{L + x^2} \frac{1}{\sqrt{2\pi}} \exp\left(-tx^2\right) dx$$

$$J(t) := \int_{-\infty}^{\infty} \frac{1}{(L + x^2)^2} \frac{1}{\sqrt{2\pi}} \exp\left(-tx^2\right) dx.$$

*Then we have*

$$I(t) = \exp(Lt)\mathrm{Erfc}(\sqrt{Lt})\sqrt{\frac{\pi}{2L}}$$

$$J(t) = \sqrt{\frac{\pi}{2L}} \left(\frac{1}{2L} - t\right) \exp(Lt)\mathrm{Erfc}(\sqrt{Lt}) + \frac{\sqrt{t}}{\sqrt{2L}}.$$

