# OpenReview forum: "Collaborative Mean Estimation Among Heterogeneous Strategic Agents: Individual Rationality, Fairness, and Truthful Contribution"
_ICML.cc/2025/Conference — ICML 2025 poster_

### Official Review · Reviewer_ZK8A · 2025-03-10

**Overall Recommendation:** 4

**Summary:**

The paper studies a mechanism design problem for agents gathering and sharing data. The authors formalize a model for agents that collect data, with some cost, and distribute it to others, potentially in a strategic manner. They demonstrate a mechanism with satisfies Nash incentive-compatibility, individual rationality, and $\sqrt{m}$-efficiency (Theorem 1, Algorithm 2). The authors go on to discuss hardness results for mechanisms with stricter properties, as well as fair solutions determined by bargaining.

**Claims And Evidence:**

The authors prove five theorems:

* (Thm 1) Alg 2 is NIC, IR, and $\sqrt{m}$-efficient.

* (Thms 2-3) Any mechanism M for the setting that is either --  always IR or dominant strategy -- has each agent's individual pentality of at least $\frac{p_i^{IND}}{2}$.

* (Thm 4): Price of stability lower-bound.

* (Thm 5): 8-approximation on efficiency for bargaining-like solutions.

These results seem thorough for this type of paper.

**Essential References Not Discussed:**

Since your paper discusses data learned across independent parties, this seems like a federated learning task where agents share raw data rather than summary parameter vectors. In general, FL is studied because of the high communication overhead for sharing so much data and in order to conserve privacy of the underlying data. How is this addressed in your current study? I'd like to see some discussion on how your work builds upon other mechanism design literature in federated learning [1], strategic classification [2], or data sharing [3, 4].

[1] Zhan, Y., Zhang, J., et al. (2021). A Survey of Incentive Mechanism Design for Federated Learning.

[2] Hardt, M., Megiddo, N., et al. (2016). Strategic classification.

[3] Zhang, X., Yang, Z., et al. (2014). Free market of crowdsourcing: Incentive mechanism design for mobile sensing.

[4] Tu, X., Zhu, K., et al. (2022). Incentive mechanisms for federated learning: From economic and game theoretic perspective.

**Experimental Designs Or Analyses:**

Experiments do appear thorough in visually demonstrated 5 different solution concepts pertaining to this work, including (a) working along, (b) minimizing social penalty without caring for IR, (c) minimizing social penalty subject to IR constraints, and (d--e) bargaining solutions. However, there is no "experiments" section which discusses how the data for Figure 1 was generated (except perhaps Appendix D, though this should be described further in the main text).

**Methods And Evaluation Criteria:**

The authors provide sufficient benchmarks with respect to (1) the number of samples and individual penalties that agents would draw by working alone, and (2) comparisons against the sample mean mechanism. Baselines are used to establish bounds on individual rationality.

**Other Comments Or Suggestions:**

Minor:
- Is the sub-title on each page supposed to be different than the main title?
- Why is some text blue in Section 2?
- Is the text following $n_i^*$ on page 8 correct?
- In "Agent's penalty" section, clarify that $\mu'$ is a fixed and commonly known distribution.
- In "Truthful submissions" paragraph, clarify that $s^*$ denotes the truthful submission and estimate, corresponding with $id$ and $h^{ACC}$.
- Please be consistent with "Fig. 1" and "Fig 1".
- In "Algorithm 2 walk through" paragraph, if you are including $G_{i,k}$ (defined in Appendix C), please at least mention what it represents. However, since this variable does not appear to be used in the main text, only in the proof (Appendix F), I would not recommend mentioning it in the main text.
- Paragraphs "Baseline for efficiency," "From compliant to strategic agents," and "Fairness considerations" are somewhat redundant to other material presented earlier in the paper. This content can be condensed if you are running out of space elsewhere.
- Please define "price of stability" and cite [1]
- Section 6: please replace "fairness" with "equitable." These terms are not synonymous.

**Other Strengths And Weaknesses:**

Strengths:
- The paper presents mechanisms to address problems of free-riding and data manipulation in the setting of data sharing. The authors conduct a through review of several problems in this area, including (1) dishonesty of the agents, (2) ability for agents to change their estimator functions $h_i(\cdot)$, (3) equitability of agents sampling the data.
- Contributions appear novel, significant, and technically rigorous.
- The paper is well-organized with its headings. Notation is clear and well-defined.

Weaknesses:
- The latter portion of Section 4, through Section 5, is harder to read than the rest of the paper, especially with a lot of technical notation. The authors employ high-level concepts, such as "leverage," without describing the intuition. Several notation for numbers of samples are introduced, such as
  * $\tilde{n}_{i,k}$,
  * $n_i^*$, and
  * $n^{L}_{i,k}$

without clearly distinguishing why these are necessary.
- How large is $n^{OPT}$, or how complicated is solving Eq (7)? This seems significant through Sections 4 and 5, though isn't properly discussed.
- Can you clarify the purpose of Algorithm 1 and what it returns?
- What does "enforceable" mean?

[1] Anshelevich et al. (2004). The price of stability for network design with fair cost allocation.

**Questions For Authors:**

- How does your constraint that the mechanism cannot use money change the setting? In "Problem formalism," you suggest that lowest-cost agent $i^*$ could collect the most data. It seems that, if this data were valuable to other agents, they would pay $i^*$. This makes sense from a labor specialization perspective: it satisfies each agent's needs in an IR manner with the minimal social cost.
- It seems that the objective vector $\mu$ has no covariance by assumption -- i..e, the d dimensions are independent of each other. In this case, it would seems that your setting is really d independent settings, though with possibly different costs across the agents for each dimension. Is this addressed in your theory, or am I mistaken? It seems that your experiments focus on the d=1 dimension case.
- In Agent's penalty, why are we suggesting that the cost $c_{i,k}$ is of the same currency as the L2 gain that agents get by being correct? Is this a good assumption, or just a mathematical convenience as a regularization variable?
- How are the allocations of Figure 1 produced? This seems to correspond to Appendix D, but this isn't described in the main text.
- Does Theorem 4 imply that the efficiency bound found by Theorem 1 (and Algorithm 2) is tight?

**Relation To Broader Scientific Literature:**

The papers focus is on free-riding problems in data sharing settings, so they provide some background (in Sec 2 and Appendix A) about free-riding, cooperative game theory approaches to data contributions, and data sharing with monetary incentives. There are not too many references, but it seems thorough enough for this type of work. This paper significantly builds on Chen et al. (2023).

**Theoretical Claims:**

The proofs appear correct as I skimmed them, though I did not check the appendix too closely for possible errors.

---

> ### Author Rebuttal · Authors · 2025-04-01
>
> Thank you for your response and questions.
>
> **Experimental Designs Or Analyses:**
>
> - *Creation of Figure 1:* The plot was created using a small amount of Mathematica code to solve each of the 5 optimization problems and a minimal amount of Python code to plot the results. We will update the paper to include this in the final version.
>
> **Essential References Not Discussed:**
>
> - *Discussion of federated learning, strategic classification, and data sharing:* These are all valid practical considerations in data sharing but for the sake of simplicity, given that our model is already technically challenging, we leave these for future work. We will make sure to discuss how our mechanism relates to the federated learning, strategic classification, and data sharing references you have provided in the updated version of the paper.
>
> **Other Strengths And Weaknesses:**
>
> - *Technical notation and notion of leverage:* We describe the mechanism as having "leverage" when condition (8) is satisfied for $(c,n)$. At a high level this means there is enough data from each agent to cross validate all the submissions. The notation mentioned is to try and emphasize the process by which the recommended strategies are determined (see bullet point 3). We will more clearly motivate and clarify these terms and notation in the revision.
>
> - *Solving for $n^{OPT}$:* The scale of $n^{OPT}$ is determined by the problem parameters (e.g. $c,\sigma,m$, etc). For example, lower costs lead to increases in $n^{OPT}$ as it is now worth it to collect more data. Solving for $n^{OPT}$ is a convex optimization problem and thus can be handled by tools like Mathematica (which we did for Fig. 1). We will highlight this practical consideration in the revision.
>
> - *Clarifying Algorithm 1:* Our mechanism can handle different bargaining solutions to suite the mechanism designer's preference. However, not every bargaining solution is feasible. For example, having the cheapest agent collect all the data maximizes social welfare but leaves the mechanism with no way to validate their submission. Thus, having the cheapest agent submit this quantity of data truthfully is not IC. Given a bargaining solution $n$, the purpose of Algorithm 1 is to compute an approximation, $\widetilde{n}$, where agents collecting data according to $\widetilde{n}$ and submitting it truthfully is IC. Along with $\widetilde{n}$, Algorithm 1 returns two
> sets $(V_k)_{k=1}^d$, $(T_k)_{k=1}^d$ which record information about $\widetilde{n}$ for technical convenience to be used in Algorithm 2.
>
> - *Enforceability:* By saying $n$ is enforceable, we mean that agents collecting data according to $n$, submitting it truthfully, and accepting the estimate from the mechanism forms a Nash equilibrium. Some bargaining solutions such as having the cheapest agent collect all the data do not result in a Nash equilibria so we would say that bargaining solution is not enforceable by the mechanism. We will elaborate on and clarify this in the revision.
>
> **Other Comments Or Suggestions:**
>
> - *Minor comments:* We will address the minor comments as recommended in the revision. We assume that in the third bullet point you are referring to the first few lines at the start of page 8? Here we mean to say that a large value of $\alpha_{i,k}$ may be needed to ensure IC but result in poor efficiency. We will make this clearer in the revision.
>
> **Questions For Authors:**
>
> - *The constraint of money:* Using money may simplify some of the calculations as payment is now replacing the estimation error. However, there is still the problem of incentivizing truthful data collection. For example, paying the cheapest agent to collect all of the data leaves the mechanism with no way to validate their submission. Therefore, the tradeoff between efficiency and a mechanism's ability to validate submissions as a result of heterogeneity remains.
>
> - *Independent dimensions:* Yes, the problem model is that agents wish to learn the means of $d$ unrelated distributions. We use a multivariable gaussian with identity covariance to represent this. You are correct that figure is limited to $d=1$ because we wanted to provide a simple example of how different bargaining solutions can lead to asymmetric divisions of work and penalties.
>
> - *Currency of collection costs:* We are assuming that the costs are normalized relative to the error so that it makes sense to add them when defining the penalty.
>
> - *Creation of Figure 1:* Please see "Experimental Designs Or Analyses" section of the rebuttal.
>
> - *Tightness of the efficiency bound:* Yes, Theorem 4 implies the efficiency bound is tight up to constants.

---

> > ### Comment · Reviewer_ZK8A · 2025-04-08
> >
> > Regarding Figure 1 -- Sorry, I did not mean the specific Mathematica / Python code used to create the figures. I meant that Figure 1 needs a few sentences in the main body describing what each of the sub-figures are and what Monte Carlo simulation you used to generate the allocations. As written, this is currently skimmed over. Some details are provided in Appendix D, but this does not appear explained thoroughly. Including an explication of Figure 1 is one aspect where you can improve the paper quality.

---

### Official Review · Reviewer_3gvh · 2025-03-12

**Overall Recommendation:** 3

**Summary:**

This paper studies a scenario of data sharing among players interested in estimating the mean of a vector $\mu \in \mathbb{R}^d$ through samples from a gaussians with mean $\mu_k, k \in [d]$. The fact that the players can exchange data (which could happen in practice if some players have specific advantages to collect specific kinds of data) requires several conditions to ensure favorable outcomes. The mechanism needs to ensure Individual Rationality (i.e. the players are better off by joining the process), Incentive Compatibility (i.e., the players are better off by being trustful) and producing "nice/fair" outcomes. Typically, the mechanism needs to prevent strategic behaviours (players misbehaving in order to improve their own personal utility).

A mechanism (designed by a principal who coordinates the players) is a tuple given by an information set $\mathcal{I}$ and a map from the datasets given by the agents to some information released to the agents. On the players' side, the strategy consists of a number of samples collected, a map that transforms the collected samples before submitting them to the mechanism designer and finally an estimator that also takes into account the information released by the principal to the agent. Of course, the agents have costs depending on the coordinate of the vector they sample and can strategically choose the components of their strategy.

The goal of the mechanism is to minimize the global penalty (kind of a non-decreasing function of the individual costs) under the Individual Rationality constraint.

**Claims And Evidence:**

Yes.

**Essential References Not Discussed:**

Yes, essentially these two references seem crucial to me:

- Capitaine et al., Unravelling in collaborative learning: this paper studies a mechanism so that rational and strategic agents can share data to the principal and receive a trained model. The best the data they provide is, the best is the model they receive. Similar questions about what happens at the Nash Equilibrium are studied and both setups seem close to me.

- Donahue et al., Model-sharing Games: Analyzing Federated Learning Under Voluntary Participation is a classic reference that studies collaborative inference under constraints on the participants' behaviors.

**Experimental Designs Or Analyses:**

NA.

**Methods And Evaluation Criteria:**

No evaluation criteria, the only example is an illustrative graph in dimension 1 but I do not feel like intensive experiments are needed to support the paper.

**Other Comments Or Suggestions:**

I do not believe that experiments are necessary for the paper. However, providing setups where this data sharing situation could happen would be absolutely great and how the proposed mechanism would experimentally compare to what exists would be a real plus.

**Other Strengths And Weaknesses:**

To me, the major weakness of the paper is the lack of novelty of the setup since designing mechanisms for trustful sharing of data among [potentially] strategic agents has already been introduced an studied before.

One the other side, the paper provides a good understanding of how it relates to the existing literature, acknowledges the differences between the previous setups and provides thorough theoretical foundations for the claims.

I also have to say that the paper is very correctly written.

**Questions For Authors:**

Could you explain more precisely how the heterogeneous costs among the agents make a fundamental difference with the work of Chen et al., Mechanism Design for Collaborative Normal Mean Estimation?

Could you discuss the current mechanisms that are implemented for data sharing in practice nowadays (in more details than the statement line 20)?

**Relation To Broader Scientific Literature:**

The problem tackled in this paper is interesting but not new and has already been studied in the literature. I appreciate the fact that the reference Chen et al., Mechanism Design for Collaborative Normal Mean Estimation has its own paragraph discussion in the related works. The paper claims that considering heterogeneous costs between the agents for the data collection raises "new IR and fairness challenges" but I would enjoy a more thorough discussion about the fundamental differences that are implied by the heterogeneous costs (while the appendix "Extended related works" mostly rephrase the paragraph from the main).

**Theoretical Claims:**

I checked as I could the theoretical proofs of the paper's theorems but I must admit that I did not go through all the details of the proofs in the supplementary. From what I checked, things seemed correct.

---

> ### Author Rebuttal · Authors · 2025-04-01
>
> Thank you for your response and questions.
>
> **Relation To Broader Scientific Literature:**
>
> - *New challenges from heterogeneity:* We apologize for not highlighting the new challenges that heterogeneity introduces well enough due to space constraints. Please see "comparisons with Chen et al. 2023" in the response to reviewer MzVR. We will ensure this is better highlighted in the revision.
>
> **Essential References Not Discussed:**
>
> - *Capitaine et al. and Donaheu et al.:* Thank you for bringing these to our attention. We will add them to the related work section in the revision. However, we wish to emphasize that neither Capitaine et al. nor Donaheu et al. address the problem of truthful reporting and both assume a more restrictive strategy space than the one we consider. Additionally, in Donahue et al. agents are drawing their data from different distributions and do not share the data itself whereas we utilize the fact that agents are drawing data from the same distributions and share data drawn from those distributions.
>
> **Other Strengths And Weaknesses:**
>
> - *Novelty of the setup:* Please see "comparisons with Chen et al. 2023" in the response to reviewer MzVR.
>
> **Other Comments Or Suggestions:**
>
> - *Real world applications of our model:* Data sharing consortia exist among research labs, hospitals, and freight/wireless spectrum monitoring/tech companies. Examples include PubMed, PubChem, CLARiTi, FLOW, BigSpec, and DeltaSharing. In particular,  BigSpec for wireless spectrum monitoring shares average power usage across different wireless spectrum bands, and is akin to our mean estimation problem.
>
>   To our knowledge, these systems and the references in line 20 rely on trust, assuming agents will always report truthfully. Mechanisms that rely on trust are susceptible to free-riding wherein participants may forgo the work of data collection and instead rely on the data collected by other participants. This can discourage agents from participating in data sharing. On the other hand our mechanism discourages agents from free-riding and incentivizes them to collect and contribute a sufficient amount of data truthfully.
>
> **Questions For Authors:**
>
> - *Novel challenges introduced by heterogeneity:* Please see "comparisons with Chen et al. 2023" in the response to reviewer MzVR.
>
> - *Current mechanisms for data sharing (more than line 20):* Current platforms fall into one of two categories. First, in some, agents contribute data and others can use this data (e.g PubChem, PubMed). In other examples, participants all submit data to a centralized planner who reallocates the data received (eg. BigSpec, CLARiTi). To our knowledge, none consider truthful reporting, nor do they consider the costs involved with data collection.

---

> > ### Comment · Reviewer_3gvh · 2025-04-05
> >
> > I thank the authors for their detailed and interesting answers: I increase my score.

---

### Official Review · Reviewer_7Eg1 · 2025-03-13

**Overall Recommendation:** 3

**Summary:**

This paper proposes a novel collaborative learning mechanism to solve the problem of individual rationality, fairness, and strategic behavior among heterogeneous strategic agents. Its contributions include mechanism design, approximate ratio analysis, hardness results, and fairness comparison.

**Claims And Evidence:**

The claims are convincing and have sufficient theoretical support.

**Essential References Not Discussed:**

N/A

**Experimental Designs Or Analyses:**

No experimental designs since it is a purely theoretical work.

**Methods And Evaluation Criteria:**

The proposed methods are technically sound and have sufficient theoretical proofs as the support.

**Other Comments Or Suggestions:**

Please refer to weakness.

**Other Strengths And Weaknesses:**

Strength:

1. This paper addressed an important problem that is ensuring individually rational and fair outcomes so all agents benefit and preventing socially undesirable outcomes in a multi-agent collaboration system.

2. The proposed collaboration mechanism with social penalty is technically sound.

3. Sufficient theoretical proofs are provided.

Weakness:

1. The model assumes that the collection cost of the agent is a known constant and that data sharing is frictionless. However, there may be dynamic costs, privacy constraints, or transmission delays in real-world scenarios. Failure to discuss the impact of these limitations on the conclusions, such as robustness analysis, may undermine the model's usefulness.

2. The trade-off between fairness and performance / efficiency sacrifice remained unknown.

3. The communication overhead of data exchange among agents has not yet been discussed, which is problematic in real-time collaboration.

**Questions For Authors:**

Please refer to the weakness.

**Relation To Broader Scientific Literature:**

Ensure individually rational and fair outcomes so all agents benefit and prevent socially undesirable outcomes in a multi-agent collaboration system.

**Theoretical Claims:**

I only checked part of the theoretical proofs, and they are correct. I didn't check all the theoretical proofs.

---

> ### Author Rebuttal · Authors · 2025-04-01
>
> Thank you for your response and questions.
>
> **Other Strengths And Weaknesses:**
>
> - *Unknown costs, privacy, communication costs:* These are important considerations in real-world systems but because our current model is already technically challenging, we leave these for future work.
>
> - *Fairness and efficiency tradeoff:* The tradeoff between fairness and efficiency is a natural question in many multi-agent systems. Axiomatic bargaining solutions are designed specifically to handle such trade-offs. In Section 6, we allow the mechanism designer to specify their favorite bargaining solution (which leaves it to them to determine how to trade-off fairness and efficiency). Our focus instead is on incentivizing truthful reporting while approximating this bargaining solution. To summarize, while we do not explicitly consider fairness vs efficiency, our framework allows a mechanism designer to handle this trade-off by choosing a bargaining solution.
>
> - *Communication overhead of data exchange:* Because our mechanism is a one time algorithm for data sharing we do not anticipate communication overhead being a practical issue and leave this consideration for future work.

---

> > ### Comment · Reviewer_7Eg1 · 2025-04-02
> >
> > Thank you very much for the response. I will keep my score.

---

### Official Review · Reviewer_MzVR · 2025-03-13

**Overall Recommendation:** 3

**Summary:**

This paper studies collaborative learning among multiple heterogeneous strategic agents.  There is a $d$ dimensional isotropic Gaussian with an unknown mean.  Each agent has a cost to derive a sample for each coordinate of the Gaussian, but also wants to reduce the square error of mean estimation.  The paper wants to design a mechanism that recommends the number of samples for each agent and returns some information for each agent.  An ideal mechanism should minimize the social cost, be individually rational (willing to participate), and be incentive compatible (e.g., truthful reporting).  The paper provides a mechanism that has a Nash equilibrium that minimizes social cost under IR constraint and further $O(\sqrt{m})$ approximation ratio to the minimum social cost without IR constraint.  The paper further shows that their mechanism can enforce different recommended numbers of samples to tailor objectives other than social cost.

**Claims And Evidence:**

Yes

**Essential References Not Discussed:**

Nothing critical.

One related work can be Kong, Yuqing, et al. "Information elicitation mechanisms for statistical estimation." Proceedings of the AAAI Conference on Artificial Intelligence. Vol. 34. No. 02. 2020, which considers eliciting truthful Gaussian signal.  However, agents do not have cost and need monetary reward.

**Experimental Designs Or Analyses:**

No

**Methods And Evaluation Criteria:**

Yes

**Other Comments Or Suggestions:**

It seems IC generalizes IR in this paper.  The discussion about prior work overlooking IR before section 1.1 confuses me.

**Other Strengths And Weaknesses:**

Strength:
- The observation of fair issue when agents have heterogeneous cost is interesting.
- The paper further shows that the mechanism can enforce any bargaining solution is interesting.
- Bounding the social cost gap with and without IR is nice.

Weakness:
- The main technique of the paper seems to mostly follows Chen et al. 2023, which solve for one dimensional homogeneous setting.  I feel this is a miss opportunity, as considering general Gaussian setting may inspire new technique.
- The technique heavily depends on the Gaussian assumption and square error setting.  It would be good if the paper discusses how to generalize the results to other setting, e.g., exponential family.

**Questions For Authors:**

Does the mechanism work for non-isotropic Gaussian?

**Relation To Broader Scientific Literature:**

At high level, this work can be viewed as information elicitation without money.  The main techniques are closely related to Chen et al. 2023.

**Theoretical Claims:**

I did not formally check the proof.

---

> ### Author Rebuttal · Authors · 2025-04-01
>
> Thank you for your response and questions.
>
> **Essential References Not Discussed:**
> Thank you for the suggestions. We will update the paper to include this reference.
>
> **Other Strengths And Weaknesses:**
> - *Comparisons with Chen et al. 2023:*
> We would like to emphasize the two (orthogonal) challenges that need to be tackled to incentivize truthful contributions while maintaining an efficient system.
>     (1) The first (addressed by Chen et al.), is to design methods to validate an agent's submission using others' data. Their approach compares an agent's data to others' submissions and ensures a Nash equilibrium (NE) where everyone submits truthfully.
>     (2) The second, **not addressed by Chen et al.** and the focus of this paper, is to ensure that there is enough data from *all* agents so that *each* agent's submission can be sufficiently validated against the others, without compromising on efficiency (social penalty).
>
>   Chen et al. avoid the second issue by assuming a single distribution and equal costs across agents (i.e., homogeneous agents), allowing equal data contribution and ensuring sufficient data from all agents. However, this breaks down with heterogeneous data collection costs. Our paper focuses on this second challenge. As noted in lines 307–309, we adopt their method (Algorithm 2, lines 20–23) to address challenge 1, though we must overcome some technical hurdles to do so.
>
>   The following example illustrates why challenge 2 is non-trivial. Even with $d=1$, unequal costs make it most efficient for the cheapest agent (say, agent 1) to collect all the data. However, this raises two issues: (i) As discussed in Section 3.2, this may violate IR for agent 1. (ii) As covered in Section 4, applying Chen et al.’s solution to challenge 1 requires sufficient data from other agents. Without their contributions, agent 1 could fabricate data, making truthful reporting hard to enforce. The same intuition applies even in the social penalty minimizing allocation of work where the cheapest agent collects most but not all of the data. Again, the mechanism may not have enough data from the other agents to cross validate the submission and incentivize truthful reporting. These considerations do not arise in Chen et al., as agents have the same collection cost so the work of data collection can be equally divided which eliminates these challenges.
>
>   These challenges worsen with multiple distributions ($d > 1$). For example, suppose there are $m$ agents and two distributions: agent 1 faces high cost for the second distribution, while agents $2,\dots,m$ face high cost for the first. Then, agents $2,\dots,m$ depend on agent 1 for data from the first distribution, but the mechanism cannot validate agent 1’s data, as no one else can sample distribution 1. This intuition underlies the $\Omega(\sqrt{m})$ hardness result in Theorem 4, unique to the heterogeneous case - in contrast, Chen et al. always achieve an $O(1)$ approximation.
>
>   The main algorithmic challenge is dividing the work to (a) maintain a large enough group for data verification, and (b) keep data collection costs low to minimize social penalty. These goals conflict, as (a) may require costly agents to collect more, increasing the social penalty. While we rely on Chen et al. for challenge 1 (lines 19–23 of Algorithm 2), the rest of Algorithm 2 and all of Algorithm 1 are novel. As reviewer ZK8A noted, this paper builds significantly on Chen et al.
>
>   In addition to solving the aforementioned challenges, we provide supplementary hardness results in Theorems 2 and 3. These results also apply to the setting studied by Chen et al. (but are not presented in their work).
>
> - *Generalizing to new distributions and losses:* We study the problem of normal mean estimation because Chen et al. have solved challenge 1 for this problem and our focus is on challenge 2. To the best of our knowledge, Chen et al. provide the only solution to challenge 1 under general strategy spaces where agents can submit untruthful data and post-process the information returned by the mechanism. If methods for solving challenge 1 were developed for different distributions or losses, we believe our techniques to solve challenge 2 could be applied in this new setting.
>
> **Other Comments Or Suggestions:**
> - *IC and IR:* Apologies for the confusion, we meant to say that defining IR in our setting is nontrivial. We will update the writing to make this more clear.
>
> **Questions For Authors:**
> - *Non-isotropic Gaussians:* Our setting assumes that $m$ different agents wish to learn $d$ unrelated normal distributions which we represent as learning an isotropic gaussian. A non-isotropic gaussian would correspond to the agents learning the means of correlated distributions so the mechanism would need to be adapted for this new problem setting.

---

> > ### Comment · Reviewer_MzVR · 2025-04-07
> >
> > Thank you very much for the response. I will keep my score.

---

### Decision · Program_Chairs · 2025-05-01

**Decision:**

Accept (poster)

**Comment:**

The paper studies a data sharing problem in collaborative learning where each agents can get some normally distributed estimates of a mean (in multi-dimension), with a cost that is heterogeneous across agents. The paper proposes an approximate mechanism for social cost minimization, proves hardness results, and analyzes variants that are more "fair" (in the repartition of cost amongst agents, formalized with the concept of axiomatic bargaining).

All reviewers found that the paper tackles an important problem and has strong theoretical results on it. The fairness part was particularly appreciated as the most novel. On the downside, it was noted that the methods heavily rely on Chen et al. 2023, so that the novelty in the proofs is somewhat incremental. Nevertheless, the consensus was that the results are worth acceptance.